# Proactive Defense Benchmark against Deepfake Generation

Joonhyuk Baek [* 1]   Wonjune Seo [* 2]   Jae-yun Kim [1]   Saerom Park [1 3]   Hoki Kim [4]

## Abstract

Despite the proliferation of proactive defenses against deepfakes, the lack of a unified evaluation protocol precludes fair comparison and masks critical vulnerabilities. To bridge this gap, we present the first comprehensive benchmark that systematically assesses disruption, robustness, and transferability encompassing pixel, perceptual, and identity metrics. Our extensive analysis reveals that fidelity and identity metrics capture orthogonal performance axes, often leading to conflicting interpretations when relied upon individually. Furthermore, we identify a fundamental trade-off where peak white-box performance signals overfitting, and we introduce a calibrated evaluation to correct generator-induced identity bias. By exposing these blind spots, we establish a rigorous standard to guide the development of genuinely generalizable protections. Project page is available at: https://proactivedefensebenchmark.github.io/

## 1. Introduction

Since the emergence of deepfake generation techniques (Tolosana et al., 2020), publishing facial images online has inherently exposed individuals to the risk of malicious misuse (Harris, 2018; Chesney & Citron, 2019). As deepfake generation becomes increasingly accessible and realistic, post-hoc detection methods (Choi et al., 2024; Ba et al., 2024; Tan et al., 2024; Yang et al., 2025) alone are often insufficient to mitigate harm, as they operate only after manipulated content has already been created. This has motivated increasing interest in proactive defense (Li

---
[*]Equal contribution [1]Department of Industrial Engineering, Ulsan National Institute of Science and Technology (UNIST), Ulsan, South Korea [2]Department of Computer Science and Engineering, Ulsan National Institute of Science and Technology (UNIST), Ulsan, South Korea [3]Artificial Intelligence Graduate School, Ulsan National Institute of Science and Technology (UNIST), Ulsan, South Korea [4]Department of Industrial Security, Chung Ang University, Seoul, South Korea. Correspondence to: Saerom Park <srompark@unist.ac.kr>, Hoki Kim <hokikim@cau.ac.kr>.

*Proceedings of the 43rd International Conference on Machine Learning*, Seoul, South Korea. PMLR 306, 2026. Copyright 2026 by the author(s).

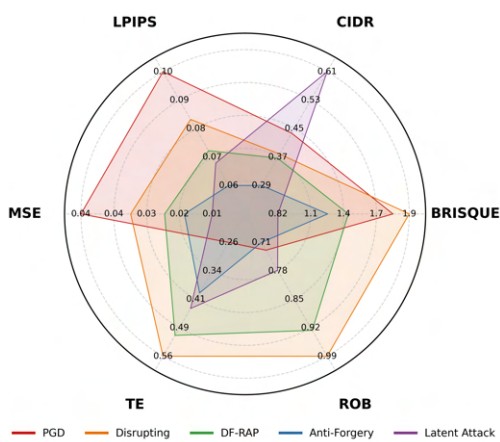

*Figure 1.* Performance of proactive defenses on SimSwap using multiple evaluation metrics. Additional results for other generators are provided in Appendix B.5.

et al., 2024), where images are protected before publication to interfere with deepfake generation itself.

Recent years have seen a rapid diversification of proactive defense strategies (Ruiz et al., 2020; Zhang et al., 2024; Lee et al., 2025). However, unlike the domain of deepfake detection where unified protocols and comprehensive benchmarks are well-established (Yan et al., 2023; Deng et al., 2024; Le et al., 2025), the evaluation for proactive defense remains critically fragmented. Although recent surveys (Nguyen-Le et al., 2025) exist, they primarily focus on methodological taxonomy rather than providing comprehensive empirical benchmarks. Consequently, existing studies rely on incompatible choices of metrics and generators, rendering cross-paper comparisons ambiguous and difficult. Furthermore, current evaluations frequently overlook the critical gap between ideal settings and real-world deployment. In practice, facial images are frequently subjected to transformations such as JPEG compression and degradation (Pasquini et al., 2021). While these transformations can substantially degrade the effectiveness of proactive defense (Guo et al., 2017; Tian et al., 2024), their effects are often neglected in current protocols. Moreover, assessments of transferability are often misleading. While some prior studies address black-box scenarios, they predominantly rely on a narrow set of target generators or those sharing similar architectures (Wang et al., 2022; Qu et al.,

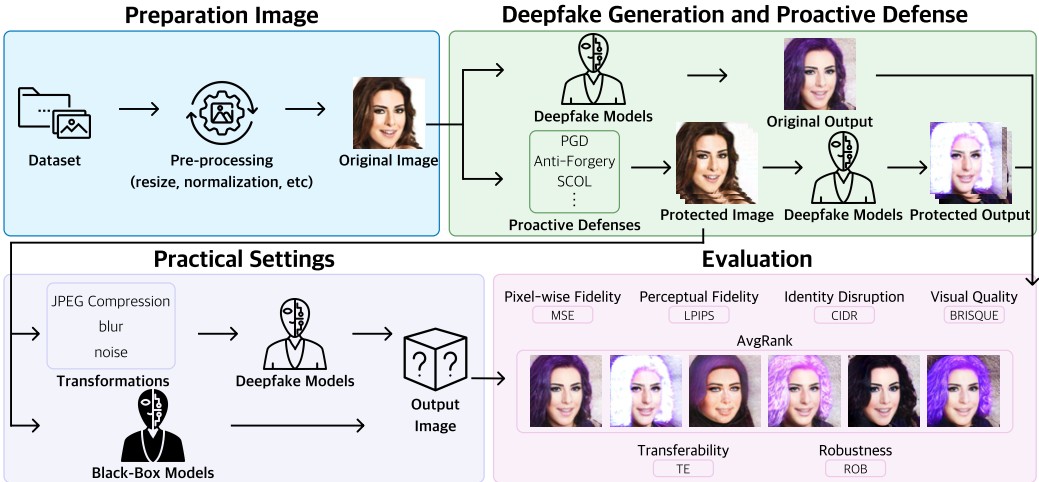

*Figure 2.* Overview of the proactive defense benchmark considered in this work. The figure illustrates the end-to-end evaluation pipeline, from image preparation and proactive defense to deepfake generation, robustness and transferability analysis, and evaluation.

2024; Shim & Yoon, 2025). This restricted evaluation scope masks the severe performance collapse that typically occurs against architecturally diverse models, thereby risking a significant overestimation of defense generalizability against realistic, unknown threats (Dong et al., 2023).

To address these limitations, establishing a comprehensive benchmark is essential not only for ensuring fair comparison but also for uncovering the complex trade-offs between disruption, robustness, and transferability, as reflected in Figure 1. Through this unified protocol, we aim to shift the research focus from maximizing performance in isolated scenarios to developing defenses that are genuinely robust and generalizable against diverse practical threats.

Our main contributions are summarized as follows:

- We present the first comprehensive benchmark for proactive defense that systematically defines and assesses the complex correlations between the three key factors: **Disruption**, **Robustness**, and **Transferability**. This work provides a rigorous standard for the integrated assessment of diverse defense mechanisms.
- We concretize the definition of "disruption" by establishing a comprehensive taxonomy of evaluation metrics across four key dimensions: **Pixel-level Fidelity**, **Perceptual Fidelity**, **Identity Disruption**, and **Visual Quality**.
- Through extensive empirical experiments, we identify critical blind spots in current practices, revealing a **fundamental trade-off** between optimization specificity and cross-model generalizability. We demonstrate that fidelity and identity metrics capture distinct performance axes, and that peak white-box performance often signals overfitting rather than genuine protection, underscoring the necessity of a balanced evaluation protocol.

## 2. Related Work

**Deepfake Generation Models.** Deepfake generation models exhibit significant diversity in their synthesis objectives and architectural designs, which leads to fundamental differences in how facial information is manipulated (Choi et al., 2018; Chen et al., 2020). These approaches can be broadly categorized into attribute manipulation and face swap (Pei et al., 2024). Attribute manipulation models (Choi et al., 2018; Patashnik et al., 2021; Preechakul et al., 2022) aim to modify specific attributes (e.g., hairstyle, age, or facial expression) while preserving the underlying identity. These methods mainly leverage Generative Adversarial Networks (GANs) or Diffusion Models (DMs) to disentangle and traverse the latent representation space for precise or localized control of facial attributes without altering identity. In contrast, face swap models (Chen et al., 2020; Richardson et al., 2021; Shiohara et al., 2023) aim to replace the identity of a target face by injecting identity-related features from a source image while preserving pose, facial expression, and lighting conditions of the target. Earlier face swap approaches (Oldpan, 2018) typically used encoder-decoder structures, in which identity transfer can be achieved through manipulation of latent representations. Subsequent works have incorporated stronger generative priors such as GANs and DMs to improve synthesis fidelity, leading to more diverse methods of deepfake generation (Preechakul et al., 2022; Kim et al., 2022; Zhao et al., 2023; Kim et al., 2025). These continuous architectural evolutions have significantly narrowed the gap between real and generated facial imagery, achieving unprecedented levels of photorealism and identity consistency.

**Proactive Defense against Deepfake Generation.** Proactive defense aims to disrupt the deepfake generation process

*Table 1.* Categorization of evaluation metrics. Metrics are grouped by their measurement goals. Metrics adopted in our main benchmark are highlighted in **bold**, while auxiliary metrics evaluated in the Appendix are underlined.

| Measurement Goal | Metric | Proposed | Used in Prior Works |
|---|---|---|---|
| **Pixel-wise Fidelity** | **MSE** | - | (Ruiz et al., 2020), (Wang et al., 2022), (Qiao et al., 2024), (Qu et al., 2024), (Tang et al., 2024), (Jeong et al., 2025), (Shim & Yoon, 2025) |
| | PSNR | - | (Wang et al., 2022), (Aneja et al., 2022), (Van Le et al., 2023), (Qiao et al., 2024), (Qu et al., 2024), (Tang et al., 2024), (Jeong et al., 2025), (Lee et al., 2025), (Wang et al., 2025b) |
| | SR | (Ruiz et al., 2020) | (Ruiz et al., 2020), (Wang et al., 2022), (Qu et al., 2024) |
| | MAE | - | (Ruiz et al., 2020), (Qu et al., 2024) |
| | $SR_{mask}$ | (Huang et al., 2022) | (Huang et al., 2022), (Zhang et al., 2024), (Qiao et al., 2024),(Tang et al., 2024), (Qu et al., 2025) |
| **Perceptual Fidelity** | **LPIPS** | (Zhang et al., 2018) | (Aneja et al., 2022), (Van Le et al., 2023), (Qu et al., 2024), (Shim & Yoon, 2025), (Wang et al., 2025b), (Wang et al., 2025a) |
| | SSIM | (Wang et al., 2004) | (Wang et al., 2022), (Dong et al., 2023), (Qiao et al., 2024), (Qu et al., 2024), (Tang et al., 2024), (Lee et al., 2025), (Wang et al., 2025b) |
| | FSIM | (Zhang et al., 2011) | (Dong et al., 2023) |
| | VGG Loss | (Johnson et al., 2016) | (Aneja et al., 2022) |
| | FID | (Heusel et al., 2017) | (Huang et al., 2022), (Tang et al., 2024) |
| | CDSR | (Shim & Yoon, 2025) | (Shim & Yoon, 2025) |
| **Identity Disruption** | **CIDR** | ours | - |
| | ID Loss | (Deng et al., 2019) | (Van Le et al., 2023), (Qu et al., 2024), (Qu et al., 2025) , (Jeong et al., 2025), (Shim & Yoon, 2025), (Wang et al., 2025b), (Lee et al., 2025) |
| | $FN_{acc}$ | (Zhang et al., 2024) | (Zhang et al., 2024) |
| | FMR | (Wang et al., 2025a) | (Wang et al., 2025a) |
| **Visual Quality** | **BRISQUE** | (Mittal et al., 2012) | (Dong et al., 2023), (Van Le et al., 2023) |
| | FDFR | (Van Le et al., 2023) | (Van Le et al., 2023) |
| | TFHC | (zeusees, 2020) | (Huang et al., 2022), (Tang et al., 2024) |
| | SER-FQA | (Terhorst et al., 2020) | (Van Le et al., 2023) |

by preemptively modifying input images prior to their release to prevent downstream misuse. Early studies on deepfake proactive defense adopted pixel-level adversarial attacks designed to maximize distortion in the output space of deepfake generators with imperceptible perturbations (Ruiz et al., 2020; Aneja et al., 2022; Huang et al., 2022). Subsequent studies extended these manipulations beyond raw pixel space or Lab color space (Wang et al., 2022) to more robust feature representations by considering latent space disruption (Shim & Yoon, 2025) or identity disruption (Lee et al., 2025). In addition, identity disruption methods have also been explored under black-box assumptions, disrupting the synthesis process by perturbing the identity extraction process (Wang et al., 2025b) or obfuscating identity-related semantic embeddings (Wang et al., 2025a) rather than requiring access to the deepfake generator itself.

## 3. Analysis of Proactive Defense Evaluations

To demonstrate the necessity of a comprehensive benchmark, we formalize the evaluation objectives of proactive defenses and identify the systemic limitations inherent in current evaluation protocols.

### 3.1. Notation and Definitions

Let $\mathcal{X}$ denote the image space. Given a target deepfake generation model $G : \mathcal{X} \rightarrow \mathcal{X}$, the objective of proactive defense is to find a transformation $f$ that solves:

$$\max_{f} \mathcal{L}\big(G(\boldsymbol{x}), G(f(\boldsymbol{x}))\big) \quad \text{s.t. } d\big(\boldsymbol{x}, f(\boldsymbol{x})\big) \leq \epsilon, \quad (1)$$

where $\mathcal{L} : \mathcal{X} \times \mathcal{X} \rightarrow \mathbb{R}^+$ is a discrepancy function (e.g., MSE, LPIPS, or identity distance) measuring the degradation of the synthesis result, with higher values indicating stronger disruption of deepfake generation. $d$ denotes perceptual constraint (typically an $\ell_p$-norm) ensuring the protected image $f(\boldsymbol{x})$ is visually indistinguishable from the original $\boldsymbol{x}$. Thus, a proactive defense is defined as a transformation function $f_G : \mathcal{X} \rightarrow \mathcal{X}$ that maps an original image $\boldsymbol{x} \in \mathcal{X}$ to a protected counterpart $\boldsymbol{x}_{\text{adv}} = f_G(\boldsymbol{x})$. In the context of adversarial perturbations, $f_G$ typically adopts an additive form, $f_G(\boldsymbol{x}) = \boldsymbol{x} + \boldsymbol{\delta}$.

In this study, we consider two categories of deepfake generation models, which differ in inputs and synthesis objectives:

- *Attribute Manipulation* aims to modify specific semantic attribute condition $c$ (e.g., age, expression, or hairstyle) of the input image $\boldsymbol{x}$. The synthesis process, denoted as $G_{\text{attr}}(\boldsymbol{x}, c)$, requires the generator to perform the manipulation while preserving the original identity and image quality.
- *Face Swap* involves a source image $\boldsymbol{x}_s$ (providing identity) and a target image $\boldsymbol{x}_t$ (providing attributes, pose, and illumination). A face swap generator $G_{\text{swap}}(\boldsymbol{x}_s, \boldsymbol{x}_t)$ aims to transfer the source identity onto the target while preserving the target's structural and visual attributes.

### 3.2. Defense Objectives and Dimensions

A proactive defense aims to neutralize deepfake generation from the input image $\boldsymbol{x}$. Given a generation model $G_{\text{att}}(\boldsymbol{x}, c)$, the defense generates a protected image $\boldsymbol{x}_{\text{adv}} =$

$f_{G_{\text{att}}}(\boldsymbol{x})$ such that the resulting output $G_{\text{att}}(\boldsymbol{x}_{\text{adv}}, c)$ fails to execute the intended manipulation. In the context of a face swapping model $G_{\text{swap}}(\boldsymbol{x}_s, \boldsymbol{x}_t)$, the defense typically produces $\boldsymbol{x}_{\text{adv}} = f_G(\boldsymbol{x}_s)$ which induces generation failure, thereby protecting the source image $\boldsymbol{x}_s$.

The primary challenge in evaluating proactive defenses lies in the lack of standardized success metrics and the disparate assumptions across prior studies. To address this, we formalize evaluation goals by identifying specific synthesis failure modes aligned with the objectives defined in Section 3.1. The effectiveness of these defenses is measured by their ability to induce specific failure modes in deepfake models. We categorize these failures into three primary disruptions:

- *Synthesis Disruption*: A model fails to apply the attribute $c$ or maintain the structural property of the target image.
    - $G_{\text{att}}(\boldsymbol{x}_{\text{adv}}, c) \neq G_{\text{att}}(\boldsymbol{x}, c)$
    - $G_{\text{swap}}(\boldsymbol{x}_{\text{adv}}, \boldsymbol{x}_t) \neq G_{\text{swap}}(\boldsymbol{x}_s, \boldsymbol{x}_t)$
- *Identity Disruption*: The synthesized output fails to preserve the identity of the source image.
    - $id(\boldsymbol{x}) \neq id\big(G_{\text{att}}(\boldsymbol{x}_{\text{adv}}, c)\big)$
    - $id(\boldsymbol{x}_s) \neq id\big(G_{\text{swap}}(\boldsymbol{x}_{\text{adv}}, \boldsymbol{x}_t)\big)$
- *Visual Quality Degradation*: The defense induces noticeable artifacts, blurring, or unnatural distortions in the synthesized output.

For notational simplicity in the subsequent analysis, we denote both generative paradigms as $G$, omit explicit reference to the target attribute $c$ or image $\boldsymbol{x}_t$, and let $\boldsymbol{x}$ represent the original image (where $\boldsymbol{x} = \boldsymbol{x}_s$ for face swap).

We also characterize proactive defenses across four fundamental technical dimensions to provide a structured analysis of the defensive landscape:

- *Access Regime (White-box vs. Black-box)*: We consider white-box settings operating under full access to the generator $G$, and black-box settings where the defender has no direct knowledge of $G$ and relies on cross-model transferability or surrogate models.
- *Imperceptibility Constraints*: Imperceptibility $d(\boldsymbol{x}, \boldsymbol{x}_{\text{adv}})$ can be formalized using pixel-space $L_p$-norm constraints or perceptual metrics such as LPIPS.
- *Methodologies (Optimization vs. Model-based Learning)*: We distinguish between optimization-based approaches that compute an explicit perturbation $\boldsymbol{\delta} = \boldsymbol{x}_{\text{adv}} - \boldsymbol{x}$, and model-based approaches that learn a perturbation or reconstruction function $f_G$ to generate protected images.
- *Defense Objectives (disruption vs. identity)*: We differentiate between disruption methods aiming for global synthesis failure, and identity-related methods specifically targeting the obfuscation or invalidation of facial identity.

Beyond these core dimensions, additional properties such as ensemble-based strategies, universal perturbations, and robustness-aware training can further differentiate defense

methods. In Appendix B, Table 5 provides a concise overview of representative deepfake generation models and proactive defense methods from prior literature.

## 4. Our Benchmark

This section introduces our benchmark for the unified evaluation of proactive deepfake defenses. Addressing the limitations identified in Section 3, our benchmark aims to standardize evaluation across three key aspects: (i) diverse deepfake generators and proactive defense methods, (ii) multi-dimensional evaluation metrics that capture distinct synthesis failure behaviors, and (iii) consideration of practical deployment conditions. Figure 2 illustrates our evaluation pipeline. Each defense generates a protected version of an original image, both of which are processed by deepfake generators. We then quantify synthesis disruption, identity disruption, and visual quality by comparing the resulting outputs. To reflect practical deployment scenarios, we further evaluate robustness and transferability.

### 4.1. Evaluation Datasets and Algorithms

**Standard Dataset.** We adopt representative face datasets that are well-suited to support both attribute manipulation and face swap tasks. We utilize the CelebA-HQ (Karras et al., 2017) dataset as our primary benchmark due to its high quality and wide adoption in the field. To ensure consistent input dimensions across diverse generators and defense methods, all images are resized to a resolution of $256 \times 256$. Appendix B.1 summarizes datasets commonly used in prior proactive defense studies. To further validate the generalizability of our findings, we additionally conduct experiments on FFHQ and VGGFace2-HQ in Appendix B.3 and B.4, where consistent trends are observed across all metrics and generators.

**Deepfake Generators.** To evaluate the generation tasks categorized in Section 3.1, we select deepfake generation models: StarGAN (Choi et al., 2018), StyleCLIP (Patashnik et al., 2021), and DiffAE (Preechakul et al., 2022) for attribute manipulation, and SimSwap (Chen et al., 2020), pSp-mix (Richardson et al., 2021), BlendFace (Shiohara et al., 2023), DiffSwap (Zhao et al., 2023), and DiffFace (Kim et al., 2025) for face swap. These models have been widely used in prior proactive defense studies and provide publicly available implementations, facilitating reproducibility and fair comparison.

**Proactive Defenses.** Regarding proactive defenses, we leverage the taxonomy defined in Section 3.1 to select a diverse set of representative baselines that span various access regimes, methodological paradigms, and defense objectives. As summarized in Table 5(b), our benchmark

includes: PGD (Madry et al., 2017), Disrupting (Ruiz et al., 2020), Anti-Forgery (Wang et al., 2022), DF-RAP (Qu et al., 2024), Latent Attack (Kos et al., 2018), SCOL (Lee et al., 2025), and NullSwap (Wang et al., 2025b). We prioritize official implementations whenever available; for NullSwap, we reproduce the method following the algorithmic details reported in the original work.

## 4.2. Multi-Dimensional Evaluation Metrics

Designing evaluation metrics for proactive defense poses unique challenges, as effective defenses may induce qualitatively different failure behaviors depending on the target deepfake generation task and model architecture. Guided by the failure mode analysis in Section 3.2, we evaluate the *protectability* of proactive defenses along several complementary dimensions that collectively capture their effectiveness, specificity, and operational reliability. Table 1 categorizes evaluation metrics used in prior work according to their measurement domain and evaluation objective.

- *Synthesis Disruption* can be measured by the discrepancies between $G(\boldsymbol{x})$ and $G(\boldsymbol{x}_{\text{adv}})$ using:
  - *Pixel-level Fidelity* captures low-level reconstruction differences such as MSE, $\mathcal{L}_{\text{MSE}}(G(\boldsymbol{x}), G(\boldsymbol{x}_{\text{adv}}))$, and PSNR, $\mathcal{S}_{\text{PSNR}}(G(\boldsymbol{x}), G(\boldsymbol{x}_{\text{adv}}))$.
  - *Perceptual Fidelity* reflects perceptually meaningful deviations in the generated outputs such as SSIM, $\mathcal{S}_{\text{SSIM}}(G(\boldsymbol{x}), G(\boldsymbol{x}_{\text{adv}}))$, and LPIPS, $\mathcal{L}_{\text{LPIPS}}(G(\boldsymbol{x}), G(\boldsymbol{x}_{\text{adv}}))$.
- *Identity Disruption* can be evaluated by failures in preserving or transferring identity, measured using a pretrained face recognition model to compute ID Loss ($\mathcal{L}_{\text{ID}}$), face matching rate (FMR), or accuracy ($FN_{acc}$) (Deng et al., 2019; Wang et al., 2025a; Zhang et al., 2024). In this study, we introduce a calibrated identity ratio (CIDR, detailed below) to decouple the inherent domain shift induced by $G(\cdot)$.
- *Visual Quality Degradation* can be measured by no-reference quality and face-specific operational metrics on $G(\boldsymbol{x}_{\text{adv}})$, such as BRISQUE, face detection failure rate (FDFR), and true face with high confidence (TFHC).

We provide detailed explanations of all evaluation metrics in Appendix A. While our primary evaluation focuses on the disruption effectiveness, we also provide comparisons between $\boldsymbol{x}$ and $\boldsymbol{x}_{\text{adv}}$ in Appendix B.7 to verify the imperceptibility.

**Calibrated Identity Metric for Unbiased Evaluation.** In most studies, identity disruption is measured by ID Loss using $\mathcal{L}_{\text{ID}}(G(\boldsymbol{x}), G(\boldsymbol{x}_{\text{adv}}))$ (Van Le et al., 2023; Shim & Yoon, 2025; Wang et al., 2025b; Lee et al., 2025) or $\mathcal{L}_{\text{ID}}(\boldsymbol{x}, G(\boldsymbol{x}_{\text{adv}}))$. However, they often fail to isolate defensive efficacy because ID Loss is inherently biased by the

domain shift and artifacts introduced by the generator $G$ itself. To decouple the defense's impact from these generative artifacts, we introduce a calibrated identity ratio (CIDR):

$$\mathcal{R}_{\text{ID}}(\mathbf{x}, \mathbf{x}_{\text{adv}}) = 1 - \frac{\mathcal{L}_{\text{ID}}(\mathbf{x}, G(\mathbf{x}))}{\mathcal{L}_{\text{ID}}(\mathbf{x}, G(\mathbf{x}_{\text{adv}}))} \qquad (2)$$

By normalizing the distance between the source identity $\boldsymbol{x}$ and the protected output $G(\boldsymbol{x}_{\text{adv}})$ against the baseline synthesis error $\mathcal{L}_{\text{ID}}(\boldsymbol{x}, G(\boldsymbol{x}))$, CIDR[1] explicitly accounts for the generator's inherent identity distortion, providing a more objective and unbiased measure of identity obfuscation.

**Robustness and Transferability.** To evaluate the practical utility of proactive defenses, we consider two critical dimensions beyond basic settings.

- *Robustness to Post-processing Transformations*: Proactive protections often encounter lossy transformations such as JPEG compression and Gaussian blurring before reaching a generator. To quantify defensive resilience under such distortions, we define a relative robustness (ROB) for a given transformation $T$ and metric $\mathcal{L}$ as:

$$\mathcal{R}_{\text{ROB}}(\boldsymbol{x}, \boldsymbol{x}_{\text{adv}}; T, \mathcal{L}) = \frac{\mathcal{L}(\boldsymbol{x}_{\text{ref}}, G(T(\boldsymbol{x}_{\text{adv}})))}{\mathcal{L}(\boldsymbol{x}_{\text{ref}}, G(\boldsymbol{x}_{\text{adv}}))} \qquad (3)$$

where $\boldsymbol{x}_{\text{ref}}$ denotes the reference input (e.g., $\boldsymbol{x}$ or $G(\boldsymbol{x})$), or is omitted for no-reference quality measures. This ratio quantifies the relative degradation in defensive effectiveness induced by the transformation $T$. To obtain a holistic measure of robustness, we report an aggregated ROB[1] for each proactive method, computed by averaging $\mathcal{R}_{\text{ROB}}$ over all evaluated transformations $T$ and metrics $\mathcal{L}$.

- *Cross-generator Transferability*: We assess transferability by applying protected images optimized for a source generator to previously unseen target generators. This setting quantifies the extent to which a defense generalizes across different architectures and generative priors. Formally, given a source generator $G_{\text{s}}$ and a target generator $G_{\text{t}}$, a protected image $\boldsymbol{x}_{\text{adv}}^{\text{s}} = f_{G_{\text{s}}}(\boldsymbol{x})$ is evaluated on $G_{\text{t}}$. To isolate the transfer effect from the target model's baseline vulnerability, we define a transfer efficiency (TE) as:

$$\mathcal{R}_{\text{TE}}^{\text{s} \to \text{t}}(\boldsymbol{x}, \boldsymbol{x}_{\text{adv}}^{\text{s}}, \boldsymbol{x}_{\text{adv}}^{\text{t}}, \mathcal{L}) = \frac{\mathcal{L}(\boldsymbol{x}_{\text{ref}}, G_{\text{t}}(\boldsymbol{x}_{\text{adv}}^{\text{s}}))}{\mathcal{L}(\boldsymbol{x}_{\text{ref}}, G_{\text{t}}(\boldsymbol{x}_{\text{adv}}^{\text{t}}))} \qquad (4)$$

where $\boldsymbol{x}_{\text{ref}}$ follows the same convention as above, with $G$ replaced by $G_{\text{t}}$, and $\boldsymbol{x}_{\text{adv}}^{\text{t}} = f_{G_{\text{t}}}(\boldsymbol{x})$. This ratio captures the percentage of the maximum possible degradation achieved through transfer. We report the aggregated TE[1] by averaging these scores across all source-target pairs $(G_s, G_t)$ and evaluation metrics $\mathcal{L}$ to provide each defense's black-box effectiveness.

---

[1]For numerical stability and consistent aggregation, ROB and TE are clipped at an upper bound of 1, while CIDR is clipped at 0.

*Table 2.* Comprehensive evaluation of proactive defense methods across attribute manipulation and face swap tasks. We report pixel-wise fidelity, perceptual fidelity, identity disruption, and visual quality, along with Average Rank (AvgRank), robustness (ROB), and transferability (TE). Additional results are provided in Appendix B.2.

| Task | Generator | Metrics | Proactive Defenses | | | | | | |
|---|---|---|---|---|---|---|---|---|---|
| | | | White-box | | | | | Black-box | |
| | | | PGD | Disrupting | DF-RAP | Anti-Forgery | Latent Attack | SCOL | NullSwap |
| **Attribute Manipulation** | StarGAN | MSE (↑) | 1.3715 | 0.3003 | 0.2694 | **1.6180** | 0.0958 | 0.0590 | 0.0054 |
| | | LPIPS (↑) | 0.6878 | 0.4872 | 0.4477 | **0.6950** | 0.4368 | 0.2317 | 0.0340 |
| | | CIDR (↑) | 0.3764 | 0.1384 | 0.1002 | **0.4102** | 0.1712 | 0.2390 | 0.2792 |
| | | BRISQUE (↑) | 36.4792 | 25.0217 | 33.9631 | 29.6181 | **37.0453** | 21.8716 | 18.3381 |
| | | AvgRank (↓) | 2.00 | 4.25 | 4.50 | **1.75** | 4.00 | 5.50 | 6.00 |
| | | ROB (↑) | 0.7008 | 0.8495 | 0.8027 | 0.6253 | 0.7250 | **0.9883** | 0.9787 |
| | | TE (↑) | 0.4440 | 0.5755 | **0.5871** | 0.5457 | 0.4290 | - | - |
| | StyleCLIP | MSE (↑) | **5.4086** | 0.3308 | 0.3917 | 0.8045 | 0.2117 | 0.0481 | 0.0108 |
| | | LPIPS (↑) | **0.7042** | 0.4069 | 0.5030 | 0.4493 | 0.6439 | 0.3344 | 0.1046 |
| | | CIDR (↑) | 0.5344 | 0.3079 | 0.3287 | 0.2809 | **0.5852** | 0.3938 | 0.5676 |
| | | BRISQUE (↑) | **31.2795** | 21.7907 | 13.3239 | 21.0689 | 12.2354 | 6.5536 | 12.8330 |
| | | AvgRank (↓) | **1.50** | 4.25 | 3.75 | 4.00 | 3.50 | 5.75 | 5.25 |
| | | ROB (↑) | 0.7493 | 0.9506 | 0.9480 | 0.7350 | 0.8448 | **0.9921** | 0.9759 |
| | | TE (↑) | 0.5075 | **0.6795** | 0.6563 | 0.6157 | 0.5282 | - | - |
| | DiffAE | MSE (↑) | **0.1186** | 0.0178 | 0.0186 | 0.0020 | 0.0619 | 0.0546 | 0.0053 |
| | | LPIPS (↑) | 0.4070 | 0.1757 | 0.1301 | 0.0315 | **0.4598** | 0.1759 | 0.0353 |
| | | CIDR (↑) | 0.1183 | 0.0275 | 0.0215 | 0.0105 | 0.0117 | 0.3769 | **0.5028** |
| | | BRISQUE (↑) | **96.5489** | 19.5244 | 33.4323 | 12.5174 | 28.0547 | 38.4324 | 28.2317 |
| | | AvgRank (↓) | **1.75** | 4.75 | 4.25 | 7.00 | 3.50 | 2.50 | 4.25 |
| | | ROB (↑) | 0.5733 | 0.8821 | 0.8196 | **0.9710** | 0.7868 | 0.9283 | 0.9255 |
| | | TE (↑) | 0.4514 | 0.5018 | **0.5359** | 0.4262 | 0.4106 | - | - |
| **Face Swap** | SimSwap | MSE (↑) | **0.0440** | 0.0320 | 0.0238 | 0.0190 | 0.0123 | 0.0050 | 0.0060 |
| | | LPIPS (↑) | **0.0963** | 0.0801 | 0.0695 | 0.0577 | 0.0653 | 0.0315 | 0.0358 |
| | | CIDR (↑) | 0.4407 | 0.3820 | 0.3708 | 0.2935 | **0.6139** | 0.3519 | 0.5113 |
| | | BRISQUE (↑) | 1.7925 | **1.9301** | 1.4151 | 1.2399 | 0.8187 | 1.7359 | 1.0041 |
| | | AvgRank (↓) | **1.75** | 2.25 | 3.75 | 5.25 | 4.25 | 5.75 | 5.00 |
| | | ROB (↑) | 0.7269 | 0.9882 | 0.9244 | 0.7077 | 0.7764 | 0.9937 | **0.9981** |
| | | TE (↑) | 0.4686 | **0.6866** | 0.6464 | 0.5588 | 0.5166 | - | - |
| | pSp-mix | MSE (↑) | **0.5427** | 0.1270 | 0.2104 | 0.2496 | 0.0830 | 0.0463 | 0.0044 |
| | | LPIPS (↑) | **0.3331** | 0.1557 | 0.1984 | 0.2038 | 0.2118 | 0.1058 | 0.0254 |
| | | CIDR (↑) | 0.1936 | 0.1090 | 0.1172 | 0.0781 | 0.2069 | 0.1678 | **0.2856** |
| | | BRISQUE (↑) | **7.1814** | 6.7925 | 6.1890 | 6.2846 | 5.0783 | 5.9274 | 5.6176 |
| | | AvgRank (↓) | **1.50** | 4.25 | 4.00 | 3.75 | 4.00 | 5.25 | 5.25 |
| | | ROB (↑) | 0.8470 | 0.9558 | 0.9476 | 0.7831 | 0.8735 | 0.9981 | **0.9995** |
| | | TE (↑) | 0.4954 | **0.6582** | 0.6519 | 0.6049 | 0.5067 | - | - |
| | BlendFace | MSE (↑) | **0.0591** | 0.0233 | 0.0088 | 0.0383 | 0.0040 | 0.0013 | 0.0018 |
| | | LPIPS (↑) | **0.0552** | 0.0291 | 0.0164 | 0.0412 | 0.0151 | 0.0059 | 0.0080 |
| | | CIDR (↑) | 0.3848 | 0.2644 | 0.2077 | 0.3668 | **0.5364** | 0.2293 | 0.3982 |
| | | BRISQUE (↑) | 1.7485 | 2.1221 | 2.2879 | 2.1080 | 2.3089 | 2.4045 | **2.4165** |
| | | AvgRank (↓) | **3.00** | 4.00 | 4.75 | 3.50 | 3.50 | 5.50 | 3.75 |
| | | ROB (↑) | 0.7876 | 0.9205 | 0.9580 | 0.6122 | 0.8478 | **0.9992** | 0.9924 |
| | | TE (↑) | 0.4654 | **0.6362** | 0.6329 | 0.5323 | 0.4645 | - | - |
| | DiffSwap | MSE (↑) | 0.0046 | 0.0046 | 0.0047 | 0.0043 | **0.0097** | 0.0051 | 0.0053 |
| | | LPIPS (↑) | 0.0293 | 0.0283 | 0.0294 | 0.0309 | **0.0693** | 0.0309 | 0.0316 |
| | | CIDR (↑) | 0.0363 | 0.0425 | 0.0459 | 0.0430 | **0.2674** | 0.1139 | 0.2131 |
| | | BRISQUE (↑) | 4.0007 | 3.8500 | **4.0877** | 3.5432 | 3.5636 | 3.8448 | 3.7283 |
| | | AvgRank (↓) | 5.13 | 5.38 | 3.50 | 5.63 | **2.25** | 3.38 | 2.75 |
| | | ROB (↑) | 0.4438 | 0.4319 | **0.4594** | 0.4137 | 0.3498 | 0.4395 | 0.4157 |
| | | TE (↑) | 0.2895 | 0.3628 | **0.4956** | 0.4128 | 0.4375 | - | - |
| | DiffFace | MSE (↑) | 0.0069 | 0.0064 | 0.0068 | 0.0065 | **0.0142** | 0.0101 | 0.0103 |
| | | LPIPS (↑) | 0.0282 | 0.0269 | 0.0283 | 0.0281 | **0.0528** | 0.0395 | 0.0410 |
| | | CIDR (↑) | 0.0359 | 0.0385 | 0.0534 | 0.0361 | **0.6913** | 0.4120 | 0.5628 |
| | | BRISQUE (↑) | 1.8878 | 1.7522 | 1.7187 | 1.7217 | 1.3969 | **2.0021** | 1.8001 |
| | | AvgRank (↓) | 4.50 | 5.75 | 4.75 | 5.75 | 2.50 | 2.50 | **2.25** |
| | | ROB (↑) | 0.9968 | 0.9984 | 0.9923 | **1.000** | 0.8331 | 0.9807 | 0.9930 |
| | | TE (↑) | 0.3015 | 0.3884 | **0.4926** | 0.4316 | 0.4612 | - | - |

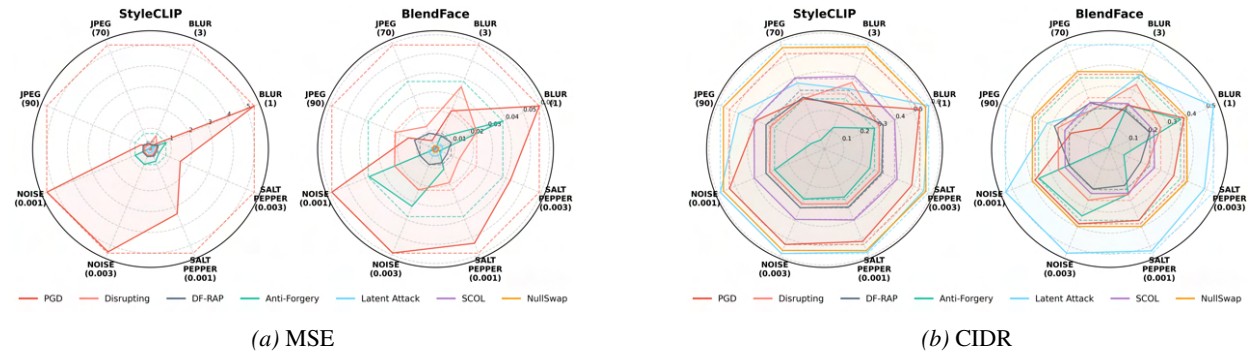

*(a)* MSE            *(b)* CIDR

*Figure 3.* Robustness against post-processing for StyleCLIP and BlendFace. The axes represent transformation types (JPEG compression, blur, Gaussian noise, and salt-and-pepper noise) with varying intensities. Subfigure (a) reports MSE, and subfigure (b) reports CIDR. Solid lines indicate performance under post-processing, while dashed lines denote reference performance without transformations. Complete visualizations for all generators are provided in Appendix C.2.

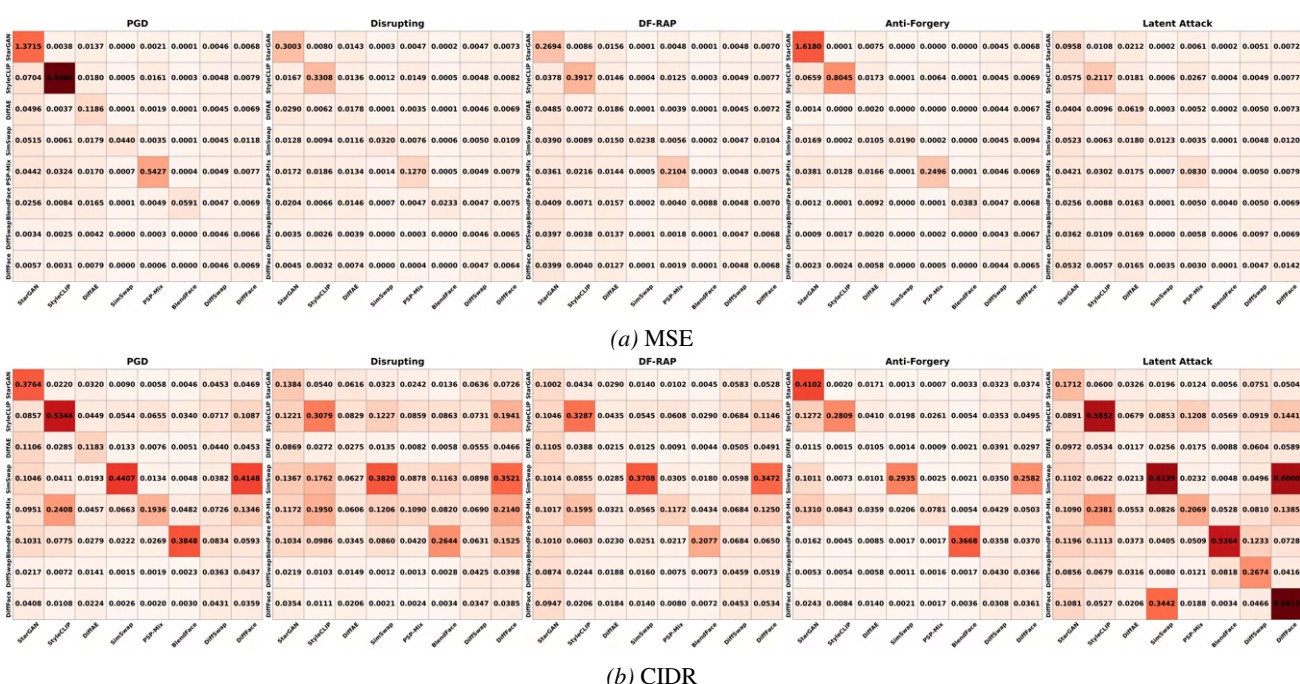

*(a)* MSE

*(b)* CIDR

*Figure 4.* Transferability of proactive defenses across deepfake generators. Rows denote source generators used to generate protected images, and columns denote target generators used for evaluation. Diagonal elements correspond to the white-box setting. Darker colors indicate better performance. (a) reports MSE, and (b) reports CIDR. All heat maps use a uniform color scale for fair comparison. Results for additional metrics are provided in Appendix D.

**Unified Performance Aggregation.** To assess the holistic effectiveness of proactive defenses, we aggregate performance across data samples and disparate metrics. At the population level, we report dataset-wide averages or ratio-based measures such as the proportion of samples exceeding predefined failure thresholds to ensure operational reliability under diverse inputs. Furthermore, to unify these multi-dimensional measurements into a single interpretable indicator, we employ Average Rank (AvgRank). This approach avoids the pitfalls of directly aggregating metrics with incompatible scales or directions by averaging a de-

fense's rank across all evaluation criteria. A lower AvgRank indicates that a defense method achieves consistently strong relative performance across multiple generators and metrics.

**Correlation of Evaluation Metrics.** To mitigate redundancy among the evaluation metrics, we conducted a pairwise correlation analysis across widely adopted metrics (detailed in Appendix E). Our results reveal that metrics sharing similar underlying objectives, such as pixel-wise fidelity (MSE, PSNR) or perceptual fidelity (SSIM, LPIPS), exhibit strong positive correlations. Conversely, metrics

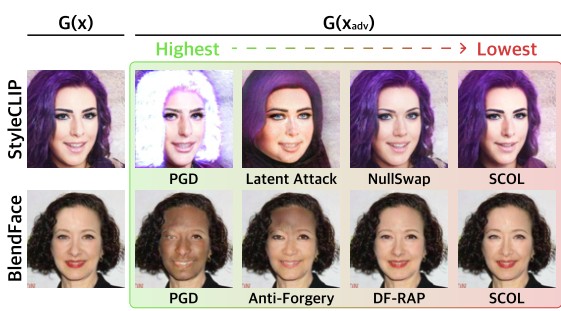

*Figure 5.* Qualitative comparison of selected proactive defenses based on the AvgRank. We visualize the most effective (Top-1, 2) and the least effective (Top-6, 7) methods for StyleCLIP and BlendFace, according to their AvgRank. The complete rankings of all evaluated methods are provided in Appendix B.6.

across distinct categories show negligible correlation, specifically visual quality and identity disruption. Based on these, we select one representative metric from each category in Table 1 to ensure a concise yet comprehensive evaluation.

## 5. Experiments and Analysis

**Defense performance relies on deepfake generator.** Table 2 summarizes the performance of proactive defenses across diverse deepfake generators. Critically, we observe that defense effectiveness varies significantly depending on the target generator. For instance, Anti-Forgery achieves strong disruption on StarGAN, yet it fails to induce meaningful disruption on DiffAE or SimSwap. This indicates that the effectiveness of Anti-Forgery is heavily conditioned on specific generative mechanisms. Conversely, while originally designed strictly for face swap tasks, SCOL demonstrates non-trivial disruption even when extended to unexpected tasks like attribute manipulation. This trend is further amplified in diffusion-based generators such as DiffSwap and DiffFace, where optimization-based methods (e.g., PGD, Disrupting, DF-RAP) show markedly reduced effectiveness despite achieving strong disruption on GAN-based models. This suggests that the iterative denoising process inherent to diffusion models attenuates pixel-level perturbations, making generator architecture a critical factor in defense evaluation. This variability highlights that evaluating defenses on a limited set of generators poses a significant risk of misrepresenting defense capabilities, underscoring the necessity of our comprehensive benchmark.

**Improving robustness inevitably degrades disruption.** As illustrated in Figure 3, PGD exhibits the strongest disruption in ideal settings but suffers sharp degradation under post-processing. In contrast, DF-RAP and Disrupting maintain stability against transformations by explicitly incorporating robustness constraints. However, this robustness comes at a cost: these methods exhibit lower visual disruption compared to PGD, confirming that improving

*Table 3.* Comparative evaluation of ensemble-based defenses against single-source baselines under white-box and black-box settings. Δ denotes the difference between the evaluation scores of ensemble-based defenses and their corresponding single-source baselines.

*(a)* CMUA vs. Single-source PGD

| Model | CMUA | | $\Delta_{PGD}^{DiffAE}$ | | $\Delta_{PGD}^{SimSwap}$ | |
|---|---|---|---|---|---|---|
| | MSE | CIDR | MSE | CIDR | MSE | CIDR |
| **White-box** | | | | | | |
| DiffAE | 0.0211 | 0.0399 | -0.0975 | -0.0784 | - | - |
| SimSwap | 0.0115 | 0.3363 | - | - | -0.0325 | -0.1044 |
| **Black-box** | | | | | | |
| StarGAN | 0.0562 | 0.1117 | +0.0066 | +0.0011 | +0.0047 | +0.0071 |
| StyleCLIP | 0.0122 | 0.0798 | +0.0085 | +0.0513 | +0.0061 | +0.0387 |
| pSp-mix | 0.0062 | 0.0166 | +0.0043 | +0.0009 | +0.0027 | +0.0032 |
| BlendFace | 0.0002 | 0.0073 | +0.0001 | +0.0022 | +0.0001 | +0.0025 |

*(b)* LEAT vs. Single-source Latent Attack

| Model | LEAT | | $\Delta_{Lat}^{DiffAE}$ | | $\Delta_{Lat}^{SimSwap}$ | |
|---|---|---|---|---|---|---|
| | MSE | CIDR | MSE | CIDR | MSE | CIDR |
| **White-box** | | | | | | |
| DiffAE | 0.0605 | 0.0145 | -0.0014 | +0.0028 | - | - |
| SimSwap | 0.0114 | 0.5846 | - | - | -0.0009 | -0.0293 |
| **Black-box** | | | | | | |
| StarGAN | 0.0562 | 0.1117 | +0.0009 | +0.0080 | -0.0110 | -0.0050 |
| StyleCLIP | 0.0122 | 0.0798 | +0.0003 | +0.0231 | +0.0036 | +0.0143 |
| pSp-mix | 0.0062 | 0.0166 | +0.0002 | +0.0105 | +0.0019 | +0.0048 |
| BlendFace | 0.0002 | 0.0103 | +0.0000 | +0.0015 | +0.0001 | +0.0055 |

robustness often compromises white-box performance.

**Severe performance drops emerge under unseen generators.** Analyzing transferability based on Figure 4 and Table 3, we observe that perturbations optimized for a single source generator exhibit limited transferability to unseen generators. In most cases, defenses that achieve strong disruption on the source generator suffer a sharp decline in effectiveness when transferred to different target generators. This indicates that high white-box scores often result from overfitting rather than genuine protection. In contrast, ensemble-based strategies (CMUA (Huang et al., 2022), LEAT (Shim & Yoon, 2025)) effectively mitigate this drop. Although they exhibit a slight degradation in white-box performance compared to PGD and Latent Attack, they demonstrate improved generalization across diverse architectures. Consequently, the marginal performance drop in ensemble methods is not a limitation but an inevitable trade-off required to ensure cross-model transferability.

**Single metrics overlook independent failure modes.** Our empirical results demonstrate that relying on a single metric category leads to conflicting interpretations of defense performance. We observe specific instances where fidelity and identity metrics diverge significantly, confirming that they capture orthogonal dimensions of effectiveness. For instance, NullSwap on DiffAE achieves significant identity disruption (high CIDR) while maintaining near-perfect

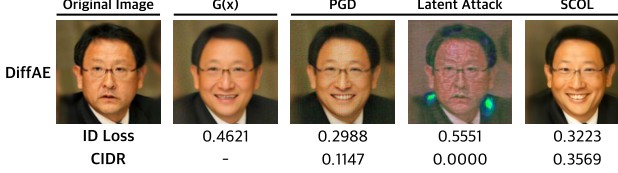

*Figure 6.* Comparison of identity metrics on DiffAE. In the ID Loss, the value for $G(x)$ denotes $L_{ID}(x, G(x))$, while values for defense methods denote $L_{ID}(G(x), G(x_{adv}))$. CIDR represents the calibrated identity ratio.

pixel fidelity (low MSE). This discrepancy highlights a critical blind spot: relying solely on fidelity metrics would falsely classify strategies with identity disruption objectives as ineffective. To validate the efficacy of unifying these diverse measurements, we examine the qualitative alignment of AvgRank. As visualized in Figure 5, methods with lower AvgRank consistently correspond to stronger qualitative disruption in the generated outputs, confirming that AvgRank effectively summarizes balanced performance across these diverse failure modes.

**CIDR corrects identity bias from generators.** Our analysis shows that standard ID Loss can provide a distorted view of defense effectiveness due to generator-induced identity bias. As shown in Figure 6, $\mathcal{L}_{ID}(x, G(x))$ is already biased, causing relative metrics to produce misleading rankings. For example, Latent Attack obtains a high ID score primarily due to background artifacts, despite largely preserving facial identity, while SCOL induces stronger identity disruption but is ranked lower by conventional metrics. Our proposed CIDR resolves this qualitative-quantitative mismatch by calibrating evaluations to the source identity. After calibration, ineffective methods are correctly suppressed, and effective defenses are re-ranked according to their true identity disruption capability.

**Diffusion-based architectures attenuate pixel-level perturbations.** We further investigate why optimization-based methods (e.g., PGD, Disrupting, DF-RAP), which maximize pixel-level perturbations via $L(G(x), G(x_{adv}))$, show significantly reduced effectiveness on diffusion-based models (e.g., DiffSwap, DiffFace). To probe this, we vary the number of sampling steps in DiffAE from T $\in \{10, 20, 50, 100, 200\}$ and measure the disruption effectiveness of PGD. As shown in Table 4, increasing the number of sampling steps leads to a consistent degradation in disruption metrics, indicating that the iterative denoising process progressively attenuates pixel-level perturbations. This explains why methods that directly optimize $L(G(x), G(x_{adv}))$ become less effective in diffusion-based architectures. On the other hand, black-box methods such as SCOL and NullSwap maintain disruption performance on these diffusion-based models, despite having no access to the target generator. Since diffusion-based models condition

their generation process on the source image, disrupting this conditioning stage effectively propagates disruption throughout the entire generation trajectory, regardless of the number of sampling steps.

*Table 4.* Disruption effectiveness of PGD on DiffAE under varying sampling steps

| Metric | T10 | T20 | T50 | T100 | T200 |
|---|---|---|---|---|---|
| MSE ↑ | 0.1200 | 0.0376 | 0.0224 | 0.0206 | 0.0193 |
| LPIPS ↑ | 0.4112 | 0.3988 | 0.3228 | 0.3024 | 0.2791 |
| CIDR ↑ | 0.1464 | 0.0630 | 0.0360 | 0.0493 | 0.0414 |
| BRISQUE ↑ | 96.9197 | 82.1362 | 65.8337 | 61.4653 | 59.4503 |

# 6. Conclusion

We address the critical fragmentation in proactive defense evaluation by establishing a unified benchmark that integrates disruption, robustness, and transferability. Beyond simply proposing a standardized protocol, our extensive empirical analysis offers a crucial reassessment of the current landscape: we confirm that fidelity and identity metrics operate on orthogonal axes, meaning that singular metrics inevitably yield biased conclusions regarding defense effectiveness. Furthermore, we identify a fundamental trade-off where improving robustness against transformations inherently compromises white-box disruption, suggesting that peak performance in isolated settings often signals overfitting rather than genuine protection. To preclude misleading interpretations caused by generator-induced biases, we demonstrate the necessity of the calibrated identity ratio (CIDR) as a fairer evaluation standard. We hope this unified protocol serves as a foundational guideline for future advancements in securing the digital media landscape.

# Acknowledgements

This work was supported by the Institute of Information & communications Technology Planning & Evaluation(IITP) grant funded by the Korea government(MSIT) (No. RS-2024-00398360, Development of a user-friendly and efficiency-optimized real-time homomorphic statistical analysis processing platform, 10%; No. RS-2020-II201336, Artificial Intelligence Graduate School Program(UNIST), 10%), the ITRC (Information Technology Research Center) grant funded by the Korea government (MSIT)(IITP-2026-RS-2024-00436936, 20%), the National Research Foundation of Korea(NRF) grant funded by the Korea government(MSIT)(RS-2025-00515481, 10%; RS-2026-25484948, 30%), and the "Advanced GPU Utilization Support Program" funded by the Government of the Republic of Korea(MSIT), (20%).

## Impact Statement

This paper establishes a comprehensive benchmark to rigorously assess the effectiveness of proactive defenses against deepfakes. We do not believe exposing the vulnerabilities of existing defenses will cause harm; rather, it serves as a critical reality check. Many current methods provide a false sense of security, working only in restricted settings while failing against real-world transformations or unknown generators. Since deepfake generation tools are already widely accessible to malicious actors, maintaining the illusion of protection is far more dangerous than revealing their limitations. In the future, as generative models continue to evolve, static defenses may become obsolete. We believe that establishing a standardized and unbiased evaluation protocol provides an essential foundation for the community. By shifting focus from overfitted results to genuine robustness, this work helps prepare society for future threats and guides the development of protection systems that actually work in the wild.

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

# A. Detailed Evaluation Metrics

We provide comprehensive descriptions of the evaluation metrics used to assess proactive defense methods against deepfake generation 1. While the main manuscript focuses on primary performance indicators, we detail a broader set of metrics for completeness. Most metrics quantify either the discrepancy or similarity between a deepfake output generated from a protected image and a reference baseline, considering two input images, $\boldsymbol{x}_1, \boldsymbol{x}_2 \in \mathcal{X}$. For notational simplicity, some metrics are formulated as loss functions $\mathcal{L} : \mathcal{X} \times \mathcal{X} \to \mathbb{R}$ or similarity functions $\mathcal{S} : \mathcal{X} \times \mathcal{X} \to \mathbb{R}$, where larger values indicate stronger or weaker disruption of deepfake generation, respectively.

Rather than being evaluated on individual image pairs, some metrics provide rate-based or distribution-based measures by using either paired datasets $D = \left\{ \left( \boldsymbol{x}_1^{(k)}, \boldsymbol{x}_2^{(k)} \right) \right\}_{k=1,\ldots,M}$ consisting of disrupted results and their corresponding reference images or two separate datasets (e.g., a set of disrupted results $D_1 = \left\{ \boldsymbol{x}_1^{(k)} \right\}_{k=1,\ldots,M_1}$ and a set of reference images $D_2 = \left\{ \boldsymbol{x}_2^{(k)} \right\}_{k=1,\ldots,M_2}$). Metrics that require additional information or alternative inputs are described separately below.

## A.1. Pixel-wise Fidelity

**Mean Absolute Error (MAE) and Mean Squared Error (MSE).** MAE and MSE distances quantify the pixel-wise discrepancy between two images. Higher values indicate larger visual distortions caused by the defense, and thus stronger defensive effectiveness.

$$\mathcal{L}_{\text{MAE}}(\boldsymbol{x}_1, \boldsymbol{x}_2) = \frac{1}{N} \|\boldsymbol{x}_1 - \boldsymbol{x}_2\|_1 \tag{5}$$

$$\mathcal{L}_{\text{MSE}}(\boldsymbol{x}_1, \boldsymbol{x}_2) = \frac{1}{N} \|\boldsymbol{x}_1 - \boldsymbol{x}_2\|_2^2 \tag{6}$$

where $N$ denotes the total number of pixels across all channels.

**Peak Signal-to-Noise Ratio (PSNR).** PSNR measures the reconstruction quality between two images at the pixel level. Lower values correspond to more severe degradation of generation quality, indicating stronger defensive impact.

$$\mathcal{S}_{\text{PSNR}}(\boldsymbol{x}_1, \boldsymbol{x}_2) = 10 \cdot \log_{10} \left( \frac{\max(\boldsymbol{x}_1)^2}{\mathcal{L}_{\text{MSE}}(\boldsymbol{x}_1, \boldsymbol{x}_2)} \right), \tag{7}$$

where $\max(\boldsymbol{x}_1)$ denotes the maximum allowable pixel intensity of the image.

**Success Rate (SR)** (Ruiz et al., 2020). SR measures the proportion of samples for which the defense successfully disrupts the generation process. A sample is considered successful if the $\mathcal{L}_{\text{MSE}}$ distance exceeds a predefined threshold. Higher values indicate that the defense consistently causes substantial deviations in generated outputs.

$$\text{SR}(D) = \frac{1}{M} \sum_{k=1}^{M} \mathbb{I}\left[ \mathcal{L}_{\text{MSE}}\left( \boldsymbol{x}_1^{(k)}, \boldsymbol{x}_2^{(k)} \right) \geq \tau \right], \tag{8}$$

$\tau = 0.05$ is used in (Ruiz et al., 2020).

**Masked Success Rate** ($SR_{mask}$) (Huang et al., 2022). $SR_{mask}$ evaluates defense effectiveness on localized facial regions by restricting the measurement to masked areas where attribute modifications occur. Higher values indicate that the defense consistently causes substantial deviations in generated outputs.

Let $\boldsymbol{x}_1, \boldsymbol{x}_2 \in \mathbb{R}^{H \times W \times C}$ denote the original and the disrupted images, respectively. We first define a binary spatial mask $\text{Mask}(\boldsymbol{x}_1, \boldsymbol{x}_2) \in \{0,1\}^{H \times W}$ that identifies the region of interest where pixel-wise deviations exceed a threshold $\tau_m$:

$$\text{Mask}_p(\boldsymbol{x}_1, \boldsymbol{x}_2) = \mathbb{I}\left( \|\boldsymbol{x}_1(p) - \boldsymbol{x}_2(p)\|_2 > \tau_m \right), \tag{9}$$

where $p \in \Omega$ denotes the pixel index and $\tau_m = 0.5$ is used in (Huang et al., 2022). The masked MSE distance is then calculated as:

$$\mathcal{L}_{\text{MSE}}^{\text{mask}}(\boldsymbol{x}_1, \boldsymbol{x}_2) = \frac{\sum_{p \in \Omega} \text{Mask}_p(\boldsymbol{x}_1, \boldsymbol{x}_2) \cdot \|\boldsymbol{x}_1(p) - \boldsymbol{x}_2(p)\|_2^2}{\|\text{Mask}(\boldsymbol{x}_1, \boldsymbol{x}_2)\|_1}, \tag{10}$$

where $\|\mathrm{Mask}(\boldsymbol{x}_1, \boldsymbol{x}_2)\|_1$ denotes the number of active pixels in the mask. Finally, the masked success rate is defined as

$$\mathrm{SR}_{\mathrm{mask}}(D) = \frac{1}{M} \sum_{k=1}^{M} \mathbb{I}\left[ \mathcal{L}_{\mathrm{MSE}}^{\mathrm{mask}}(\boldsymbol{x}_1^{(k)}, \boldsymbol{x}_2^{(k)}) > \tau \right] \tag{11}$$

where $M$ is the total number of evaluated image pairs and $\tau = 0.05$ is a predefined success threshold.

## A.2. Perceptual Fidelity

**Learned Perceptual Image Patch Similarity (LPIPS)** (Zhang et al., 2018). LPIPS computes the perceptual distance between two images in the feature space of a pretrained deep network, which correlates well with human visual perception. Higher values indicate larger perceptual differences and more effective disruption by the defense.

$$\mathcal{L}_{\mathrm{LPIPS}}(\boldsymbol{x}_1, \boldsymbol{x}_2) = \sum_{l} \frac{1}{H_l W_l} \sum_{h,w} \| w_l \odot (\Phi_l(\boldsymbol{x}_1) - \Phi_l(\boldsymbol{x}_2)) \|_2^2 \tag{12}$$

where $\Phi^l : \mathcal{X} \to \mathbb{R}^{C_l \times H_l \times W_l}$ denotes the feature map extracted from the $l$-th layer of a pretrained network, $H_l$ and $W_l$ are the spatial height and width of the feature map at layer $l$, and $w_l \in \mathbb{R}^{C_l}$ is a learned vector that scales each channel based on its perceptual relevance.

**Structural Similarity Index (SSIM)** (Wang et al., 2004). SSIM evaluates the perceptual similarity between two images by jointly comparing luminance, contrast, and structural information. Lower values indicate greater structural degradation in the generated images, implying a more effective defense.

$$\mathcal{S}_{\mathrm{SSIM}}(\boldsymbol{x}_1, \boldsymbol{x}_2) = \frac{(2\mu_{\boldsymbol{x}_1}\mu_{\boldsymbol{x}_2} + C_1)(2\sigma_{\boldsymbol{x}_1 \boldsymbol{x}_2} + C_2)}{(\mu_{\boldsymbol{x}_1}^2 + \mu_{\boldsymbol{x}_2}^2 + C_1)(\sigma_{\boldsymbol{x}_1}^2 + \sigma_{\boldsymbol{x}_2}^2 + C_2)}, \tag{13}$$

where $\mu_{\boldsymbol{x}_1}$ and $\mu_{\boldsymbol{x}_2}$ denote the local mean intensities of $\boldsymbol{x}_1$ and $\boldsymbol{x}_2$, respectively; $\sigma_{\boldsymbol{x}_1}^2$ and $\sigma_{\boldsymbol{x}_2}^2$ denote the corresponding variances; $\sigma_{\boldsymbol{x}_1 \boldsymbol{x}_2}$ denotes their covariance; and $C_1, C_2$ are small constants.

**Feature Similarity Index (FSIM)** (Zhang et al., 2011). FSIM evaluates image similarity by comparing perceptually important structural features between two images. Lower values indicate stronger disruption of essential structural information.

$$\mathcal{S}_{\mathrm{FSIM}}(\boldsymbol{x}_1, \boldsymbol{x}_2) = \frac{\sum_{(i,j) \in \Omega} S_L(\boldsymbol{x}_1, \boldsymbol{x}_2; i, j) \cdot PC_m(\boldsymbol{x}_1, \boldsymbol{x}_2; i, j)}{\sum_{(i,j) \in \Omega} PC_m(\boldsymbol{x}_1, \boldsymbol{x}_2; i, j)} \tag{14}$$

where $S_L(\boldsymbol{x}_1, \boldsymbol{x}_2; i, j)$ denotes the local structural similarity at pixel $(i, j)$ computed from phase congruency and gradient magnitude, $PC_m(\boldsymbol{x}_1, \boldsymbol{x}_2; i, j)$ represents the corresponding perceptual importance weight, $(i, j)$ index pixel locations, and $\Omega$ denotes the image domain.

**VGG Loss** (Johnson et al., 2016). VGG Loss computes the perceptual and semantic discrepancy between two images using feature representations extracted from a pretrained VGG-19 network. Higher VGG Loss values indicate stronger disruption of high-level semantic and structural information.

$$\mathcal{L}_{\mathrm{VGG}}(\boldsymbol{x}_1, \boldsymbol{x}_2) = \sum_{l} \lambda_l \| \Phi_l(\boldsymbol{x}_1) - \Phi_l(\boldsymbol{x}_2) \|_2^2 \tag{15}$$

where $\Phi^l(\boldsymbol{x})$ denotes the feature map from the $l$-th layer of the VGG-19 network and $\lambda_l$ is a hyperparameter that balances the contribution of each layer.

**Fréchet Inception Distance (FID)** (Heusel et al., 2017). FID measures the distributional distance between two sets of images by comparing the statistics of deep features extracted from a pretrained Inception v3 network. Higher values indicate lower perceptual quality and greater deviation between the two image distributions.

$$\text{FID}(D_1, D_2) = \|\mu_1 - \mu_2\|_2^2 + \text{Tr}\left(\Sigma_1 + \Sigma_2 - 2(\Sigma_1\Sigma_2)^{1/2}\right) \tag{16}$$

where $D_1$ and $D_2$ denote two sets of images; $(\mu_1, \Sigma_1)$ and $(\mu_2, \Sigma_2)$ are the mean and covariance of Inception v3 feature embeddings computed over each set; and $\text{Tr}(\cdot)$ denotes the trace operator.

**Comprehensive Defense Success Rate (CDSR)** (Shim & Yoon, 2025).    CDSR provides a holistic evaluation of defense effectiveness by jointly considering pixel-level distortion, perceptual degradation, and identity disruption. A sample is considered successfully defended if *at least one* of the following conditions is satisfied: (i) large pixel-level deviation, (ii) significant identity inconsistency, or (iii) strong perceptual dissimilarity. It is computed as a dataset-level percentage over multiple image pairs. Higher CDSR values indicate that the defense succeeds under at least one of the three complementary criteria, reflecting robust and comprehensive defensive performance.

$$\text{CDSR}(D) = \frac{1}{M}\sum_{k=1}^{M}\mathbb{I}\left[(\mathcal{L}_{\text{MSE}}(\boldsymbol{x}_1^{(k)}, \boldsymbol{x}_2^{(k)}) > \tau_1) \vee (\mathcal{L}_{\text{ID}}(\boldsymbol{x}_1^{(k)}, \boldsymbol{x}_2^{(k)}) > \tau_2) \vee (\mathcal{L}_{\text{LPIPS}}(\boldsymbol{x}_1^{(k)}, \boldsymbol{x}_2^{(k)}) > \tau_3)\right] \tag{17}$$

where $\tau_1 = 0.05$, $\tau_2 = 0.6$, and $\tau_3 = 0.35$ are used in (Shim & Yoon, 2025).

### A.3. Identity Disruption

**Identity Loss (ID Loss)** (Deng et al., 2019).    ID Loss measures identity dissimilarity between two images in an embedding space of ArcFace. Higher values indicate greater identity disruption.

$$\mathcal{L}_{\text{ID}}(\boldsymbol{x}_1, \boldsymbol{x}_2) = 1 - \frac{\Phi(\boldsymbol{x}_1)^\top \Phi(\boldsymbol{x}_2)}{\|\Phi(\boldsymbol{x}_1)\|_2 \|\Phi(\boldsymbol{x}_2)\|_2} \tag{18}$$

where $\Phi(\boldsymbol{x})$ denotes the identity embedding extracted by a pretrained face recognition model.

**Face Matching Rate (FMR)** (Wang et al., 2025a).    FMR measures the proportion of generated images $\boldsymbol{x}_1$ that are recognized as belonging to the same identity as their corresponding source images $\boldsymbol{x}_2$ by a face recognition model. Lower FMR values indicate stronger defense.

$$\text{FMR}(D) = \frac{1}{M}\sum_{k=1}^{M}\mathbb{I}\left[\|\Phi(\boldsymbol{x}_1^{(k)}) - \Phi(\boldsymbol{x}_2^{(k)})\| < \tau\right] \tag{19}$$

where $\Phi(\cdot)$ denotes the feature extraction by face recognition models and $\tau$ is a predefined matching threshold in the face recognition embedding space.

**FaceNet Recognition Accuracy Index** ($FN_{acc}$) (Zhang et al., 2024).    $FN_{acc}$ measures how effectively a defense prevents FaceNet-based identity recognition systems from classifying face-swapped outputs as the source identity. Lower values indicate stronger defense, as the protected deepfakes are less frequently recognized as belonging to the intended source person. The success rate on identity recognition ($\text{SR}_{IR}$) is defined as an auxiliary measure, using a classifier $c_{FN} : \mathcal{X} \mapsto \mathcal{Y}$ that predicts identity labels from facial embeddings of FaceNet for image $\boldsymbol{x} \in \mathcal{X}$ (Schroff et al., 2015):

$$\text{SR}_{IR}(D) = \frac{1}{M}\sum_{k=1}^{M}\mathbb{I}\left[\mathcal{C}_{FN}(\boldsymbol{x}_{\text{adv}}^{(k)}) = id(\boldsymbol{x}_{\text{src}}^{(k)})\right] \tag{20}$$

where $D = \{(\boldsymbol{x}_{\text{adv}}^{(k)}, \boldsymbol{x}_{\text{src}}^{(k)})\}$ for the defended images $\boldsymbol{x}_{\text{adv}}^{(k)}$ and the source images $\boldsymbol{x}_{\text{src}}^{(k)}$ for faceswap models $id(\cdot)$ is the identity of the source image $\boldsymbol{x}_2^{(k)}$. $FN_{acc}$ consists of the ratio between $\text{SR}_{IR}$, measuring how often defended face-swapped images in $D$ are recognized as the source identity, compared with the baseline recognition rate for undefended face-swapped images in $D_{\text{undef}} = \{(\boldsymbol{x}_{\text{undef}}^{(k)}, \boldsymbol{x}_{\text{src}}^{(k)})\}$.

$$FN_{acc}(D) = \frac{\text{SR}_{IR}(D)}{\text{SR}_{IR}(D_{\text{undef}})} \tag{21}$$

## A.4. Visual Quality

**BRISQUE** (Mittal et al., 2012). BRISQUE is a no-reference image quality assessment metric that evaluates the naturalness and artifact level of generated images without requiring a reference. Higher scores indicate stronger visual artifacts and lower image quality, reflecting a more disruptive defense.

**Face Detection Failure Rate (FDFR)** (Van Le et al., 2023). FDFR quantifies the proportion of generated images in which no face is detected by a face detection model. Higher values indicate that the defense disrupts facial structure to the extent that generated images are no longer recognized as valid faces.

**True Face with High Confidence (TFHC)** (zeusees, 2020). TFHC evaluates the proportion of generated faces that are classified as genuine with high confidence by a liveness detection system. Lower values indicate that fewer generated images are recognized as real faces with high confidence, implying that the defense effectively hinders their classification as authentic.

**SER-FQA** (Terhorst et al., 2020). SER-FQA is a face-specific image quality assessment metric designed to capture subtle degradations in facial structure and perceptual quality. Lower scores indicate significant degradation of facial quality caused by the defense.

## B. Deepfake Generation Models and Proactive Methods

*Table 5.* Overview of deepfake generation models and proactive defense methods.(a) summarizes representative deepfake generation models. (b) overviews proactive defense methods in terms of access regime, methodology, imperceptibility, and publication metadata. (c) summarizes their evaluation coverage across pixel-wise fidelity, perceptual fidelity, identity disruption, visual quality, robustness, and transferability.

*(a)* Deepfake generation models

| Task | Model | Backbone | Venue | Year / Citation | Code |
|---|---|---|---|---|---|
| **Attribute Manipulation** | StarGAN (Choi et al., 2018) | GAN | CVPR | 2018 / 5357 | ✓ |
| | AGGAN (Tang et al., 2019) | GAN | IJCNN | 2019 / 192 | ✓ |
| | StyleCLIP (Patashnik et al., 2021) | GAN | ICCV | 2021 / 1574 | ✓ |
| | DiffAE (Preechakul et al., 2022) | Diffusion | CVPR | 2022 / 617 | ✓ |
| | DiffusionCLIP (Kim et al., 2022) | Diffusion | CVPR | 2022 / 934 | ✓ |
| **Face Swap** | Faceswap-pytorch (Oldpan, 2018) | Auto Encoder | - | - | ✓ |
| | SimSwap (Chen et al., 2020) | Auto Encoder | ACM MM | 2020 / 490 | ✓ |
| | pSp-mix (Richardson et al., 2021) | GAN | CVPR | 2021 / 1555 | ✓ |
| | BlendFace (Shiohara et al., 2023) | Auto Encoder | ICCV | 2023 / 91 | ✓ |
| | DiffSwap (Zhao et al., 2023) | Diffusion | CVPR | 2023 / 123 | ✓ |
| | DiffFace (Kim et al., 2025) | Diffusion | PR | 2025 / 112 | ✓ |

*(b)* Proactive defense methods: design and access assumptions

| Objective | Method | Access regime | Methodology | Imperceptibility | Venue | Year / Citation | Code |
|---|---|---|---|---|---|---|---|
| **Disruption** | Disrupting (Ruiz et al., 2020) | White-box | Optimization | $L_p$-norm | ECCV | 2020 / 187 | ✓ |
| | Anti-Forgery (Wang et al., 2022) | White-box | Optimization | $L_p$-norm | IJCAI | 2022 / 89 | ✓ |
| | CMUA (Huang et al., 2022) | White-box | Optimization | $L_p$-norm | AAAI | 2022 / 142 | ✓ |
| | TCA-GAN (Dong et al., 2023) | Black-box | Model-based Learning | $L_p$-norm | IEEE TIFS | 2023 / 95 | - |
| | ID-Guard (Qu et al., 2025) | White-box | Model-based Learning | $L_p$-norm | TPAMI | 2024 / 2 | - |
| | SUA (Qiao et al., 2024) | White-box | Model-based Learning | $L_p$-norm | IEEE TIFS | 2024 / 16 | - |
| | FOUND (Tang et al., 2024) | White-box | Model-based Learning | $L_p$-norm | ACM TOMM | 2024 / 18 | - |
| | Dual Defense (Zhang et al., 2024) | White-box | Model-based Learning | $L_p$-norm, SSIM | IEEE TIFS | 2024 / 38 | ✓ |
| | DF-RAP (Qu et al., 2024) | White-box | Optimization | $L_p$-norm | IEEE TIFS | 2024 / 35 | ✓ |
| | LEAT (Shim & Yoon, 2025) | White-box | Optimization | $L_p$-norm | ESWA | 2025 / 4 | - |
| | Faceshield (Jeong et al., 2025) | White-box | Optimization | $L_p$-norm | ICCV | 2025 / 2 | ✓ |
| **Identity** | SCOL (Lee et al., 2025) | Black-box | Model-based Learning | $L_p$-norm, LPIPS | ACM MM | 2025 / 0 | ✓ |
| | NullSwap (Wang et al., 2025b) | Black-box | Model-based Learning | $L_p$-norm, LPIPS | ICCV | 2025 / 6 | - |
| | FaceSwapGuard (Wang et al., 2025a) | Black-box | Optimization | $L_p$-norm | - | 2025 / 2 | - |

*(c)* Evaluation coverage

| Method | Pixel-wise Fidelity | Perceptual Fidelity | Visual Quality | Identity Disruption | Robustness | Transferability |
|---|---|---|---|---|---|---|
| Disrupting (Ruiz et al., 2020) | ✓ | - | - | - | ✓ | - |
| Anti-Forgery (Wang et al., 2022) | ✓ | ✓ | - | - | ✓ | ✓ |
| CMUA (Huang et al., 2022) | ✓ | - | - | - | - | ✓ |
| TCA-GAN (Dong et al., 2023) | - | ✓ | ✓ | - | - | ✓ |
| ID-Guard (Qu et al., 2025) | ✓ | - | - | ✓ | ✓ | ✓ |
| SUA (Qiao et al., 2024) | ✓ | ✓ | - | - | - | ✓ |
| FOUND (Tang et al., 2024) | ✓ | - | - | - | - | ✓ |
| Dual Defense (Zhang et al., 2024) | ✓ | ✓ | - | - | ✓ | - |
| DF-RAP (Qu et al., 2024) | ✓ | ✓ | - | - | ✓ | ✓ |
| LEAT (Shim & Yoon, 2025) | ✓ | ✓ | - | ✓ | - | ✓ |
| Faceshield (Jeong et al., 2025) | ✓ | - | - | ✓ | ✓ | ✓ |
| SCOL (Lee et al., 2025) | ✓ | ✓ | - | ✓ | - | ✓ |
| NullSwap (Wang et al., 2025b) | ✓ | ✓ | - | ✓ | - | ✓ |
| FaceSwapGuard (Wang et al., 2025a) | - | ✓ | - | ✓ | ✓ | ✓ |
| **Our Benchmark** | ✓ | ✓ | ✓ | ✓ | ✓ | ✓ |

### B.1. Details of Proactive Defenses

In this section, we provide the detailed mathematical formulations and algorithmic procedures for the proactive defense methods used in our experiment.

**Optimization-based Approaches.** Optimization-based defenses explicitly solve the maximization problem defined in Eq. 1 by directly optimizing an additive perturbation $\delta$. To encompass both standard and robustness-aware scenarios, we

adopt a general formulation incorporating a transformation function $T(\cdot)$:

$$\delta^{\star} = \arg\max_{\delta} \ \mathcal{L}(G(\boldsymbol{x}), G(T(\boldsymbol{x} + \delta))), \quad \text{s.t.} \|\delta\| \leq \epsilon \tag{22}$$

Here, $T(\cdot)$ represents a post-processing transformation applied to the protected image. In the standard setting (PGD), $T$ is simply the identity function.

However, to enhance robustness, specific methods define $T$ as a differentiable approximation of image transformations:

- **Disrupting Deepfakes** utilizes operations such as Gaussian blur within $T$, ensuring the defense remains effective even after low-pass filtering.
- **DF-RAP** incorporates a differentiable JPEG approximation into $T$ to learn perturbations robust to compression.
- **Anti-Forgery** transforms the optimization domain from RGB to the LAB color space. It explicitly optimizes perturbations within the chromatic channels ($a$ and $b$) while preserving the luminance component ($L$).
- **Latent Attack** diverges from the pixel-level objective. It replaces the generator $G$ in the formulation with the encoder $E$, maximizing discrepancy within the latent feature space instead of the final output space.

**Model-based Learning Approaches.** Unlike optimization-based methods that iteratively compute perturbations for each input instance, model-based defenses train a parameterized reconstruction network $r_\theta : \mathcal{X} \to \mathcal{X}$ to instantaneously generate protected images $\boldsymbol{x}_{\text{adv}} = r_{\theta^\star}(\boldsymbol{x})$.

$$\theta^{\star} = \arg\min_{\theta} \ \lambda_{\text{recon}} L_{\text{recon}}(r_\theta(\boldsymbol{x}), \boldsymbol{x}) - \lambda_{\text{ID}} \mathcal{L}_{\text{ID}}(r_\theta(\boldsymbol{x}), \boldsymbol{x}) \tag{23}$$

Here, $\mathcal{L}_{\text{recon}}$ combines MSE and LPIPS to ensure the protected image remains visual fidelity to the original.

- **SCOL** induces manifold collapse by manipulating the generator's latent representations. Instead of relying on external recognition models, it employs an identity inversion strategy within the latent space. Consequently, the output fails to synthesize the intended source identity, effectively neutralizing the manipulation.
- **NullSwap** targets the identity embedding space under a black-box assumption. It employs adaptive loss weighting to shift the source identity vector toward a null or invalid state. This prevents the generator from extracting consistent identity features, resulting in an output where the identity is unidentifiable or collapsed.

**Datasets Used in Prior Proactive Defense Studies.** Most prior proactive defense methods are evaluated on a limited set of face datasets, with CelebA-HQ being the most commonly adopted benchmark. For fair comparison with existing works, we also conduct our evaluations on CelebA-HQ.

*Table 6.* Datasets adopted in prior proactive defense studies.

| Dataset | Used |
|---|---|
| CelebA | (Ruiz et al., 2020), (Wang et al., 2022), (Huang et al., 2022), (Qiao et al., 2024), (Tang et al., 2024), (Qu et al., 2024) |
| CelebA-HQ | (Aneja et al., 2022), (Van Le et al., 2023), (Shim & Yoon, 2025), (Lee et al., 2025), (Wang et al., 2025a), (Jeong et al., 2025), (Wang et al., 2025b) |
| CelebAMask-HQ | (Qu et al., 2025) |
| LFW | (Huang et al., 2022), (Qiao et al., 2024), (Qu et al., 2024), (Wang et al., 2025b), (Qu et al., 2025) |
| VGGFace2 | (Van Le et al., 2023), (Zhang et al., 2024) |
| VGGFace2-HQ | (Aneja et al., 2022), (Jeong et al., 2025), (Lee et al., 2025) |
| CASIA-WebFace | (Zhang et al., 2024) |
| FFHQ | (Aneja et al., 2022), (Qu et al., 2024), (Qu et al., 2025) |
| Film100 | (Huang et al., 2022) |
| FaceScrub | (Dong et al., 2023) |

## B.2. Additional Quantitative Results of Table 2

This section reports additional quantitative results that complement Table 2 in the main paper. While Table 2 presents a subset of representative metrics for concise comparison, this appendix provides further evaluation results across multiple generators, defenses, and commonly used metrics.

*Table 7.* Additional quantitative results for proactive defense methods across multiple deepfake generators and evaluation metrics.

| Task | Generator | Metrics | Proactive Defenses | | | | | | |
| --- | --- | --- | --- | --- | --- | --- | --- | --- | --- |
| | | | White-box | | | | | Black-box | |
| | | | PGD | Disrupting | DF-RAP | Anti-Forgery | Latent Attack | SCOL | NullSwap |
| Attribute Manipulation | StarGAN | MSE (↑) | 1.3715 | 0.3003 | 0.2694 | **1.6180** | 0.0958 | 0.0590 | 0.0054 |
| | | PSNR (↓) | 4.7026 | 11.6732 | 12.2191 | **4.0173** | 16.5134 | 18.6093 | 29.2641 |
| | | SSIM (↓) | 0.1283 | 0.4556 | 0.4833 | **0.1156** | 0.5125 | 0.5459 | 0.9410 |
| | | LPIPS (↑) | 0.6878 | 0.4872 | 0.4477 | **0.6950** | 0.4368 | 0.2317 | 0.0340 |
| | | CIDR (↑) | 0.3764 | 0.1384 | 0.1002 | **0.4102** | 0.1712 | 0.2390 | 0.2792 |
| | | ID Loss (↑) | 0.8912 | 0.7031 | 0.6700 | **0.8923** | 0.5647 | 0.2878 | 0.2684 |
| | | BRI (↑) | 36.4792 | 25.0217 | 33.9631 | 29.6181 | **37.0453** | 21.8716 | 18.3381 |
| | | SR (↑) | **1.00** | **1.00** | **1.00** | **1.00** | 0.94 | 0.60 | 0.00 |
| | | FDFR (↑) | **0.57** | 0.03 | 0.01 | 0.55 | 0.02 | 0.00 | 0.00 |
| | StyleCLIP | MSE (↑) | **5.4086** | 0.3308 | 0.3917 | 0.8045 | 0.2117 | 0.0481 | 0.0108 |
| | | PSNR (↓) | **6.9544** | 13.2088 | 11.1592 | 10.5119 | 14.7425 | 19.5295 | 25.9558 |
| | | SSIM (↓) | **0.3185** | 0.6071 | 0.5106 | 0.5483 | 0.4450 | 0.7050 | 0.8885 |
| | | LPIPS (↑) | **0.7042** | 0.4069 | 0.5030 | 0.4493 | 0.6439 | 0.3344 | 0.1046 |
| | | CIDR (↑) | 0.5344 | 0.3079 | 0.3287 | 0.2809 | **0.5852** | 0.3938 | 0.5676 |
| | | ID Loss (↑) | 0.6428 | 0.2900 | 0.3158 | 0.2787 | **0.7477** | 0.2379 | 0.5358 |
| | | BRI (↑) | **31.2795** | 21.7907 | 13.3239 | 21.0689 | 12.2354 | 6.5536 | 12.833 |
| | | SR (↑) | **1.00** | **1.00** | **1.00** | **1.00** | 0.99 | 0.40 | 0.00 |
| | | FDFR (↑) | **0.09** | 0.00 | 0.00 | 0.00 | 0.08 | 0.00 | 0.00 |
| | DiffAE | MSE (↑) | **0.1186** | 0.0178 | 0.0186 | 0.0020 | 0.0619 | 0.0546 | 0.0053 |
| | | PSNR (↓) | **15.3987** | 23.9081 | 23.5901 | 36.5044 | 18.2288 | 19.1150 | 29.1722 |
| | | SSIM (↓) | **0.0288** | 0.5364 | 0.4468 | 0.9393 | 0.4993 | 0.6475 | 0.9235 |
| | | LPIPS (↑) | 0.4070 | 0.1757 | 0.1301 | 0.0315 | **0.4598** | 0.1759 | 0.0353 |
| | | CIDR (↑) | 0.1183 | 0.0275 | 0.0215 | 0.0105 | 0.0117 | 0.3769 | 0.5028 |
| | | ID Loss (↑) | 0.2842 | 0.0915 | 0.0750 | 0.0078 | **0.5059** | 0.3269 | 0.4564 |
| | | BRI (↑) | 96.5489 | 19.5244 | 33.4323 | 12.5174 | 28.0547 | 38.4324 | 28.2317 |
| | | SR (↑) | **0.99** | 0.02 | 0.02 | 0.00 | 0.78 | 0.49 | 0.00 |
| | | FDFR (↑) | 0.00 | 0.00 | 0.00 | 0.00 | 0.00 | 0.00 | 0.00 |
| Face Swap | SimSwap | MSE (↑) | **0.0440** | 0.0320 | 0.0238 | 0.0190 | 0.0123 | 0.0050 | 0.0060 |
| | | PSNR (↓) | **19.9665** | 21.4425 | 22.6389 | 23.7911 | 25.3029 | 29.2693 | 28.3813 |
| | | SSIM (↓) | **0.7870** | 0.8168 | 0.8323 | 0.8594 | 0.8256 | 0.9158 | 0.9016 |
| | | LPIPS (↑) | **0.0963** | 0.0801 | 0.0695 | 0.0577 | 0.0653 | 0.0315 | 0.0358 |
| | | CIDR (↑) | 0.4407 | 0.3820 | 0.3708 | 0.2935 | **0.6139** | 0.3519 | 0.5113 |
| | | ID Loss (↑) | 0.5753 | 0.4897 | 0.4377 | 0.3338 | **1.0160** | 0.3487 | 0.5737 |
| | | BRI (↑) | 1.7925 | **1.9301** | 1.4151 | 1.2399 | 0.8187 | 1.7359 | 1.0041 |
| | | SR (↑) | **0.33** | 0.10 | 0.03 | 0.02 | 0.00 | 0.00 | 0.00 |
| | | FDFR (↑) | 0.00 | 0.00 | 0.00 | 0.00 | 0.00 | 0.00 | 0.00 |
| | pSp-mix | MSE (↑) | **0.5427** | 0.1270 | 0.2104 | 0.2496 | 0.0830 | 0.0463 | 0.0044 |
| | | PSNR (↓) | **8.8960** | 16.2547 | 13.2501 | 12.5052 | 17.5581 | 19.7260 | 29.7747 |
| | | SSIM (↓) | **0.4774** | 0.7232 | 0.6532 | 0.6500 | 0.6378 | 0.7486 | 0.9411 |
| | | LPIPS (↑) | **0.3331** | 0.1557 | 0.1984 | 0.2038 | 0.2118 | 0.1058 | 0.0254 |
| | | CIDR (↑) | 0.1936 | 0.1090 | 0.1172 | 0.0781 | 0.2069 | 0.1678 | **0.2856** |
| | | ID Loss (↑) | **0.3643** | 0.1595 | 0.1786 | 0.1443 | 0.3434 | 0.1574 | 0.2376 |
| | | BRI (↑) | **7.1814** | 6.7925 | 6.1890 | 6.2846 | 5.0783 | 5.9274 | 5.6176 |
| | | SR (↑) | **1.00** | 0.77 | 0.99 | **1.00** | 0.66 | 0.37 | 0.00 |
| | | FDFR (↑) | 0.00 | 0.00 | 0.00 | 0.00 | 0.00 | 0.00 | 0.00 |
| | BlendFace | MSE (↑) | **0.0591** | 0.0233 | 0.0088 | 0.0383 | 0.0040 | 0.0013 | 0.0018 |
| | | PSNR (↓) | **19.2648** | 23.8034 | 27.4301 | 21.7302 | 30.1689 | 35.0762 | 33.6978 |
| | | SSIM (↓) | **0.8994** | 0.9422 | 0.9647 | 0.9227 | 0.9525 | 0.9816 | 0.9764 |
| | | LPIPS (↑) | **0.0552** | 0.0291 | 0.0164 | 0.0412 | 0.0151 | 0.0059 | 0.0080 |
| | | CIDR (↑) | 0.3848 | 0.2644 | 0.2077 | 0.3668 | **0.5364** | 0.2293 | 0.3982 |
| | | ID Loss (↑) | 0.6401 | 0.4343 | 0.3252 | 0.5582 | **0.8814** | 0.2779 | 0.4834 |
| | | BRI (↑) | 1.7485 | 2.1221 | 2.2879 | 2.1080 | 2.3089 | 2.4045 | **2.4165** |
| | | SR (↑) | **0.54** | 0.13 | 0.00 | 0.21 | 0.00 | 0.00 | 0.00 |
| | | FDFR (↑) | 0.00 | 0.00 | 0.00 | 0.00 | 0.00 | 0.00 | 0.00 |

## B.3. Additional Quantitative Results using FFHQ

*Table 8.* Additional results on FFHQ using the same evaluation setup as Table 2.

| Task | Generator | Metrics | Proactive Defenses | | | | | | |
|---|---|---|---|---|---|---|---|---|---|
| | | | White-box | | | | | Black-box | |
| | | | PGD | Disrupting | DF-RAP | Anti-Forgery | Latent Attack | SCOL | NullSwap |
| **Attribute Manipulation** | StarGAN | MSE (↑) | 1.2800 | 0.2417 | 0.2499 | **1.4402** | 0.0817 | 0.0653 | 0.0053 |
| | | LPIPS (↑) | 0.6847 | 0.4515 | 0.4058 | **0.6915** | 0.3832 | 0.2528 | 0.0387 |
| | | CIDR (↑) | 0.3498 | 0.1107 | 0.0766 | **0.3768** | 0.1360 | 0.2209 | 0.2328 |
| | | BRISQUE (↑) | 34.6629 | 24.9974 | 33.3575 | 28.6003 | **35.8447** | 21.9457 | 18.7795 |
| | | AvgRank (↓) | 2.00 | 4.50 | 4.25 | **1.75** | 4.00 | 5.50 | 6.00 |
| | | ROB (↑) | 0.7070 | 0.8612 | 0.8116 | 0.6388 | 0.7406 | 0.9875 | **0.9987** |
| | | TE (↑) | 0.3120 | 0.4456 | **0.4585** | 0.3373 | 0.2929 | - | - |
| | StyleCLIP | MSE (↑) | **4.1773** | 0.2394 | 0.3327 | 0.5960 | 0.1806 | 0.0540 | 0.0107 |
| | | LPIPS (↑) | **0.6647** | 0.3772 | 0.4744 | 0.4515 | 0.6248 | 0.3198 | 0.1048 |
| | | CIDR (↑) | 0.5644 | 0.3450 | 0.3470 | 0.3143 | **0.6227** | 0.4226 | 0.5611 |
| | | BRISQUE (↑) | **28.0250** | 18.5784 | 12.0067 | 18.3265 | 12.7946 | 7.0981 | 11.3217 |
| | | AvgRank (↓) | **1.25** | 4.25 | 4.25 | 4.00 | 3.00 | 5.75 | 5.50 |
| | | ROB (↑) | 0.7691 | 0.9749 | 0.9528 | 0.7534 | 0.8542 | **0.9950** | 0.9846 |
| | | TE (↑) | 0.3954 | **0.5563** | 0.5249 | 0.4460 | 0.4100 | - | - |
| | DiffAE | MSE (↑) | **0.1016** | 0.0134 | 0.0158 | 0.0274 | 0.0597 | 0.0548 | 0.0056 |
| | | LPIPS (↑) | 0.4120 | 0.1074 | 0.0965 | 0.2240 | **0.4261** | 0.2064 | 0.0469 |
| | | CIDR (↑) | 0.1795 | 0.1062 | 0.0613 | 0.0558 | 0.0538 | 0.3361 | **0.4175** |
| | | BRISQUE (↑) | **96.9813** | 18.4010 | 31.2229 | 18.4908 | 31.8727 | 42.2994 | 19.9031 |
| | | AvgRank (↓) | **1.75** | 5.50 | 5.00 | 4.75 | 3.25 | 2.75 | 5.00 |
| | | ROB (↑) | 0.6331 | **0.9708** | 0.8997 | 0.8350 | 0.7938 | 0.9258 | 0.9505 |
| | | TE (↑) | 0.3243 | 0.3692 | **0.4011** | 0.3027 | 0.2843 | - | - |
| **Face Swap** | SimSwap | MSE (↑) | **0.0369** | 0.0279 | 0.0214 | 0.0169 | 0.0131 | 0.0053 | 0.0058 |
| | | LPIPS (↑) | **0.1151** | 0.0995 | 0.0892 | 0.0745 | 0.0885 | 0.0422 | 0.0435 |
| | | CIDR (↑) | 0.4160 | 0.3640 | 0.3388 | 0.2902 | **0.5810** | 0.3163 | 0.4597 |
| | | BRISQUE (↑) | 4.8177 | **5.1509** | 4.3073 | 3.4305 | 2.8553 | 1.5410 | 1.8451 |
| | | AvgRank (↓) | **1.75** | 2.25 | 3.50 | 5.00 | 3.75 | 6.75 | 5.00 |
| | | ROB (↑) | 0.7154 | 0.9896 | 0.9069 | 0.6596 | 0.7505 | 0.9967 | **0.9993** |
| | | TE (↑) | 0.3565 | **0.5587** | 0.5301 | 0.3728 | 0.3858 | - | - |
| | pSp-mix | MSE (↑) | **0.2789** | 0.0625 | 0.1001 | 0.1186 | 0.0532 | 0.0274 | 0.0037 |
| | | LPIPS (↑) | **0.3107** | 0.1445 | 0.1857 | 0.1888 | 0.2180 | 0.1131 | 0.0258 |
| | | CIDR (↑) | 0.2100 | 0.1276 | 0.1301 | 0.0847 | **0.2138** | 0.1801 | 0.2818 |
| | | BRISQUE (↑) | 5.2977 | **6.0136** | 3.7429 | 4.7629 | 3.9077 | 3.9905 | 3.3939 |
| | | AvgRank (↓) | **1.75** | 4.00 | 4.50 | 3.75 | 3.50 | 5.00 | 5.50 |
| | | ROB (↑) | 0.9420 | **1.000** | 0.9991 | 0.9962 | 0.9757 | 0.9994 | **1.000** |
| | | TE (↑) | 0.3916 | **0.5488** | 0.5252 | 0.4215 | 0.3770 | - | - |
| | BlendFace | MSE (↑) | **0.0644** | 0.0282 | 0.0162 | 0.0349 | 0.0059 | 0.0018 | 0.0024 |
| | | LPIPS (↑) | **0.0666** | 0.0365 | 0.0245 | 0.0419 | 0.0249 | 0.0090 | 0.0116 |
| | | CIDR (↑) | 0.4044 | 0.2861 | 0.2372 | 0.3168 | **0.5614** | 0.2419 | 0.3848 |
| | | BRISQUE (↑) | -6.8507 | -6.4102 | -6.1881 | -6.3018 | **-5.6363** | -6.1336 | -6.1690 |
| | | AvgRank (↓) | **2.75** | 4.00 | 4.75 | 3.50 | **2.75** | 6.00 | 4.25 |
| | | ROB (↑) | 0.8074 | 0.9397 | 0.9543 | 0.6324 | 0.8609 | **0.9997** | 0.9995 |
| | | TE (↑) | 0.3733 | **0.5419** | 0.5339 | 0.3803 | 0.3598 | - | - |
| | DiffSwap | MSE (↑) | 0.0051 | 0.0050 | 0.0052 | 0.0063 | **0.0115** | 0.0057 | 0.0059 |
| | | LPIPS (↑) | 0.0372 | 0.0372 | 0.0378 | 0.0414 | **0.0939** | 0.0406 | 0.0416 |
| | | CIDR (↑) | 0.0435 | 0.0331 | 0.0446 | 0.0524 | **0.2939** | 0.1322 | 0.2249 |
| | | BRISQUE (↑) | -4.8245 | -4.7708 | **-4.6004** | -5.2541 | -5.1350 | -5.0286 | -5.0176 |
| | | AvgRank (↓) | 5.38 | 5.63 | 4.00 | 4.50 | **2.25** | 3.75 | 2.50 |
| | | ROB (↑) | 0.9539 | **0.9894** | 0.9745 | 0.8873 | 0.8501 | 0.9762 | 0.9856 |
| | | TE (↑) | 0.1861 | 0.2620 | **0.3926** | 0.2480 | 0.3357 | - | - |
| | DiffFace | MSE (↑) | 0.0103 | 0.0106 | 0.0107 | 0.0103 | **0.0196** | 0.0134 | 0.0151 |
| | | LPIPS (↑) | 0.0539 | 0.0547 | 0.0547 | 0.0545 | **0.0887** | 0.0664 | 0.0710 |
| | | CIDR (↑) | 0.0394 | 0.0428 | 0.0447 | 0.0284 | **0.6259** | 0.2994 | 0.4539 |
| | | BRISQUE (↑) | -2.4824 | -2.4362 | -2.2375 | -2.7412 | -2.2422 | -2.2695 | **-1.9022** |
| | | AvgRank (↓) | 6.38 | 5.13 | 3.88 | 6.13 | **1.50** | 3.25 | 1.75 |
| | | ROB (↑) | 0.5242 | 0.5201 | 0.4983 | **0.5493** | 0.4644 | 0.5119 | 0.4730 |
| | | TE (↑) | 0.1955 | 0.2626 | **0.3988** | 0.2396 | 0.3566 | - | - |

## B.4. Additional Quantitative Results using VGGFace2-HQ

*Table 9.* Additional results on VGGFace2-HQ using the same evaluation setup as Table 2.

| Task | Generator | Metrics | Proactive Defenses | | | | | | |
|---|---|---|---|---|---|---|---|---|---|
| | | | White-box | | | | | Black-box | |
| | | | PGD | Disrupting | DF-RAP | Anti-Forgery | Latent Attack | SCOL | NullSwap |
| **Attribute Manipulation** | StarGAN | MSE (↑) | 1.2887 | 0.2424 | 0.2404 | **1.4225** | 0.0737 | 0.0154 | 0.0080 |
| | | LPIPS (↑) | **0.6701** | 0.4395 | 0.3876 | 0.6677 | 0.3691 | 0.1073 | 0.0362 |
| | | CIDR (↑) | 0.3564 | 0.0895 | 0.0346 | **0.3900** | 0.1140 | 0.1563 | 0.2197 |
| | | BRISQUE (↑) | 33.6946 | 23.7841 | 31.5060 | 28.4784 | **34.8548** | 18.3388 | 17.9882 |
| | | AvgRank (↓) | **1.75** | 4.25 | 4.50 | 2.00 | 4.00 | 5.50 | 6.00 |
| | | ROB (↑) | 0.7068 | 0.8693 | 0.8255 | 0.6329 | 0.7474 | 0.9770 | **0.9986** |
| | | TE (↑) | 0.4821 | **0.5810** | 0.5772 | 0.5481 | 0.4139 | - | - |
| | StyleCLIP | MSE (↑) | **2.7490** | 0.2223 | 0.2908 | 0.4597 | 0.1876 | 0.0092 | 0.0108 |
| | | LPIPS (↑) | **0.6537** | 0.3401 | 0.4407 | 0.4166 | 0.5911 | 0.1941 | 0.0951 |
| | | CIDR (↑) | 0.3909 | 0.1854 | 0.2028 | 0.1996 | **0.4386** | 0.2193 | 0.3832 |
| | | BRISQUE (↑) | **24.1783** | 20.5644 | 11.6138 | 18.8123 | 12.2644 | 5.7525 | 15.0691 |
| | | AvgRank (↓) | **1.25** | 4.50 | 4.25 | 3.75 | 3.25 | 6.00 | 5.00 |
| | | ROB (↑) | 0.7647 | 0.9670 | 0.9460 | 0.7600 | 0.8353 | 0.9791 | **0.9859** |
| | | TE (↑) | 0.5214 | **0.6701** | 0.6350 | 0.5780 | 0.4889 | - | - |
| | DiffAE | MSE (↑) | **0.0962** | 0.0170 | 0.0207 | 0.0051 | 0.0644 | 0.0107 | 0.0067 |
| | | LPIPS (↑) | 0.3468 | 0.1686 | 0.1273 | 0.0798 | **0.4443** | 0.1396 | 0.0436 |
| | | CIDR (↑) | 0.0956 | 0.0973 | 0.0690 | 0.0213 | 0.0207 | 0.1517 | **0.3100** |
| | | BRISQUE (↑) | **88.9888** | 17.2793 | 32.5011 | 11.9436 | 31.3392 | 42.4998 | 18.1879 |
| | | AvgRank (↓) | **2.00** | 4.00 | 3.75 | 6.75 | 3.50 | 3.50 | 4.50 |
| | | ROB (↑) | 0.6439 | 0.9366 | 0.8663 | 0.9376 | 0.7932 | 0.9093 | **0.9431** |
| | | TE (↑) | 0.4594 | 0.5158 | **0.5330** | 0.4356 | 0.3979 | - | - |
| **Face Swap** | SimSwap | MSE (↑) | **0.0346** | 0.0241 | 0.0169 | 0.0129 | 0.0100 | 0.0029 | 0.0042 |
| | | LPIPS (↑) | **0.0701** | 0.0542 | 0.0476 | 0.0369 | 0.0500 | 0.0163 | 0.0229 |
| | | CIDR (↑) | 0.4183 | 0.3499 | 0.3321 | 0.2678 | **0.5860** | 0.2491 | 0.4913 |
| | | BRISQUE (↑) | **26.6891** | 26.4644 | 25.8577 | 25.6680 | 24.7518 | 25.6100 | 25.7300 |
| | | AvgRank (↓) | **1.50** | 2.50 | 3.50 | 5.00 | 4.25 | 6.75 | 4.50 |
| | | ROB (↑) | 0.7612 | 0.9927 | 0.9196 | 0.7135 | 0.7847 | 0.9973 | **0.9993** |
| | | TE (↑) | 0.4852 | **0.6857** | 0.6362 | 0.5616 | 0.4991 | - | - |
| | pSp-mix | MSE (↑) | **0.3753** | 0.0863 | 0.1345 | 0.1591 | 0.0642 | 0.0051 | 0.0037 |
| | | LPIPS (↑) | **0.2871** | 0.1206 | 0.1524 | 0.1605 | 0.1739 | 0.0244 | 0.0188 |
| | | CIDR (↑) | 0.1359 | 0.0685 | 0.0741 | 0.0597 | 0.1409 | 0.0877 | **0.2004** |
| | | BRISQUE (↑) | **19.6125** | 18.1347 | 18.7614 | 19.0509 | 16.4725 | 18.8466 | 18.3905 |
| | | AvgRank (↓) | **1.50** | 5.25 | 4.00 | 3.50 | 4.00 | 4.75 | 5.00 |
| | | ROB (↑) | 0.8398 | 0.9605 | 0.9486 | 0.7850 | 0.8389 | 0.9940 | **0.9999** |
| | | TE (↑) | 0.5127 | **0.6635** | 0.6433 | 0.5838 | 0.4741 | - | - |
| | BlendFace | MSE (↑) | **0.0751** | 0.0337 | 0.0174 | 0.0414 | 0.0070 | 0.0012 | 0.0019 |
| | | LPIPS (↑) | **0.0596** | 0.0319 | 0.0200 | 0.0370 | 0.0227 | 0.0052 | 0.0073 |
| | | CIDR (↑) | 0.4336 | 0.3112 | 0.2319 | 0.3524 | **0.5820** | 0.1641 | 0.3720 |
| | | BRISQUE (↑) | 18.9811 | 19.1206 | 19.0711 | 19.1808 | 19.3738 | **20.0599** | 19.6909 |
| | | AvgRank (↓) | **2.75** | 4.25 | 5.00 | 3.00 | 3.25 | 5.50 | 4.25 |
| | | ROB (↑) | 0.7969 | 0.9358 | 0.9512 | 0.6181 | 0.8215 | **0.9997** | 0.9995 |
| | | TE (↑) | 0.4850 | **0.6471** | 0.6269 | 0.5418 | 0.4452 | - | - |
| | DiffSwap | MSE (↑) | 0.0061 | 0.0057 | 0.0062 | 0.0060 | **0.0143** | 0.0062 | 0.0066 |
| | | LPIPS (↑) | 0.0380 | 0.0378 | 0.0390 | 0.0379 | **0.0923** | 0.0395 | 0.0414 |
| | | CIDR (↑) | 0.0413 | 0.0391 | 0.0337 | 0.0450 | **0.2418** | 0.0639 | 0.1593 |
| | | BRISQUE (↑) | 15.5201 | 15.6934 | 15.5980 | **15.8499** | 14.8338 | 15.5943 | 15.3361 |
| | | AvgRank (↓) | 5.50 | 5.50 | 4.63 | 3.75 | **2.50** | 3.13 | 3.00 |
| | | ROB (↑) | 0.6744 | 0.6736 | 0.6735 | **0.6751** | 0.6209 | 0.6733 | 0.6699 |
| | | TE (↑) | 0.3139 | 0.3766 | **0.4952** | 0.3992 | 0.4271 | - | - |
| | DiffFace | MSE (↑) | 0.0125 | 0.0127 | 0.0129 | 0.0135 | **0.0232** | 0.0152 | 0.0166 |
| | | LPIPS (↑) | 0.0470 | 0.0477 | 0.0493 | 0.0498 | **0.0804** | 0.0564 | 0.0608 |
| | | CIDR (↑) | 0.0397 | 0.0330 | 0.0464 | 0.0381 | **0.6275** | 0.2389 | 0.4437 |
| | | BRISQUE (↑) | 9.9247 | 9.6763 | 9.5813 | **10.2244** | 9.5931 | 9.3581 | 9.5360 |
| | | AvgRank (↓) | 5.25 | 5.50 | 4.75 | 3.75 | **1.75** | 4.00 | 3.00 |
| | | ROB (↑) | 0.7992 | **0.8029** | 0.8015 | 0.7996 | 0.7304 | 0.7989 | **0.8029** |
| | | TE (↑) | 0.3158 | 0.3860 | **0.4999** | 0.3996 | 0.4504 | - | - |

## B.5. Visualization of Principal Results

This section presents a comprehensive visualization of proactive defense performance across all evaluated generators and datasets. As shown in Figure 7 and Figure 8, generators are organized into two categories: attribute manipulation models and face swapping models. Each row corresponds to a generator, while columns represent different datasets (CelebA-HQ, FFHQ, and VGGFace2-HQ).

Each radar plot summarizes multiple evaluation metrics, capturing complementary aspects of performance, including pixel-level fidelity (MSE), perceptual similarity (LPIPS), identity disruption (CIDR), visual quality (BRISQUE), robustness (ROB), and transferability (TE). This visualization complements the detailed analysis on SimSwap in Figure 1 by providing a holistic comparison across diverse generators and data distributions.

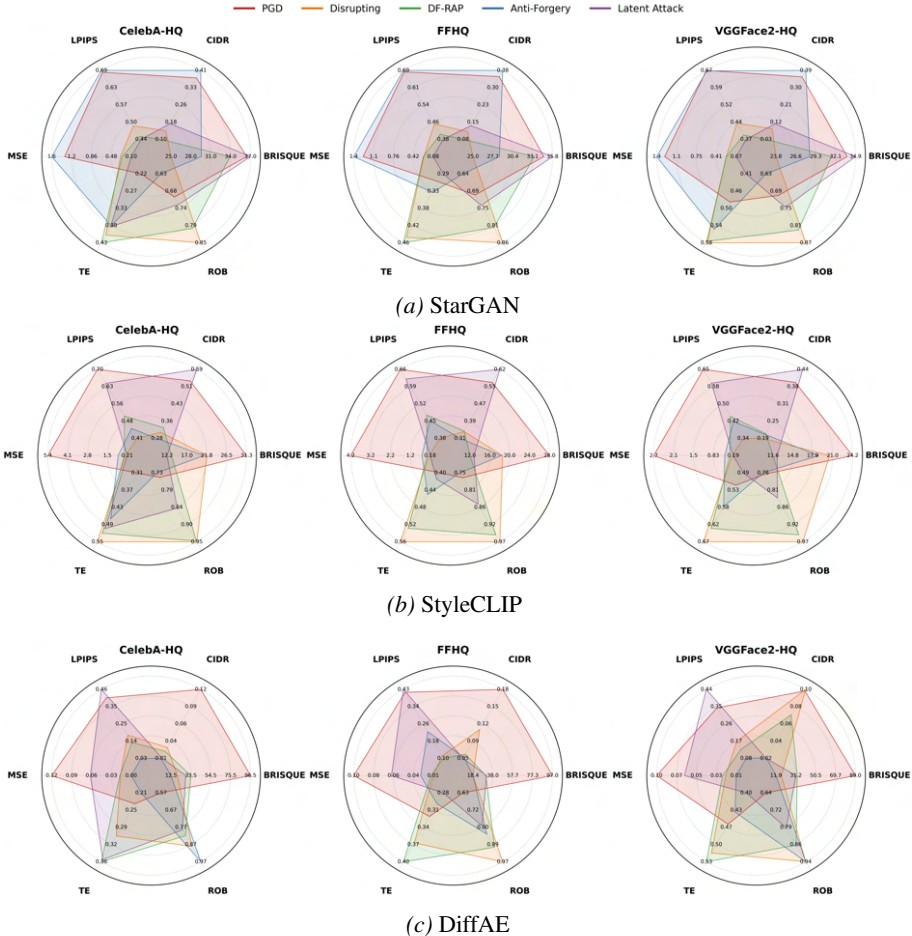

*Figure 7.* Radar charts of proactive defense performance on attribute manipulation models across three datasets (CelebA-HQ, FFHQ, and VGGFace2-HQ). Rows denote generators and columns denote datasets.

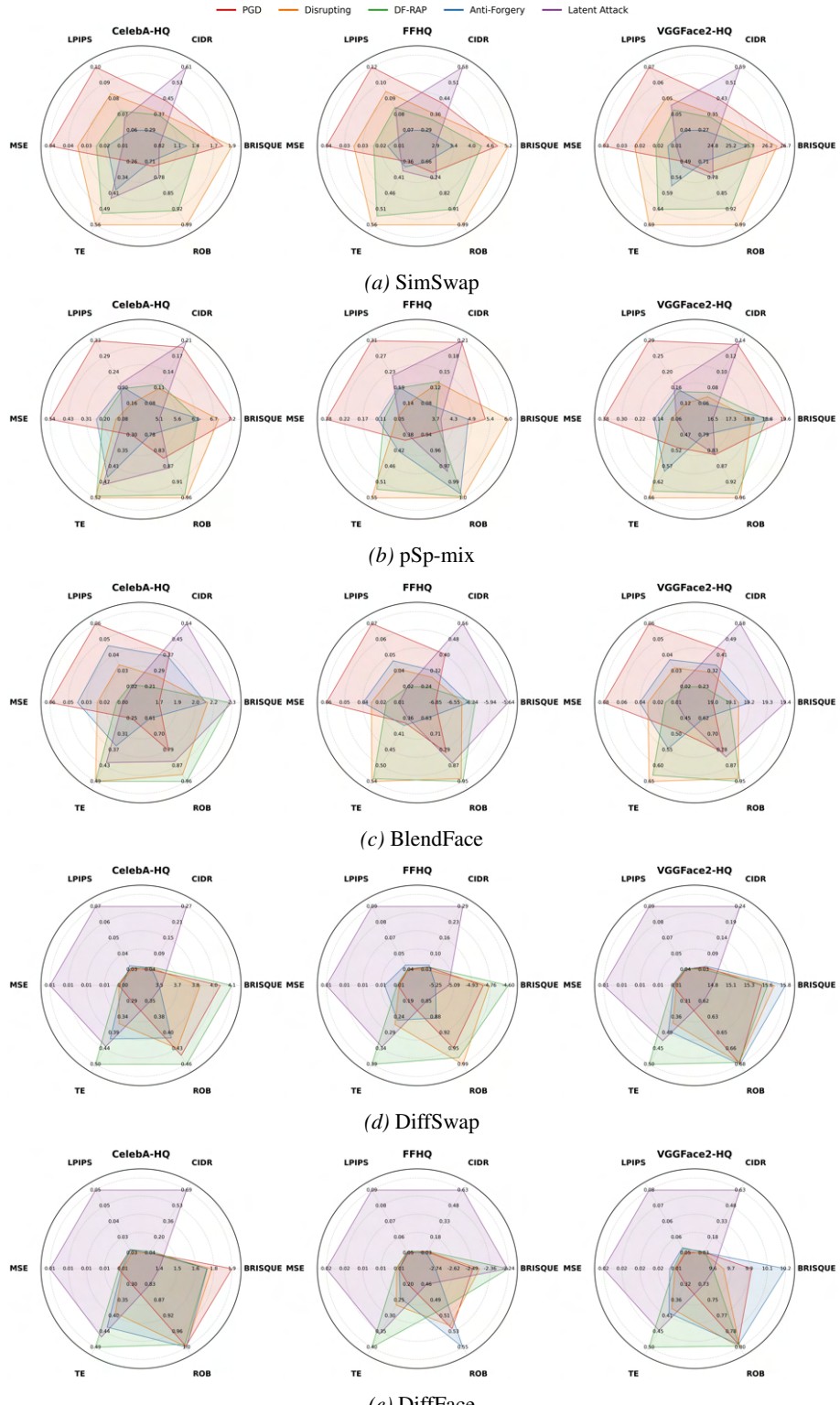

*Figure 8.* Radar charts of proactive defense performance on face swap models across three datasets (CelebA-HQ, FFHQ, and VGGFace2-HQ). Rows denote generators and columns denote datasets.

## B.6. Visualization of AvgRank Results

This section visualizes the AvgRank-based rankings of proactive defenses across all evaluated generators. While Figure 5 presents representative results on a subset of models, we provide complete ranking visualizations here for comprehensive comparison.

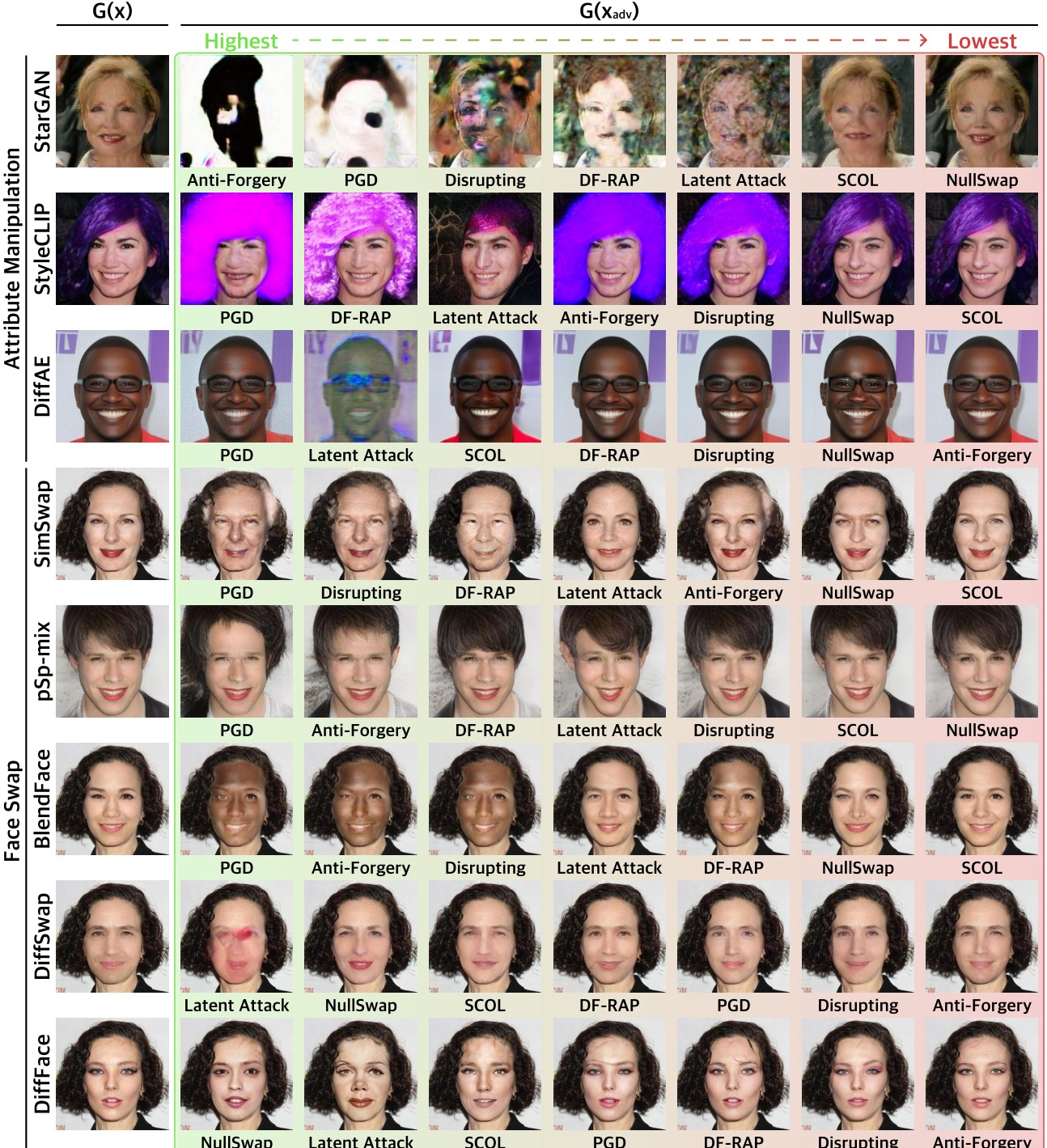

*Figure 9.* Qualitative comparison of proactive defenses sorted by AvgRank. For each generator, defense methods are arranged in ascending order of their AvgRank (from highest effectiveness on the left to lowest on the right).

## B.7. Imperceptibility

To examine the visual imperceptibility of proactive defenses reported in Table 10, we report pixel-wise and perceptual fidelity metrics, including MSE, PSNR, SSIM, and LPIPS, across representative generators and defense methods. Lower MSE and LPIPS values and higher PSNR and SSIM values correspond to higher visual fidelity.

As shown in Figure 10, optimization-based approaches such as DF-RAP and LEAT tend to preserve visual fidelity, as they apply constrained perturbations directly to the original images under a fixed budget. In contrast, model-based learning approaches, including NullSwap and SCOL, tend to exhibit relatively larger visual differences compared to optimization methods, as they rely on image reconstruction mechanisms.

*Table 10.* The tendency of imperceptibility results for proactive defense methods on StarGAN (attribute manipulation) and SimSwap (face swap). Other generators exhibit similar trends.

| Generator | Metric | White-box | | | | | Black-box | |
|---|---|---|---|---|---|---|---|---|
| | | PGD | Disrupting | DF-RAP | Anti-Forgery | LEAT | SCOL | NullSwap |
| StarGAN | MSE (↓) | 0.0013 | 0.0018 | 0.0012 | 0.0002 | 0.0024 | 0.0617 | 0.0036 |
| | PSNR (↑) | 34.7939 | 33.3090 | 34.9363 | 42.4811 | 32.1176 | 18.5619 | 30.5156 |
| | SSIM (↑) | 0.8705 | 0.8967 | 0.8761 | 0.9764 | 0.7930 | 0.5813 | 0.9456 |
| | LPIPS (↓) | 0.0225 | 0.0639 | 0.0772 | 0.0004 | 0.1086 | 0.2145 | 0.0137 |
| SimSwap | MSE (↓) | 0.0009 | 0.0017 | 0.0012 | 0.0004 | 0.0010 | 0.0617 | 0.0036 |
| | PSNR (↑) | 36.0479 | 33.6250 | 35.2073 | 39.1039 | 35.9519 | 18.5619 | 30.5156 |
| | SSIM (↑) | 0.8994 | 0.8951 | 0.8917 | 0.9494 | 0.8977 | 0.5813 | 0.9456 |
| | LPIPS (↓) | 0.0601 | 0.0641 | 0.0828 | 0.0077 | 0.0621 | 0.2145 | 0.0137 |

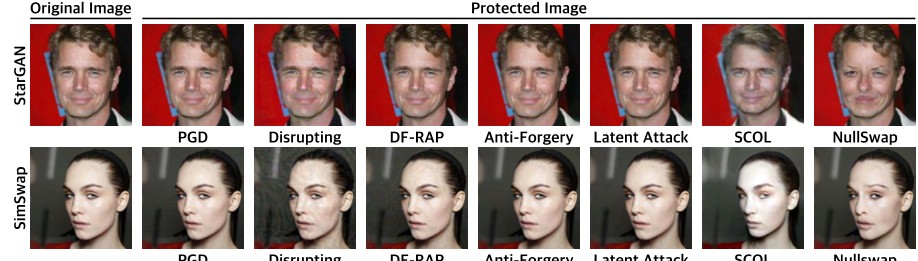

*Figure 10.* Representative visual comparisons between original images and their protected images produced by different proactive defense methods on StarGAN and SimSwap.

# C. Robustness

## C.1. post-processing transformations for Robustness

To design a realistic and representative robustness evaluation protocol, we systematically review post-processing perturbations adopted in prior proactive deepfake defense studies. Table 11 summarizes commonly used transformations, including JPEG compression, blur, noise injection, color manipulation, and image reconstruction. Based on this analysis, we select a subset of widely adopted perturbations that reflect practical image degradation scenarios encountered in real-world deployment.

| Perturbation Robustness Evaluation | Evaluated in Papers |
|---|---|
| JPEG Compression | (Wang et al., 2022), (Qu et al., 2024), (Zhang et al., 2024), (Qu et al., 2025), (Jeong et al., 2025), (Wang et al., 2025a) |
| Gaussian Blur | (Ruiz et al., 2020), (Wang et al., 2022), (Zhang et al., 2024), (Qu et al., 2025), (Wang et al., 2025a) |
| Box Blur | (Ruiz et al., 2020) |
| Image Resizing | (Zhang et al., 2024) |
| Gaussian Noise | (Zhang et al., 2024) |
| Salt-and-Pepper Noise | (Zhang et al., 2024) |
| Image Reconstruction (MagDR) | (Wang et al., 2022) |

*Table 11.* Comparison of robustness in proactive deepfake defense methods.

## C.2. Visualizations of robustness Results

This section presents comprehensive visualizations of defense robustness across all evaluated generators. While Figure 3 reports representative results on StyleCLIP and BlendFace, we extend the analysis to all generators to provide a complete view of robustness under post-processing transformations.

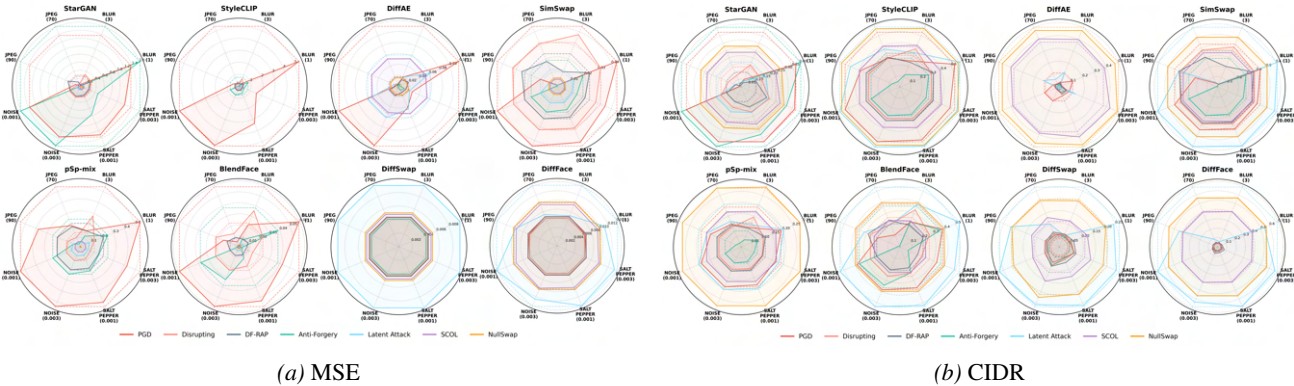

*(a)* MSE                    *(b)* CIDR

*Figure 11.* Robustness against post-processing across all evaluated generators. The axes represent transformation types (JPEG compression, Blur, Gaussian Noise, Salt & Pepper) with varying intensities. Solid lines represent performance under transformations, while dashed lines indicate the reference performance without post-processing.

## C.3. Quantitative Robustness Results

Following the visual analysis in Appendix C.2, we report detailed quantitative robustness results for each disrupting method in this section. While Figure 11 provides intuitive comparisons under post-processing transformations, the following tables summarize per-metric performance values to support rigorous quantitative evaluation.

*Table 12.* Quantitative robustness evaluation of PGD under various post-processing transformations.

*(a)* MSE

| Post-processing | StarGAN | StyleCLIP | DiffAE | SimSwap | pSp-mix | BlendFace | DiffSwap | DiffFace |
|---|---|---|---|---|---|---|---|---|
| No Post-processing | 1.3715 | 5.4086 | 0.1185 | 0.0440 | 0.5427 | 0.0591 | 0.0046 | 0.0069 |
| Blur ($\sigma$=1) | 1.3715 | 5.4081 | 0.1186 | 0.0440 | 0.5427 | 0.0591 | 0.0046 | 0.0065 |
| Blur ($\sigma$=3) | 0.0562 | 0.1976 | 0.0178 | 0.0007 | 0.1443 | 0.0216 | 0.0046 | 0.0069 |
| JPEG (Q=70) | 0.0021 | 0.2187 | 0.0030 | 0.0010 | 0.1913 | 0.0048 | 0.0046 | 0.0068 |
| JPEG (Q=90) | 0.1190 | 0.5121 | 0.0041 | 0.0123 | 0.3809 | 0.0153 | 0.0046 | 0.0066 |
| Noise ($\sigma$=0.001) | 1.3715 | 5.3970 | 0.1184 | 0.0440 | 0.5425 | 0.0591 | 0.0046 | 0.0068 |
| Noise ($\sigma$=0.003) | 1.3708 | 5.3416 | 0.1173 | 0.0439 | 0.5419 | 0.0590 | 0.0046 | 0.0069 |
| S&P (p=0.001) | 1.3144 | 3.3594 | 0.0422 | 0.0412 | 0.5037 | 0.0535 | 0.0046 | 0.0069 |
| S&P (p=0.003) | 1.2037 | 1.5547 | 0.0265 | 0.0353 | 0.4232 | 0.0426 | 0.0046 | 0.0068 |

*(b)* LPIPS

| Post-processing | StarGAN | StyleCLIP | DiffAE | SimSwap | pSp-mix | BlendFace | DiffSwap | DiffFace |
|---|---|---|---|---|---|---|---|---|
| No Post-processing | 0.6878 | 0.7042 | 0.4070 | 0.0963 | 0.3331 | 0.0552 | 0.0293 | 0.0282 |
| Blur ($\sigma$=1) | 0.6878 | 0.7042 | 0.4070 | 0.0963 | 0.3331 | 0.0552 | 0.0293 | 0.0282 |
| Blur ($\sigma$=3) | 0.2314 | 0.4352 | 0.1854 | 0.0055 | 0.1722 | 0.0261 | 0.0293 | 0.0282 |
| JPEG (Q=70) | 0.0238 | 0.4264 | 0.0325 | 0.0069 | 0.1945 | 0.0091 | 0.0293 | 0.0282 |
| JPEG (Q=90) | 0.2166 | 0.5628 | 0.0529 | 0.0436 | 0.2691 | 0.0199 | 0.0293 | 0.0282 |
| Noise ($\sigma$=0.001) | 0.6877 | 0.7041 | 0.4068 | 0.0963 | 0.3330 | 0.0552 | 0.0293 | 0.0282 |
| Noise ($\sigma$=0.003) | 0.6876 | 0.7039 | 0.4044 | 0.0963 | 0.3326 | 0.0551 | 0.0293 | 0.0282 |
| S&P (p=0.001) | 0.6793 | 0.6922 | 0.2008 | 0.0929 | 0.3174 | 0.0511 | 0.0293 | 0.0282 |
| S&P (p=0.003) | 0.6693 | 0.6694 | 0.1951 | 0.0860 | 0.2899 | 0.0431 | 0.0293 | 0.0282 |

*(c)* CIDR

| Post-processing | StarGAN | StyleCLIP | DiffAE | SimSwap | pSp-mix | BlendFace | DiffSwap | DiffFace |
|---|---|---|---|---|---|---|---|---|
| No Post-processing | 0.3765 | 0.5345 | 0.1184 | 0.4407 | 0.1937 | 0.3848 | 0.0363 | 0.0362 |
| Blur ($\sigma$=1) | 0.3765 | 0.5345 | 0.1184 | 0.4407 | 0.1936 | 0.3848 | 0.0477 | 0.0445 |
| Blur ($\sigma$=3) | 0.0531 | 0.2546 | 0.0378 | 0.0284 | 0.0947 | 0.2268 | 0.0527 | 0.0388 |
| JPEG (Q=70) | 0.0213 | 0.2811 | 0.0243 | 0.0394 | 0.1113 | 0.1081 | 0.0524 | 0.0407 |
| JPEG (Q=90) | 0.0105 | 0.3847 | 0.0461 | 0.2289 | 0.1490 | 0.1982 | 0.0546 | 0.0394 |
| Noise ($\sigma$=0.001) | 0.3765 | 0.5344 | 0.1183 | 0.4407 | 0.1936 | 0.3847 | 0.0547 | 0.0466 |
| Noise ($\sigma$=0.003) | 0.3763 | 0.5339 | 0.1173 | 0.4407 | 0.1934 | 0.3845 | 0.0697 | 0.0514 |
| S&P (p=0.001) | 0.3770 | 0.5174 | 0.0750 | 0.4306 | 0.1845 | 0.3678 | 0.0483 | 0.0515 |
| S&P (p=0.003) | 0.3678 | 0.4904 | 0.0898 | 0.4104 | 0.1656 | 0.3319 | 0.0684 | 0.0501 |

*(d)* BRISQUE

| Post-processing | StarGAN | StyleCLIP | DiffAE | SimSwap | pSp-mix | BlendFace | DiffSwap | DiffFace |
|---|---|---|---|---|---|---|---|---|
| No Post-processing | 36.4792 | 31.2795 | 96.5489 | 1.7925 | 7.1814 | 1.7485 | 4.0007 | 1.8878 |
| Blur ($\sigma$=1) | 36.4822 | 31.2840 | 96.5569 | 1.7925 | 7.1804 | 1.7531 | -4.8031 | 1.8878 |
| Blur ($\sigma$=3) | 21.4911 | 33.1753 | 17.8541 | 1.2955 | 7.1325 | 1.7993 | -5.1175 | 1.8878 |
| JPEG (Q=70) | 17.3788 | 13.0823 | 16.1227 | 1.3277 | 6.3278 | 2.2660 | -4.8531 | 1.8878 |
| JPEG (Q=90) | 21.4017 | 13.9339 | 13.7555 | 1.5447 | 6.2618 | 1.8718 | -5.0995 | 1.8878 |
| Noise ($\sigma$=0.001) | 36.4790 | 31.2698 | 96.4781 | 1.8038 | 7.1659 | 1.7687 | -4.9357 | 1.8876 |
| Noise ($\sigma$=0.003) | 36.5107 | 31.0834 | 95.8322 | 1.8021 | 7.1531 | 1.7579 | -4.4992 | 1.8878 |
| S&P (p=0.001) | 35.8280 | 26.5648 | 71.7397 | 1.6926 | 7.0919 | 1.7087 | -4.9699 | 1.8878 |
| S&P (p=0.003) | 35.7167 | 20.1636 | 39.5108 | 1.6302 | 7.0361 | 1.7185 | -4.9134 | 1.8878 |

*Table 13.* Quantitative robustness evaluation of Disrupting under various post-processing transformations.

*(a)* MSE

| Post-processing | StarGAN | StyleCLIP | DiffAE | SimSwap | pSp-mix | BlendFace | DiffSwap | DiffFace |
|---|---|---|---|---|---|---|---|---|
| No Post-processing | 0.3003 | 0.3308 | 0.0178 | 0.0320 | 0.1270 | 0.0233 | 0.0046 | 0.0064 |
| Blur ($\sigma$=1) | 0.3003 | 0.3308 | 0.0178 | 0.0320 | 0.1270 | 0.0233 | 0.0046 | 0.0064 |
| Blur ($\sigma$=3) | 0.3166 | 0.6951 | 0.0160 | 0.0378 | 0.2790 | 0.0354 | 0.0046 | 0.0064 |
| JPEG (Q=70) | 0.0596 | 0.2117 | 0.0044 | 0.0312 | 0.0835 | 0.0127 | 0.0046 | 0.0064 |
| JPEG (Q=90) | 0.0796 | 0.2879 | 0.0066 | 0.0327 | 0.1155 | 0.0228 | 0.0046 | 0.0064 |
| Noise ($\sigma$=0.001) | 0.3003 | 0.3305 | 0.0178 | 0.0320 | 0.1269 | 0.0233 | 0.0046 | 0.0064 |
| Noise ($\sigma$=0.003) | 0.2998 | 0.3292 | 0.0179 | 0.0320 | 0.1265 | 0.0231 | 0.0046 | 0.0064 |
| S&P (p=0.001) | 0.2706 | 0.2949 | 0.0133 | 0.0309 | 0.1130 | 0.0190 | 0.0046 | 0.0064 |
| S&P (p=0.003) | 0.2479 | 0.2433 | 0.0145 | 0.0285 | 0.0918 | 0.0135 | 0.0046 | 0.0064 |

*(b)* LPIPS

| Post-processing | StarGAN | StyleCLIP | DiffAE | SimSwap | pSp-mix | BlendFace | DiffSwap | DiffFace |
|---|---|---|---|---|---|---|---|---|
| No Post-processing | 0.4872 | 0.4069 | 0.1757 | 0.0801 | 0.1557 | 0.0291 | 0.0283 | 0.0269 |
| Blur ($\sigma$=1) | 0.4872 | 0.4069 | 0.1757 | 0.0801 | 0.1557 | 0.0291 | 0.0283 | 0.0269 |
| Blur ($\sigma$=3) | 0.3431 | 0.5391 | 0.1778 | 0.0886 | 0.2418 | 0.0397 | 0.0283 | 0.0269 |
| JPEG (Q=70) | 0.1285 | 0.3555 | 0.0564 | 0.0788 | 0.1283 | 0.0186 | 0.0283 | 0.0269 |
| JPEG (Q=90) | 0.1458 | 0.3962 | 0.0846 | 0.0811 | 0.1490 | 0.0283 | 0.0283 | 0.0269 |
| Noise ($\sigma$=0.001) | 0.4872 | 0.4068 | 0.1756 | 0.0800 | 0.1556 | 0.0291 | 0.0283 | 0.0269 |
| Noise ($\sigma$=0.003) | 0.4876 | 0.4066 | 0.1747 | 0.0800 | 0.1553 | 0.0289 | 0.0283 | 0.0269 |
| S&P (p=0.001) | 0.5072 | 0.4079 | 0.2102 | 0.0785 | 0.1510 | 0.0252 | 0.0283 | 0.0269 |
| S&P (p=0.003) | 0.5354 | 0.4198 | 0.2076 | 0.0757 | 0.1434 | 0.0199 | 0.0283 | 0.0269 |

*(c)* CIDR

| Post-processing | StarGAN | StyleCLIP | DiffAE | SimSwap | pSp-mix | BlendFace | DiffSwap | DiffFace |
|---|---|---|---|---|---|---|---|---|
| No Post-processing | 0.1385 | 0.3080 | 0.0275 | 0.3821 | 0.1091 | 0.2645 | 0.0425 | 0.0387 |
| Blur ($\sigma$=1) | 0.1385 | 0.3080 | 0.0276 | 0.3821 | 0.1090 | 0.2645 | 0.0547 | 0.0421 |
| Blur ($\sigma$=3) | 0.1524 | 0.3721 | 0.0454 | 0.4101 | 0.1452 | 0.3336 | 0.0564 | 0.0402 |
| JPEG (Q=70) | 0.0778 | 0.2824 | 0.0295 | 0.3803 | 0.0981 | 0.2051 | 0.0580 | 0.0372 |
| JPEG (Q=90) | 0.0869 | 0.3033 | 0.0434 | 0.3878 | 0.1085 | 0.2658 | 0.0648 | 0.0383 |
| Noise ($\sigma$=0.001) | 0.1382 | 0.3079 | 0.0276 | 0.3821 | 0.1090 | 0.2643 | 0.0670 | 0.0521 |
| Noise ($\sigma$=0.003) | 0.1403 | 0.3077 | 0.0277 | 0.3821 | 0.1089 | 0.2633 | 0.0828 | 0.0578 |
| S&P (p=0.001) | 0.1717 | 0.3064 | 0.0568 | 0.3785 | 0.1053 | 0.2353 | 0.0580 | 0.0478 |
| S&P (p=0.003) | 0.1948 | 0.3066 | 0.0723 | 0.3737 | 0.1013 | 0.1964 | 0.0653 | 0.0600 |

*(d)* BRISQUE

| Post-processing | StarGAN | StyleCLIP | DiffAE | SimSwap | pSp-mix | BlendFace | DiffSwap | DiffFace |
|---|---|---|---|---|---|---|---|---|
| No Post-processing | 25.0217 | 21.7907 | 19.5244 | 1.9301 | 6.7925 | 2.1221 | 3.8500 | 1.7522 |
| Blur ($\sigma$=1) | 25.0177 | 21.7900 | 19.4999 | 1.9301 | 6.7891 | 2.1205 | -5.1296 | 1.7522 |
| Blur ($\sigma$=3) | 23.8909 | 44.6598 | 14.1459 | 1.8704 | 8.0037 | 2.1624 | -4.9978 | 1.7522 |
| JPEG (Q=70) | 19.6231 | 19.2952 | 16.3151 | 1.8700 | 6.9016 | 2.1358 | -4.8022 | 1.7522 |
| JPEG (Q=90) | 20.0939 | 20.8985 | 15.5738 | 1.8959 | 6.7626 | 2.1550 | -4.7816 | 1.7522 |
| Noise ($\sigma$=0.001) | 25.0113 | 21.7492 | 19.5536 | 1.9294 | 6.7982 | 2.1143 | -4.9020 | 1.7522 |
| Noise ($\sigma$=0.003) | 25.1751 | 21.5600 | 19.4216 | 1.9392 | 6.7723 | 2.1359 | -4.9118 | 1.7522 |
| S&P (p=0.001) | 28.5428 | 19.7225 | 51.9769 | 1.9540 | 6.6676 | 2.1561 | -4.8912 | 1.7522 |
| S&P (p=0.003) | 32.8004 | 17.7209 | 56.5645 | 1.9630 | 6.6697 | 2.1447 | -4.7649 | 1.7522 |

*Table 14.* Quantitative robustness evaluation of DF-RAP under various post-processing transformations.

*(a)* MSE

| Post-processing | StarGAN | StyleCLIP | DiffAE | SimSwap | pSp-mix | BlendFace | DiffSwap | DiffFace |
|---|---|---|---|---|---|---|---|---|
| No Post-processing | 0.2694 | 0.3917 | 0.0186 | 0.0238 | 0.2104 | 0.0088 | 0.0047 | 0.0068 |
| Blur ($\sigma$=1) | 0.2694 | 0.3917 | 0.0186 | 0.0238 | 0.2104 | 0.0088 | 0.0047 | 0.0068 |
| Blur ($\sigma$=3) | 0.0185 | 0.2238 | 0.0129 | 0.0106 | 0.1413 | 0.0076 | 0.0047 | 0.0068 |
| JPEG (Q=70) | 0.1142 | 0.3074 | 0.0037 | 0.0181 | 0.1803 | 0.0089 | 0.0047 | 0.0068 |
| JPEG (Q=90) | 0.3342 | 0.3813 | 0.0043 | 0.0260 | 0.2078 | 0.0120 | 0.0047 | 0.0068 |
| Noise ($\sigma$=0.001) | 0.2693 | 0.3915 | 0.0186 | 0.0238 | 0.2103 | 0.0088 | 0.0047 | 0.0068 |
| Noise ($\sigma$=0.003) | 0.2695 | 0.3907 | 0.0187 | 0.0238 | 0.2101 | 0.0088 | 0.0047 | 0.0068 |
| S&P (p=0.001) | 0.2417 | 0.3624 | 0.0125 | 0.0227 | 0.1951 | 0.0078 | 0.0047 | 0.0068 |
| S&P (p=0.003) | 0.2259 | 0.3068 | 0.0139 | 0.0206 | 0.1660 | 0.0065 | 0.0047 | 0.0068 |

*(b)* LPIPS

| Post-processing | StarGAN | StyleCLIP | DiffAE | SimSwap | pSp-mix | BlendFace | DiffSwap | DiffFace |
|---|---|---|---|---|---|---|---|---|
| No Post-processing | 0.4477 | 0.5030 | 0.1301 | 0.0695 | 0.1984 | 0.0164 | 0.0294 | 0.0283 |
| Blur ($\sigma$=1) | 0.4477 | 0.5030 | 0.1301 | 0.0695 | 0.1984 | 0.0164 | 0.0294 | 0.0283 |
| Blur ($\sigma$=3) | 0.1050 | 0.4124 | 0.1906 | 0.0401 | 0.1583 | 0.0149 | 0.0294 | 0.0283 |
| JPEG (Q=70) | 0.1890 | 0.4536 | 0.0439 | 0.0577 | 0.1803 | 0.0172 | 0.0294 | 0.0283 |
| JPEG (Q=90) | 0.3483 | 0.5015 | 0.0502 | 0.0737 | 0.1976 | 0.0215 | 0.0294 | 0.0283 |
| Noise ($\sigma$=0.001) | 0.4478 | 0.5030 | 0.1300 | 0.0695 | 0.1984 | 0.0164 | 0.0294 | 0.0283 |
| Noise ($\sigma$=0.003) | 0.4490 | 0.5031 | 0.1298 | 0.0695 | 0.1982 | 0.0163 | 0.0294 | 0.0283 |
| S&P (p=0.001) | 0.4782 | 0.5056 | 0.1998 | 0.0676 | 0.1920 | 0.0149 | 0.0294 | 0.0283 |
| S&P (p=0.003) | 0.5178 | 0.5096 | 0.1976 | 0.0640 | 0.1788 | 0.0128 | 0.0294 | 0.0283 |

*(c)* CIDR

| Post-processing | StarGAN | StyleCLIP | DiffAE | SimSwap | pSp-mix | BlendFace | DiffSwap | DiffFace |
|---|---|---|---|---|---|---|---|---|
| No Post-processing | 0.1003 | 0.3288 | 0.0215 | 0.3709 | 0.1172 | 0.2078 | 0.0459 | 0.0535 |
| Blur ($\sigma$=1) | 0.1003 | 0.3288 | 0.0215 | 0.3709 | 0.1172 | 0.2078 | 0.0639 | 0.0530 |
| Blur ($\sigma$=3) | 0.0279 | 0.2414 | 0.0344 | 0.2113 | 0.0902 | 0.1972 | 0.0573 | 0.0477 |
| JPEG (Q=70) | 0.0084 | 0.2902 | 0.0226 | 0.3064 | 0.1060 | 0.2381 | 0.0663 | 0.0488 |
| JPEG (Q=90) | 0.0383 | 0.3285 | 0.0318 | 0.3929 | 0.1177 | 0.2848 | 0.0606 | 0.0513 |
| Noise ($\sigma$=0.001) | 0.1001 | 0.3287 | 0.0215 | 0.3708 | 0.1172 | 0.2078 | 0.0618 | 0.0595 |
| Noise ($\sigma$=0.003) | 0.1017 | 0.3287 | 0.0216 | 0.3709 | 0.1171 | 0.2055 | 0.0867 | 0.0755 |
| S&P (p=0.001) | 0.1379 | 0.3248 | 0.0463 | 0.3623 | 0.1129 | 0.1877 | 0.0701 | 0.0663 |
| S&P (p=0.003) | 0.1768 | 0.3218 | 0.0637 | 0.3540 | 0.1039 | 0.1609 | 0.0745 | 0.0795 |

*(d)* BRISQUE

| Post-processing | StarGAN | StyleCLIP | DiffAE | SimSwap | pSp-mix | BlendFace | DiffSwap | DiffFace |
|---|---|---|---|---|---|---|---|---|
| No Post-processing | 33.9631 | 13.3239 | 33.4323 | 1.4151 | 6.1890 | 2.2879 | 4.0877 | 1.7187 |
| Blur ($\sigma$=1) | 33.9614 | 13.3217 | 33.4494 | 1.4151 | 6.2113 | 2.2824 | -4.8291 | 1.7187 |
| Blur ($\sigma$=3) | 25.4032 | 29.2013 | 4.0871 | 1.4561 | 6.7555 | 2.2844 | -4.7393 | 1.7187 |
| JPEG (Q=70) | 23.7501 | 13.8232 | 13.7654 | 1.4813 | 6.4292 | 2.3695 | -4.9117 | 1.7187 |
| JPEG (Q=90) | 28.2732 | 13.5132 | 14.1164 | 1.4518 | 6.2196 | 2.2799 | -4.6799 | 1.7187 |
| Noise ($\sigma$=0.001) | 33.9377 | 13.3278 | 33.4539 | 1.4258 | 6.1950 | 2.2775 | -4.9241 | 1.7187 |
| Noise ($\sigma$=0.003) | 34.0946 | 13.2977 | 33.4567 | 1.4253 | 6.2000 | 2.2636 | -4.5391 | 1.7187 |
| S&P (p=0.001) | 35.2406 | 13.5996 | 35.1595 | 1.3816 | 6.2819 | 2.2891 | -4.6808 | 1.7187 |
| S&P (p=0.003) | 37.3602 | 14.1166 | 44.4481 | 1.3522 | 6.6610 | 2.2468 | -4.7007 | 1.7187 |

*Table 15.* Quantitative robustness evaluation of Anti-Forgery under various post-processing transformations.

*(a)* MSE

| Post-processing | StarGAN | StyleCLIP | DiffAE | SimSwap | pSp-mix | BlendFace | DiffSwap | DiffFace |
|---|---|---|---|---|---|---|---|---|
| No Post-processing | 1.6180 | 0.8045 | 0.0020 | 0.0190 | 0.2496 | 0.0383 | 0.0043 | 0.0065 |
| Blur ($\sigma$=1) | 1.6180 | 0.8045 | 0.0020 | 0.0190 | 0.2496 | 0.0383 | 0.0043 | 0.0065 |
| Blur ($\sigma$=3) | 0.0130 | 0.1248 | 0.0041 | 0.0007 | 0.0955 | 0.0066 | 0.0043 | 0.0065 |
| JPEG (Q=70) | 0.0016 | 0.0279 | 0.0022 | 0.0000 | 0.0253 | 0.0000 | 0.0043 | 0.0065 |
| JPEG (Q=90) | 0.0012 | 0.0681 | 0.0021 | 0.0002 | 0.0591 | 0.0001 | 0.0043 | 0.0065 |
| Noise ($\sigma$=0.001) | 1.6165 | 0.8015 | 0.0021 | 0.0190 | 0.2493 | 0.0377 | 0.0043 | 0.0065 |
| Noise ($\sigma$=0.003) | 1.6052 | 0.7892 | 0.0032 | 0.0189 | 0.2483 | 0.0325 | 0.0043 | 0.0065 |
| S&P (p=0.001) | 0.7597 | 0.6411 | 0.0082 | 0.0172 | 0.2191 | 0.0114 | 0.0043 | 0.0065 |
| S&P (p=0.003) | 0.4791 | 0.4531 | 0.0107 | 0.0145 | 0.1701 | 0.0023 | 0.0043 | 0.0065 |

*(b)* LPIPS

| Post-processing | StarGAN | StyleCLIP | DiffAE | SimSwap | pSp-mix | BlendFace | DiffSwap | DiffFace |
|---|---|---|---|---|---|---|---|---|
| No Post-processing | 0.6950 | 0.4493 | 0.0315 | 0.0577 | 0.2038 | 0.0412 | 0.0309 | 0.0281 |
| Blur ($\sigma$=1) | 0.6950 | 0.4493 | 0.0315 | 0.0577 | 0.2038 | 0.0412 | 0.0309 | 0.0281 |
| Blur ($\sigma$=3) | 0.0931 | 0.3330 | 0.0931 | 0.0050 | 0.1196 | 0.0137 | 0.0309 | 0.0281 |
| JPEG (Q=70) | 0.0159 | 0.1031 | 0.0175 | 0.0007 | 0.0548 | 0.0003 | 0.0309 | 0.0281 |
| JPEG (Q=90) | 0.0121 | 0.1706 | 0.0206 | 0.0017 | 0.0872 | 0.0007 | 0.0309 | 0.0281 |
| Noise ($\sigma$=0.001) | 0.6944 | 0.4491 | 0.0329 | 0.0577 | 0.2037 | 0.0407 | 0.0309 | 0.0281 |
| Noise ($\sigma$=0.003) | 0.6891 | 0.4484 | 0.0483 | 0.0575 | 0.2031 | 0.0364 | 0.0309 | 0.0281 |
| S&P (p=0.001) | 0.5664 | 0.4434 | 0.1905 | 0.0541 | 0.1918 | 0.0171 | 0.0309 | 0.0281 |
| S&P (p=0.003) | 0.5683 | 0.4462 | 0.2044 | 0.0492 | 0.1742 | 0.0053 | 0.0309 | 0.0281 |

*(c)* CIDR

| Post-processing | StarGAN | StyleCLIP | DiffAE | SimSwap | pSp-mix | BlendFace | DiffSwap | DiffFace |
|---|---|---|---|---|---|---|---|---|
| No Post-processing | 0.4103 | 0.2809 | 0.0105 | 0.2935 | 0.0781 | 0.3668 | 0.0430 | 0.0361 |
| Blur ($\sigma$=1) | 0.4103 | 0.2809 | 0.0105 | 0.2935 | 0.0781 | 0.3668 | 0.0502 | 0.0381 |
| Blur ($\sigma$=3) | 0.0402 | 0.1209 | 0.0259 | 0.0253 | 0.0346 | 0.2262 | 0.0479 | 0.0386 |
| JPEG (Q=70) | 0.0183 | 0.0512 | 0.0124 | 0.0067 | 0.0182 | 0.0080 | 0.0600 | 0.0365 |
| JPEG (Q=90) | 0.0094 | 0.0746 | 0.0113 | 0.0112 | 0.0249 | 0.0131 | 0.0537 | 0.0409 |
| Noise ($\sigma$=0.001) | 0.4110 | 0.2806 | 0.0115 | 0.2934 | 0.0781 | 0.3636 | 0.0512 | 0.0404 |
| Noise ($\sigma$=0.003) | 0.4102 | 0.2794 | 0.0181 | 0.2930 | 0.0778 | 0.3443 | 0.0819 | 0.0486 |
| S&P (p=0.001) | 0.3270 | 0.2689 | 0.0627 | 0.2810 | 0.0714 | 0.2325 | 0.0524 | 0.0443 |
| S&P (p=0.003) | 0.3124 | 0.2558 | 0.0728 | 0.2669 | 0.0631 | 0.0741 | 0.0637 | 0.0496 |

*(d)* BRISQUE

| Post-processing | StarGAN | StyleCLIP | DiffAE | SimSwap | pSp-mix | BlendFace | DiffSwap | DiffFace |
|---|---|---|---|---|---|---|---|---|
| No Post-processing | 29.6181 | 21.0689 | 12.5174 | 1.2399 | 6.2846 | 2.1080 | 3.5432 | 1.7217 |
| Blur ($\sigma$=1) | 29.6181 | 21.0695 | 12.5174 | 1.2399 | 6.2941 | 2.1131 | -4.9171 | 1.7217 |
| Blur ($\sigma$=3) | 23.7931 | 40.2378 | 16.3093 | 1.2516 | 7.1598 | 2.4618 | -4.7663 | 1.7217 |
| JPEG (Q=70) | 17.0328 | 12.7457 | 12.7115 | 1.3822 | 6.6151 | 2.5316 | -5.0345 | 1.7217 |
| JPEG (Q=90) | 17.8007 | 13.8513 | 11.6284 | 1.3743 | 6.7212 | 2.4979 | -4.8489 | 1.7217 |
| Noise ($\sigma$=0.001) | 29.5659 | 20.9855 | 11.6774 | 1.2426 | 6.3005 | 2.1284 | -4.6537 | 1.7217 |
| Noise ($\sigma$=0.003) | 28.7932 | 20.6193 | 13.5594 | 1.2438 | 6.3168 | 2.1217 | -4.2948 | 1.7217 |
| S&P (p=0.001) | 22.8764 | 18.2544 | 47.0265 | 1.2638 | 6.5509 | 2.3390 | -4.7678 | 1.7217 |
| S&P (p=0.003) | 30.5374 | 15.5473 | 55.8376 | 1.2535 | 7.0424 | 2.3709 | -4.8391 | 1.7217 |

*Table 16.* Quantitative robustness evaluation of Latent Attack under various post-processing transformations.

*(a)* MSE

| Post-processing | StarGAN | StyleCLIP | DiffAE | SimSwap | pSp-mix | BlendFace | DiffSwap | DiffFace |
|---|---|---|---|---|---|---|---|---|
| No Post-processing | 0.0958 | 0.2117 | 0.0619 | 0.0123 | 0.0830 | 0.0040 | 0.0097 | 0.0142 |
| Blur ($\sigma$=1) | 0.0958 | 0.2117 | 0.0618 | 0.0123 | 0.0830 | 0.0040 | 0.0097 | 0.0112 |
| Blur ($\sigma$=3) | 0.0067 | 0.0403 | 0.0178 | 0.0006 | 0.0341 | 0.0025 | 0.0097 | 0.0071 |
| JPEG (Q=70) | 0.0036 | 0.0470 | 0.0050 | 0.0007 | 0.0406 | 0.0011 | 0.0097 | 0.0073 |
| JPEG (Q=90) | 0.0157 | 0.0884 | 0.0089 | 0.0058 | 0.0607 | 0.0018 | 0.0097 | 0.0098 |
| Noise ($\sigma$=0.001) | 0.0958 | 0.2111 | 0.0619 | 0.0123 | 0.0830 | 0.0040 | 0.0097 | 0.0142 |
| Noise ($\sigma$=0.003) | 0.0958 | 0.2108 | 0.0620 | 0.0123 | 0.0829 | 0.0040 | 0.0097 | 0.0126 |
| S&P (p=0.001) | 0.0996 | 0.1677 | 0.0453 | 0.0119 | 0.0773 | 0.0038 | 0.0097 | 0.0140 |
| S&P (p=0.003) | 0.1069 | 0.1206 | 0.0211 | 0.0111 | 0.0690 | 0.0034 | 0.0097 | 0.0135 |

*(b)* LPIPS

| Post-processing | StarGAN | StyleCLIP | DiffAE | SimSwap | pSp-mix | BlendFace | DiffSwap | DiffFace |
|---|---|---|---|---|---|---|---|---|
| No Post-processing | 0.4368 | 0.6439 | 0.4598 | 0.0653 | 0.2118 | 0.0151 | 0.0693 | 0.0528 |
| Blur ($\sigma$=1) | 0.4368 | 0.6439 | 0.4598 | 0.0653 | 0.2118 | 0.0151 | 0.0693 | 0.0462 |
| Blur ($\sigma$=3) | 0.0977 | 0.3922 | 0.2018 | 0.0055 | 0.1154 | 0.0103 | 0.0693 | 0.0317 |
| JPEG (Q=70) | 0.0624 | 0.4322 | 0.0640 | 0.0064 | 0.1295 | 0.0051 | 0.0693 | 0.0314 |
| JPEG (Q=90) | 0.1072 | 0.5679 | 0.1151 | 0.0374 | 0.1686 | 0.0079 | 0.0693 | 0.0420 |
| Noise ($\sigma$=0.001) | 0.4368 | 0.6439 | 0.4597 | 0.0653 | 0.2118 | 0.0151 | 0.0693 | 0.0528 |
| Noise ($\sigma$=0.003) | 0.4370 | 0.6438 | 0.4593 | 0.0652 | 0.2114 | 0.0151 | 0.0693 | 0.0513 |
| S&P (p=0.001) | 0.4540 | 0.6382 | 0.4507 | 0.0640 | 0.2017 | 0.0146 | 0.0693 | 0.0528 |
| S&P (p=0.003) | 0.4786 | 0.6252 | 0.3099 | 0.0615 | 0.1872 | 0.0134 | 0.0693 | 0.0528 |

*(c)* CIDR

| Post-processing | StarGAN | StyleCLIP | DiffAE | SimSwap | pSp-mix | BlendFace | DiffSwap | DiffFace |
|---|---|---|---|---|---|---|---|---|
| No Post-processing | 0.1713 | 0.5852 | 0.0117 | 0.6140 | 0.2070 | 0.5364 | 0.2674 | 0.6911 |
| Blur ($\sigma$=1) | 0.1713 | 0.5852 | 0.0117 | 0.6140 | 0.2070 | 0.5364 | 0.2437 | 0.5457 |
| Blur ($\sigma$=3) | 0.0428 | 0.3360 | 0.1253 | 0.0337 | 0.1079 | 0.3825 | 0.0513 | 0.0611 |
| JPEG (Q=70) | 0.0266 | 0.3694 | 0.0641 | 0.0471 | 0.1250 | 0.2012 | 0.1021 | 0.0878 |
| JPEG (Q=90) | 0.0402 | 0.4821 | 0.1302 | 0.3962 | 0.1660 | 0.3213 | 0.2173 | 0.4416 |
| Noise ($\sigma$=0.001) | 0.1708 | 0.5852 | 0.0117 | 0.6140 | 0.2069 | 0.5364 | 0.2686 | 0.6909 |
| Noise ($\sigma$=0.003) | 0.1709 | 0.5851 | 0.0113 | 0.6138 | 0.2068 | 0.5362 | 0.2588 | 0.6379 |
| S&P (p=0.001) | 0.1753 | 0.5773 | 0.0174 | 0.6048 | 0.2007 | 0.5243 | 0.2617 | 0.6864 |
| S&P (p=0.003) | 0.1891 | 0.5583 | 0.1433 | 0.5881 | 0.1827 | 0.4908 | 0.2546 | 0.6645 |

*(d)* BRISQUE

| Post-processing | StarGAN | StyleCLIP | DiffAE | SimSwap | pSp-mix | BlendFace | DiffSwap | DiffFace |
|---|---|---|---|---|---|---|---|---|
| No Post-processing | 37.0453 | 12.2354 | 28.0547 | 0.8187 | 5.0783 | 2.3089 | 3.5636 | 1.3969 |
| Blur ($\sigma$=1) | 37.0473 | 12.2369 | 28.0543 | 0.8187 | 5.0929 | 2.3143 | -4.9853 | 1.3969 |
| Blur ($\sigma$=3) | 25.8909 | 20.4972 | 19.7663 | 1.2985 | 6.5402 | 2.2945 | -4.8327 | 1.3969 |
| JPEG (Q=70) | 18.3917 | 10.4782 | 18.4249 | 1.3436 | 5.5357 | 2.3985 | -4.8356 | 1.3969 |
| JPEG (Q=90) | 17.9137 | 11.1897 | 22.1910 | 1.2787 | 5.0502 | 2.2447 | -4.6161 | 1.3969 |
| Noise ($\sigma$=0.001) | 37.0622 | 12.2373 | 28.0319 | 0.8243 | 5.0980 | 2.3101 | -5.2181 | 1.3969 |
| Noise ($\sigma$=0.003) | 37.0252 | 12.2361 | 28.1310 | 0.8229 | 5.1026 | 2.3068 | -4.9371 | 1.3969 |
| S&P (p=0.001) | 37.3004 | 12.0137 | 40.3490 | 0.8119 | 5.2966 | 2.3233 | -5.0029 | 1.3969 |
| S&P (p=0.003) | 37.9511 | 12.4351 | 59.4013 | 0.9720 | 5.7272 | 2.3136 | -4.7754 | 1.3969 |

*Table 17.* Quantitative robustness evaluation of SCOL under various post-processing transformations.

*(a)* MSE

| Post-processing | StarGAN | StyleCLIP | DiffAE | SimSwap | pSp-mix | BlendFace | DiffSwap | DiffFace |
|---|---|---|---|---|---|---|---|---|
| No Post-processing | 0.0590 | 0.0481 | 0.0546 | 0.0050 | 0.0463 | 0.0013 | 0.005 | 0.0101 |
| Blur ($\sigma$=1) | 0.0590 | 0.0481 | 0.0528 | 0.0050 | 0.0463 | 0.0013 | 0.005 | 0.0095 |
| Blur ($\sigma$=3) | 0.0562 | 0.0490 | 0.0543 | 0.0051 | 0.0470 | 0.0013 | 0.005 | 0.0096 |
| JPEG (Q=70) | 0.0575 | 0.0476 | 0.0539 | 0.0050 | 0.0449 | 0.0013 | 0.005 | 0.0097 |
| JPEG (Q=90) | 0.0571 | 0.0481 | 0.0538 | 0.0050 | 0.0459 | 0.0013 | 0.005 | 0.0096 |
| Noise ($\sigma$=0.001) | 0.0591 | 0.0482 | 0.0526 | 0.0050 | 0.0463 | 0.0013 | 0.005 | 0.0097 |
| Noise ($\sigma$=0.003) | 0.0593 | 0.0482 | 0.0519 | 0.0050 | 0.0464 | 0.0013 | 0.005 | 0.0099 |
| S&P (p=0.001) | 0.0892 | 0.0493 | 0.0541 | 0.0052 | 0.0470 | 0.0013 | 0.005 | 0.0098 |
| S&P (p=0.003) | 0.1112 | 0.0526 | 0.0563 | 0.0054 | 0.0496 | 0.0014 | 0.005 | 0.0098 |

*(b)* LPIPS

| Post-processing | StarGAN | StyleCLIP | DiffAE | SimSwap | pSp-mix | BlendFace | DiffSwap | DiffFace |
|---|---|---|---|---|---|---|---|---|
| No Post-processing | 0.2317 | 0.3344 | 0.1759 | 0.0315 | 0.1058 | 0.0059 | 0.0309 | 0.0395 |
| Blur ($\sigma$=1) | 0.2317 | 0.3344 | 0.3506 | 0.0315 | 0.1058 | 0.0059 | 0.0309 | 0.0393 |
| Blur ($\sigma$=3) | 0.2260 | 0.3577 | 0.4508 | 0.0320 | 0.1081 | 0.0059 | 0.0309 | 0.0395 |
| JPEG (Q=70) | 0.2133 | 0.2695 | 0.3522 | 0.0312 | 0.1052 | 0.0062 | 0.0309 | 0.0390 |
| JPEG (Q=90) | 0.2162 | 0.3205 | 0.3616 | 0.0313 | 0.1055 | 0.0059 | 0.0309 | 0.0387 |
| Noise ($\sigma$=0.001) | 0.2322 | 0.3345 | 0.3467 | 0.0315 | 0.1058 | 0.0059 | 0.0309 | 0.0395 |
| Noise ($\sigma$=0.003) | 0.2369 | 0.3357 | 0.3176 | 0.0316 | 0.1059 | 0.0060 | 0.0309 | 0.0395 |
| S&P (p=0.001) | 0.4084 | 0.3486 | 0.2926 | 0.0325 | 0.1129 | 0.0061 | 0.0309 | 0.0395 |
| S&P (p=0.003) | 0.4941 | 0.3843 | 0.2882 | 0.0333 | 0.1242 | 0.0062 | 0.0309 | 0.0395 |

*(c)* CIDR

| Post-processing | StarGAN | StyleCLIP | DiffAE | SimSwap | pSp-mix | BlendFace | DiffSwap | DiffFace |
|---|---|---|---|---|---|---|---|---|
| No Post-processing | 0.2391 | 0.3939 | 0.3769 | 0.3519 | 0.1679 | 0.2293 | 0.1139 | 0.4115 |
| Blur ($\sigma$=1) | 0.2391 | 0.3939 | 0.3869 | 0.3519 | 0.1679 | 0.2293 | 0.1251 | 0.4022 |
| Blur ($\sigma$=3) | 0.2391 | 0.4076 | 0.4004 | 0.3527 | 0.1733 | 0.2345 | 0.1161 | 0.4072 |
| JPEG (Q=70) | 0.2382 | 0.3987 | 0.4054 | 0.3519 | 0.1707 | 0.2410 | 0.1315 | 0.4022 |
| JPEG (Q=90) | 0.2318 | 0.3957 | 0.4020 | 0.3505 | 0.1688 | 0.2318 | 0.1244 | 0.3993 |
| Noise ($\sigma$=0.001) | 0.2399 | 0.3939 | 0.3870 | 0.3520 | 0.1679 | 0.2293 | 0.1187 | 0.4068 |
| Noise ($\sigma$=0.003) | 0.2419 | 0.3940 | 0.3892 | 0.3521 | 0.1680 | 0.2293 | 0.1537 | 0.4104 |
| S&P (p=0.001) | 0.2556 | 0.3972 | 0.4201 | 0.3552 | 0.1680 | 0.2278 | 0.1325 | 0.4044 |
| S&P (p=0.003) | 0.2684 | 0.4081 | 0.4324 | 0.3602 | 0.1722 | 0.2326 | 0.1406 | 0.3944 |

*(d)* BRISQUE

| Post-processing | StarGAN | StyleCLIP | DiffAE | SimSwap | pSp-mix | BlendFace | DiffSwap | DiffFace |
|---|---|---|---|---|---|---|---|---|
| No Post-processing | 21.8716 | 6.5536 | 38.4324 | 1.7359 | 5.9274 | 2.4045 | 3.8448 | 2.0021 |
| Blur ($\sigma$=1) | 21.8667 | 6.5527 | 63.4258 | 1.7359 | 5.9304 | 2.4065 | -4.8641 | 2.0021 |
| Blur ($\sigma$=3) | 27.6213 | 34.1819 | 4.2680 | 1.6475 | 6.4855 | 2.4577 | -5.0424 | 2.0021 |
| JPEG (Q=70) | 22.9816 | 12.0078 | 18.6510 | 1.7836 | 6.5595 | 2.5256 | -4.8158 | 2.0021 |
| JPEG (Q=90) | 20.5043 | 6.5675 | 13.8583 | 1.7642 | 5.9662 | 2.4475 | -4.9022 | 2.0021 |
| Noise ($\sigma$=0.001) | 21.8907 | 6.5516 | 62.7166 | 1.7200 | 5.9241 | 2.4163 | -4.8007 | 1.9668 |
| Noise ($\sigma$=0.003) | 22.1237 | 6.5594 | 58.1992 | 1.7190 | 5.8789 | 2.4151 | -4.5728 | 1.8562 |
| S&P (p=0.001) | 26.5708 | 6.9881 | 34.9108 | 1.6219 | 6.3522 | 2.4385 | -4.6815 | 2.0021 |
| S&P (p=0.003) | 32.1346 | 8.6657 | 67.2802 | 1.6968 | 6.8238 | 2.4335 | -4.5129 | 1.9756 |

*Table 18.* Quantitative robustness evaluation of NullSwap under various post-processing transformations.

*(a)* MSE

| Post-processing | StarGAN | StyleCLIP | DiffAE | SimSwap | pSp-mix | BlendFace | DiffSwap | DiffFace |
|---|---|---|---|---|---|---|---|---|
| No Post-processing | 0.0054 | 0.0108 | 0.0053 | 0.0060 | 0.0044 | 0.0018 | 0.0053 | 0.0103 |
| Blur ($\sigma$=1) | 0.0054 | 0.0108 | 0.0154 | 0.0060 | 0.0044 | 0.0018 | 0.0053 | 0.0100 |
| Blur ($\sigma$=3) | 0.0060 | 0.0156 | 0.0165 | 0.0061 | 0.0071 | 0.0018 | 0.0053 | 0.0103 |
| JPEG (Q=70) | 0.0046 | 0.0114 | 0.0165 | 0.0060 | 0.0046 | 0.0018 | 0.0053 | 0.0103 |
| JPEG (Q=90) | 0.0042 | 0.0108 | 0.0163 | 0.0060 | 0.0044 | 0.0018 | 0.0053 | 0.0103 |
| Noise ($\sigma$=0.001) | 0.0055 | 0.0108 | 0.0154 | 0.0060 | 0.0044 | 0.0017 | 0.0053 | 0.0103 |
| Noise ($\sigma$=0.003) | 0.0062 | 0.0108 | 0.0153 | 0.0060 | 0.0044 | 0.0017 | 0.0053 | 0.0103 |
| S&P (p=0.001) | 0.0526 | 0.0124 | 0.0175 | 0.0061 | 0.0069 | 0.0017 | 0.0053 | 0.0103 |
| S&P (p=0.003) | 0.0816 | 0.0161 | 0.0202 | 0.0061 | 0.0101 | 0.0018 | 0.0053 | 0.0103 |

*(b)* LPIPS

| Post-processing | StarGAN | StyleCLIP | DiffAE | SimSwap | pSp-mix | BlendFace | DiffSwap | DiffFace |
|---|---|---|---|---|---|---|---|---|
| No Post-processing | 0.0340 | 0.1046 | 0.0353 | 0.0358 | 0.0254 | 0.0080 | 0.0316 | 0.0410 |
| Blur ($\sigma$=1) | 0.0340 | 0.1046 | 0.2684 | 0.0358 | 0.0254 | 0.0080 | 0.0316 | 0.0403 |
| Blur ($\sigma$=3) | 0.0647 | 0.2558 | 0.2753 | 0.0364 | 0.0352 | 0.0083 | 0.0316 | 0.0410 |
| JPEG (Q=70) | 0.0314 | 0.1144 | 0.2388 | 0.0358 | 0.0261 | 0.0080 | 0.0316 | 0.0410 |
| JPEG (Q=90) | 0.0284 | 0.1049 | 0.2511 | 0.0358 | 0.0255 | 0.0080 | 0.0316 | 0.0410 |
| Noise ($\sigma$=0.001) | 0.0347 | 0.1047 | 0.2690 | 0.0358 | 0.0254 | 0.0080 | 0.0316 | 0.0410 |
| Noise ($\sigma$=0.003) | 0.0424 | 0.1051 | 0.2686 | 0.0358 | 0.0254 | 0.0077 | 0.0316 | 0.0410 |
| S&P (p=0.001) | 0.2871 | 0.1286 | 0.2133 | 0.0361 | 0.0410 | 0.0078 | 0.0316 | 0.0410 |
| S&P (p=0.003) | 0.3953 | 0.2088 | 0.1904 | 0.0366 | 0.0575 | 0.0079 | 0.0316 | 0.0410 |

*(c)* CIDR

| Post-processing | StarGAN | StyleCLIP | DiffAE | SimSwap | pSp-mix | BlendFace | DiffSwap | DiffFace |
|---|---|---|---|---|---|---|---|---|
| No Post-processing | 0.2792 | 0.5677 | 0.5028 | 0.5114 | 0.2856 | 0.3983 | 0.2131 | 0.5627 |
| Blur ($\sigma$=1) | 0.2792 | 0.5677 | 0.4739 | 0.5114 | 0.2856 | 0.3983 | 0.2076 | 0.5659 |
| Blur ($\sigma$=3) | 0.2811 | 0.5726 | 0.4748 | 0.5120 | 0.2909 | 0.3976 | 0.2065 | 0.5671 |
| JPEG (Q=70) | 0.2807 | 0.5684 | 0.4755 | 0.5116 | 0.2864 | 0.3990 | 0.2079 | 0.5661 |
| JPEG (Q=90) | 0.2764 | 0.5678 | 0.4746 | 0.5112 | 0.2857 | 0.3986 | 0.2128 | 0.5610 |
| Noise ($\sigma$=0.001) | 0.2796 | 0.5676 | 0.4741 | 0.5114 | 0.2856 | 0.3989 | 0.2049 | 0.5666 |
| Noise ($\sigma$=0.003) | 0.2811 | 0.5675 | 0.4739 | 0.5115 | 0.2856 | 0.3989 | 0.2264 | 0.5659 |
| S&P (p=0.001) | 0.2910 | 0.5680 | 0.4865 | 0.5122 | 0.2856 | 0.3987 | 0.2119 | 0.5660 |
| S&P (p=0.003) | 0.2951 | 0.5696 | 0.4937 | 0.5100 | 0.2871 | 0.4015 | 0.1965 | 0.5703 |

*(d)* BRISQUE

| Post-processing | StarGAN | StyleCLIP | DiffAE | SimSwap | pSp-mix | BlendFace | DiffSwap | DiffFace |
|---|---|---|---|---|---|---|---|---|
| No Post-processing | 18.3381 | 12.8330 | 28.2317 | 1.0041 | 5.6176 | 2.4165 | 3.7283 | 1.8001 |
| Blur ($\sigma$=1) | 18.3371 | 12.8382 | 23.4864 | 1.0041 | 5.6122 | 2.4151 | -4.6729 | 1.8001 |
| Blur ($\sigma$=3) | 25.8780 | 39.9484 | 14.1503 | 1.0441 | 5.7040 | 2.4535 | -5.1286 | 1.7832 |
| JPEG (Q=70) | 17.1786 | 10.0821 | 12.9199 | 0.9735 | 5.5468 | 2.4049 | -4.7219 | 1.8001 |
| JPEG (Q=90) | 18.4471 | 12.9163 | 10.1845 | 1.0331 | 5.6290 | 2.4007 | -5.0886 | 1.6849 |
| Noise ($\sigma$=0.001) | 18.4011 | 12.7664 | 24.1845 | 0.9946 | 5.6112 | 2.4029 | -4.9700 | 1.6655 |
| Noise ($\sigma$=0.003) | 18.7338 | 12.3520 | 28.1653 | 0.9987 | 5.6128 | 2.3489 | -4.4995 | 1.7538 |
| S&P (p=0.001) | 23.5852 | 10.2667 | 44.5269 | 0.9955 | 6.0849 | 2.3170 | -5.0703 | 1.8001 |
| S&P (p=0.003) | 30.7863 | 8.8471 | 57.5830 | 1.0936 | 6.4697 | 2.2831 | -4.9616 | 1.8001 |

# D. Transferability

This appendix provides comprehensive transferability results across all evaluated generators and defense methods. While Figure 4 visualizes heatmaps based on MSE and CIDR, we additionally report results for other evaluation metrics to enable a more complete and transparent analysis.

Source Generators denote the generator used to generate perturbations, and Target Generators denote the generator used for evaluation. In this section, we consider a black-box transfer setting, where the source and target generators are different. Specifically, perturbations generated on a given source generator are applied to different target generators, and their effects are evaluated by measuring discrepancies between the generated outputs $G(x_{\mathrm{adv}})$ and the corresponding clean outputs $G(x)$. All transferability experiments are conducted without access to the architecture or parameters of the target generator.

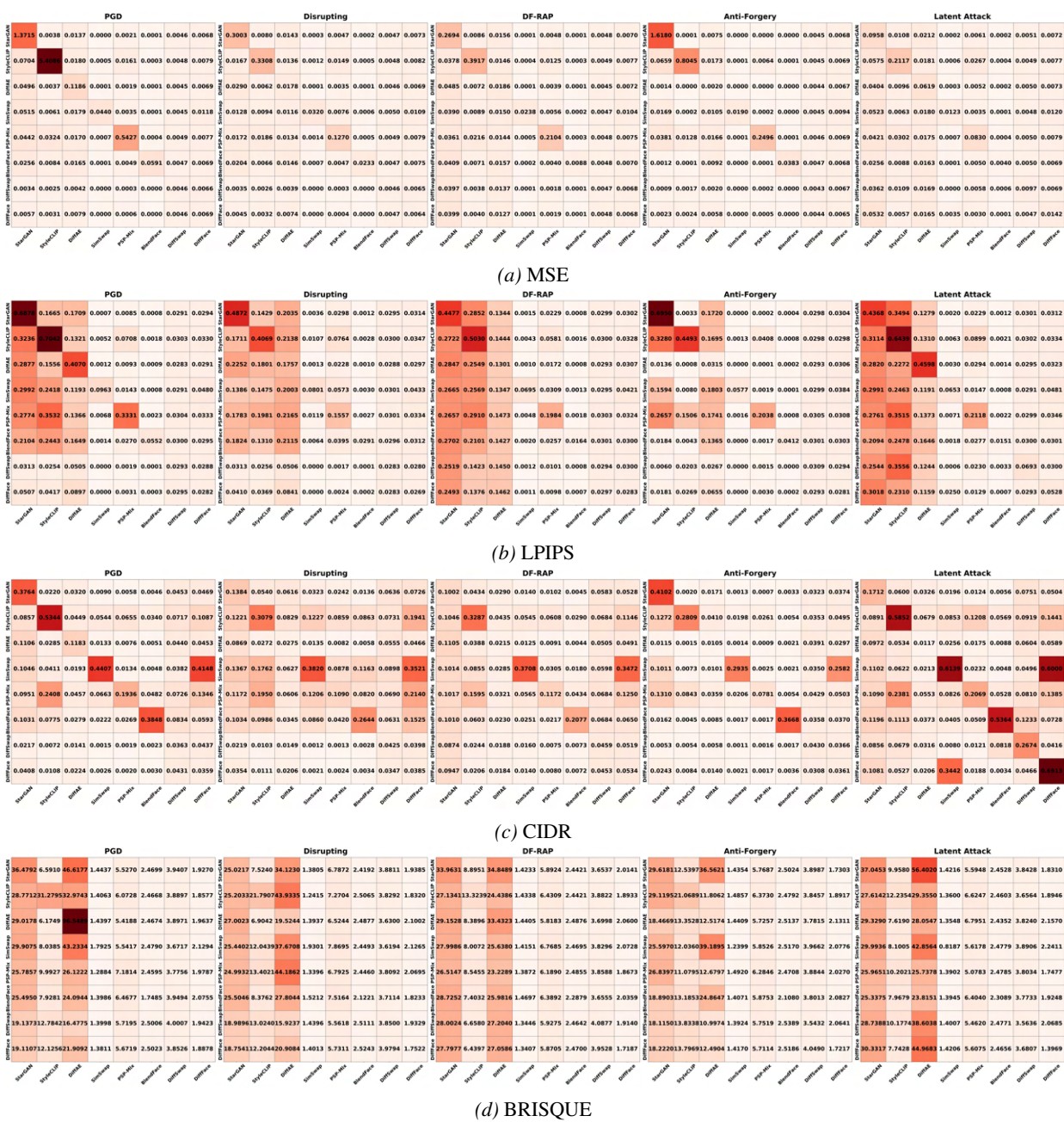

*(a)* MSE

*(b)* LPIPS

*(c)* CIDR

*(d)* BRISQUE

*Figure 12.* Transferability of proactive defenses across deepfake generators.

# E. Correlation between Evaluation Metrics

We analyze the pairwise Spearman correlations among representative metrics across various generators and defenses, as summarized in Figure 13. First, we observe strong redundancy between pixel-level metrics (MSE and PSNR). However, the relationship between pixel-level and perceptual fidelity varies by generator architecture. Face swap models exhibit consistent degradation patterns across all fidelity metrics, whereas attribute manipulation models show high variance, indicating that defenses can degrade pixel-level fidelity and perceptual quality independently.

Second, identity-based metrics generally demonstrate weaker correlations with synthesis fidelity than the inter-correlation among fidelity metrics themselves. This observation necessitates the simultaneous use of both metric types for a holistic evaluation. Notably, CIDR exhibits a stronger correlation with ID Loss than with synthesis metrics, maintaining a low correlation with synthesis metrics. This independence suggests that CIDR successfully isolates defense-induced identity disruption by mitigating the generator-specific biases inherent in ID Loss. Finally, BRISQUE shows minimal correlation with all other metrics, capturing an orthogonal dimension of image degradation.

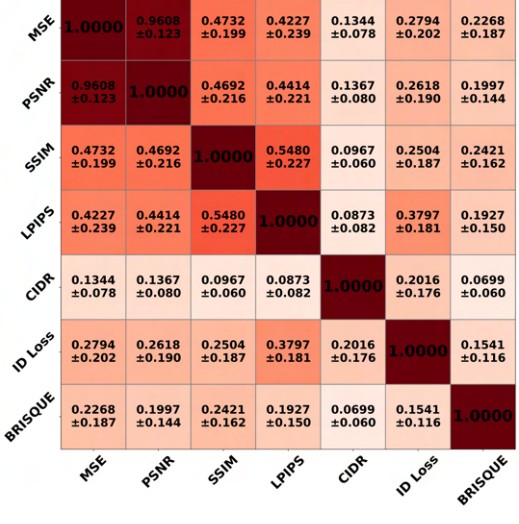

*(a)* Attribute Manipulation

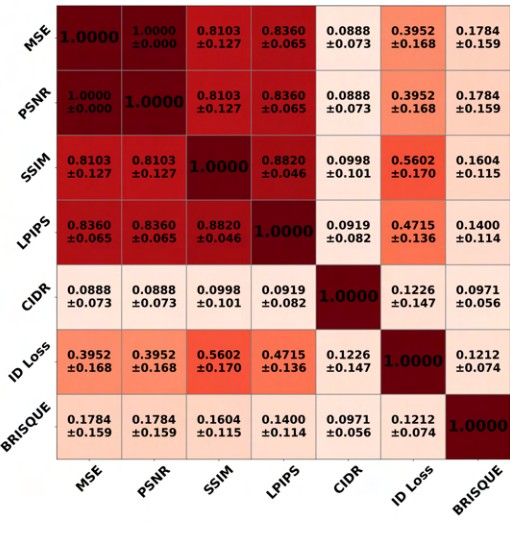

*(b)* Face Swap

*Figure 13.* Pairwise correlation patterns of evaluation metrics across proactive defenses. Rows and columns denote evaluation metrics, with cell values representing the average absolute Spearman correlation ($\pm$ standard deviation) across all defense methods. Diagonal elements are 1.0 (self-correlation). (a) shows attribute manipulation models, and (b) shows face swap models. Darker colors indicate stronger correlations.

