# OpenReview forum: "Proactive Defense Benchmark against Deepfake Generation"
_ICML.cc/2026/Conference — ICML 2026 regular_

### Official Review · Reviewer_5dnm · 2026-03-06

**Soundness:** 2
**Presentation:** 2
**Significance:** 3
**Originality:** 3
**Overall Recommendation:** 4
**Confidence:** 3

**Summary:**

The paper targets proactive defenses: pre-applied image perturbations intended to hinder downstream deepfake generation, and proposes a unified benchmark to evaluate disruption, robustness to post-processing, and cross-generator transferability across multiple face generators and defenses. It introduces normalized metrics (CIDR for identity calibration; TE/ROB ratios) and a rank-based aggregation (AvgRank) to compare methods. This is a promising research direction, but the current paper does not yet deliver comprehensive experiments, broader settings/datasets, or sufficiently new insights to justify acceptance.

**Compliance With Llm Reviewing Policy:**

Affirmed.

**Final Justification:**

Authors addressed some of my main concerns. I do believe this benchmark can be beneficial to the community and raise my score to 4.

**Key Questions For Authors:**

* How often do the clipping rules affect the results in practice? ROB and TE are clipped at 1, and CIDR is clipped at 0. What fraction of samples or averages actually hit these clipping boundaries, and could the clipping materially alter method ranking or compress meaningful differences between defenses?

* How stable is CIDR when the baseline generator identity distortion varies a lot across generators? CIDR normalizes by the generator’s own identity loss $L_{ID}(x,G(x))$. Can the authors report the distribution of this baseline term across generators and explain whether CIDR becomes unstable or overly favorable when the denominator is very small or very large?

**Limitations:**

yes

**Strengths And Weaknesses:**

### Strengths
* The standardized protocol across disruption/robustness/transfer is directionally valuable.
* Reasonable metric design intuition: CIDR explicitly corrects for generator-induced identity distortion confounds; TE/ROB are simple normalized diagnostics that can improve comparability across generators/transformations.
* The benchmark evaluates multiple face editing and face-swap generators and several defenses under one pipeline, reducing cherry-picking.

### Weaknesses
* Primary evaluation is on CelebA-HQ at 256×256, which is too narrow to claim general conclusions about proactive defenses. It would benefit to add at least two additional datasets (e.g., FFHQ + an in-the-wild set)
* The set of evaluated models (StarGAN/StyleCLIP/DiffAE; SimSwap/pSp-mix/BlendFace) omits modern diffusion-based identity editing/swapping stacks that dominate current practice. For a novel benchmark, it is necessary to include various models, including state-of-the-art diffusion-based ones.
* Robustness setting lacks realism: robustness largely uses simple transformations; real sharing/removal pipelines involve chained operations (resize→crop→multi-stage recompress, screenshots, colorspace/gamma, enhancement/denoise).
* Insight novelty is limited: main findings (generator dependence, robustness–disruption trade-off, weak transfer, etc.) largely restate known robustness/transfer phenomena; the paper does not yet extract mechanism-level or design-rule insights (what perturbation properties predict transfer/robustness, which generator components are sensitive, etc.).
* Some figures are too small to be easily understand (for example, Figures 4, 5, 10)

---

> ### Author Rebuttal · Authors · 2026-03-30
>
> We appreciate your careful and detailed feedback. We respond to your comments below.
>
> **[W1] Evaluation on Additional Datasets:** Please see the reply to Reviewer *qeeX* [W1] due to the space limit.
>
> **[W2] Scalability to Diffusion-Based Models:** Please see the reply to Reviewer *fZU9* [W2][Q2][Q3] due to the space limit.
>
> **[W3] Realism of Robustness Evaluation:** We acknowledge that real-world pipelines involve chained transformations and agree this is an important future direction. However, our evaluation deliberately isolates individual transformations to attribute degradation to specific causes. Even a single mild transformation (e.g., JPEG Q=70) reduces PGD’s MSE on StarGAN from 1.3715 to 0.0021, indicating near-complete collapse. This suggests that chained evaluation offers limited additional diagnostic value when methods already fail under a single transformation. Therefore, improving performance on single transformations, as guided by our benchmark, is a crucial first step toward more realistic evaluation settings.
>
> **[W4] Novelty of Insights Beyond Known Phenomena:** We partially agree that some phenomena were hinted at in prior work. Although robustness under adversarial perturbations and transferability are well-established topics in the broader adversarial machine learning literature, prior work in proactive deepfake defense has only explored them in a fragmented and limited manner. For instance, Anti-Forgery evaluated robustness only on GAN-based generators. Similarly, Disrupting tested robustness only under blur transformations, and LEAT evaluated transferability only on a limited set of GAN models. As shown in Table 4(c), no prior work jointly evaluates all four metric dimensions; our benchmark fills this gap by providing systematic and cross-comparable evaluation. Beyond consolidation, we provide the following insights: (i) fidelity and identity metrics capture independent performance axes, meaning single-category evaluation yields misleading conclusions; and (ii) CIDR corrects generator-induced identity bias in standard ID Loss, resolving the qualitative–quantitative mismatch in Figure 6.
>
> **[Q1] Impact of Clipping on Method Ranking:** For CIDR, ROB, and TE, we apply clipping to handle edge cases: a negative CIDR indicates identity becoming closer to the original after defense, a ROB ratio > 1.0 means disruption increased after post-processing, and a TE ratio > 1.0 indicates rare cases where black-box transfer exceeds white-box performance. For ROB, clipping to 1.0 reflects a conservative design choice, treating any robustness exceeding the original performance as 100%.
>
> |Metrics|Ratio|
> |-|-|
> |ROB|36.76%|
> |TE|14.17%|
> |CIDR|10.57%|
> ---
> - **CIDR:** Negative values occur in 10.57% of cases, concentrated in generators with large intrinsic identity distortion such as DiffAE. We recomputed AvgRank without clipping: rankings remained unchanged in 5 out of 6 generators, with an average Kendall's τ of 0.984, confirming clipping does not affect method rankings.
> - **ROB:** Methods like SCOL (56.94%) and NullSwap (58.39%) are inherently weak, resulting in a small denominator. This leads to frequent clipped at 1.0. We recomputed ROB rankings without clipping, yielding a Kendall's τ of 0.857, confirming clipping has minimal impact on method rankings.
> - **TE:** Clipping is concentrated in the no-reference metric BRISQUE (37.41%), while rarely occurring in MSE (3.97%) and CIDR (6.13%).
>
> **[Q2] Stability of CIDR Across Generators:** We report the per-generator baseline identity distortion $L_{ID}(x, G(x))$ below.
>
> |Generator|Mean|Std|Min|Max|
> |-|-|-|-|-|
> |StarGAN|0.557|0.106|0.284|0.821|
> |StyleCLIP|0.317|0.070|0.156|0.519|
> |DiffAE|0.413|0.088|0.122|0.676|
> |SimSwap|0.397|0.088|0.196|0.695|
> |pSp-mix|0.540|0.106|0.311|0.837|
> |BlendFace|0.521|0.092|0.331|0.774|
> ---
> The variation in $L_{ID}(x, G(x))$ across generators motivates CIDR. Conventional ID Loss ignores this baseline difference, making cross-generator comparisons unfair. For instance, directly comparing BlendFace's baseline of 0.521 against StyleCLIP's 0.317 would overestimate defense performance against BlendFace. CIDR normalizes for this generator-specific bias, isolating only the defense's additional contribution. Regarding edge cases:
> - **When $L_{ID}(x, G(x))$ is small:** In this case, even a slight identity change induced by the defense results in a high CIDR value. However, this means that the defense successfully induces identity changes in a situation where the generator itself barely alters identity, indicating sensitivity under low baseline distortion.
> - **When $L_{ID}(x, G(x))$ is large:** In this case, CIDR becomes relatively smaller, indicating that the additional contribution of the defense is limited because the generator itself already introduces substantial identity changes.
> - **Numerical stability:** The minimum observed $L_{ID}(x, G(x))$ is 0.122 (DiffAE); near-zero denominator instability is not observed in practice.

---

> > ### Author Rebuttal · Reviewer_5dnm · 2026-04-02
> >
> > Thank you for the detailed rebuttal; the added results and analysis strengthen the paper and address some of my technical concerns. However, my main concern remains that the benchmark still feels too narrow in scope and setting for this venue, and the paper still does not extract sufficiently strong mechanism-level insights beyond a unified empirical comparison. I therefore appreciate the improvements and keep my current recommendation.

---

> > > ### Author Response · Authors · 2026-04-07
> > >
> > > We appreciate the reviewer’s acknowledgement that our response and additional results have addressed the technical concerns raised. To address the remaining concerns regarding the scope and mechanism-level insights, we would like to offer the following additional perspectives.
> > >
> > > **Regarding the concern about the narrow scope,** we would like to clarify that our benchmark is designed to reflect the current landscape of proactive defenses, which inherently lacks standardized evaluation protocols due to the generative nature of disruption tasks. Unlike deepfake detection benchmarks that rely on well-defined metrics such as detection accuracy, disruption benchmarks face fundamental challenges in evaluation because the outputs of generative models are diverse and difficult to assess with a single unified metric. To address this, our primary goal is to establish a principled and unified evaluation protocol within an extensible framework, rather than exhaustively enumerating all possible methods. While our current evaluation includes a representative subset of proactive defenses, the framework is explicitly modular, allowing new methods and generative models to be seamlessly integrated into the pipeline without modifying its core design. We believe this design choice is essential for a rapidly evolving area where new defenses and models continue to emerge.
> > >
> > > **Regarding mechanism-level insights**, while the initial submission focused on unified empirical comparison, we conducted additional analysis during the rebuttal period to better understand architectural factors, particularly in modern diffusion-based models.
> > >
> > > Specifically, we investigate why optimization-based methods (e.g., PGD, Disrupting, DF-RAP), which maximize pixel-level perturbations via $L(G(x), G(x_{adv}))$, show significantly reduced effectiveness on diffusion-based models (e.g., DiffSwap, DiffFace). To probe this, we vary the number of sampling steps in DiffAE from $T \in {10, 20, 50, 100, 200}$ and measure the disruption effectiveness of PGD. As shown below, increasing the number of sampling steps leads to a consistent degradation in disruption metrics (e.g., MSE: 0.1200 $\rightarrow$ 0.0193, LPIPS: 0.4112 $\rightarrow$ 0.2791, BRISQUE: 96.92 $\rightarrow$ 59.45), indicating that the iterative denoising process progressively attenuates pixel-level perturbations. This explains why methods that directly optimize $L(G(x), G(x_{adv}))$ become less effective in diffusion-based architectures.
> > >
> > > ### Disruption effectiveness of PGD on DiffAE under varying sampling steps
> > > |Metric|T10|T20|T50|T100|T200|
> > > |:-:|:-:|:-:|:-:|:-:|:-:|
> > > |MSE↑|0.1200|0.0376|0.0224|0.0206|0.0193|
> > > |LPIPS↑|0.4112|0.3988|0.3228|0.3024|0.2791|
> > > |CIDR↑|0.1464|0.0630|0.0360|0.0493|0.0414|
> > > |BRISQUE↑|96.9197|82.1362|65.8337|61.4653|59.4503|
> > > ---
> > >
> > > On the other hand, black-box methods such as SCOL and NullSwap maintain disruption performance on these diffusion-based models (*fZU9* [W2][Q2][Q3]), despite having no access to the target generator. Since diffusion-based models condition their generation process on the source image, disrupting this conditioning stage can effectively propagate disruption throughout the generation process, regardless of the number of sampling steps.
> > >
> > > We believe this analysis provides a concrete mechanism-level explanation for why different attack formulations interact differently with diffusion-based architectures. Importantly, this insight also offers practical guidance for designing more effective disruption methods tailored to modern diffusion models.

---

### Official Review · Reviewer_qeeX · 2026-03-12

**Soundness:** 3
**Presentation:** 4
**Significance:** 2
**Originality:** 3
**Overall Recommendation:** 4
**Confidence:** 4

**Summary:**

This paper proposes the first benchmarking on evaluating proactive defense methods against facial deepfakes over disruption, robustness, and transferability. Overall, I found the evaluation aspects proposed by this benchmark well align with real-world needs. The takeaway insights from this paper are clear and provide guidance for future development of proactive defense methods. On the other side, I found the theme to be quite concentrated at a specific group of generation models - face swap and facial attribute generation. As image deepfakes are evolving into full-frame and multimodal, the insights here may not be transferrable to other domains. Overall, I think it is a solid benchmark work that can be quite insightful for the researchers who work on proactive defense against facial deepfakes. I am not very sure if this is a good fit for ICML though.

**Compliance With Llm Reviewing Policy:**

Affirmed.

**Final Justification:**

Authors addressed some of my concerns on the technical aspect. I do believe this benchmark work can be beneficial to the community  hence raise my score correspondingly.

**Key Questions For Authors:**

Please refer to my comments above.

**Limitations:**

yes

**Strengths And Weaknesses:**

- Soundness: I found the experiments and evaluations well support its motivations. As a result, the findings clearly demonstrate the pros and cons of different type of proactive methods and their shared weaknesses. This shows that the benchmark is carefully designed. I do think the comprehensiveness of this benchmark can be improved, e.g., adding more facial datasets, especially that the adopted face dataset is several years old now. It may sound cliche but for a benchmark to be widely used and acknowledged by the community, having one source dataset may limit its usability.
- Presentation: Authors did a good job outlining challenges and weakness of current proactive defense models, and further motivate the design of this benchmark. The key elements of this benchmark are well elaborated and support its motivation. Results demonstration is also clear.
- Significance: I do think the topic of this benchmark is somewhat limited. While deepfakes are going full-frame and multimodal, how much of the insight obtained here can be transferred to other domains is unknown. Furthermore, the limit of source facial datasets adopted by this benchmark may hinder its usability.
- Originality: This is the first benchmark to evaluate proactive defense methods against face deepfakes. The insights are clear and provide crucial guidance for researchers in this field on how to properly eval the newly developed models.

---

> ### Author Rebuttal · Authors · 2026-03-30
>
> Thank you for your insightful comments. We are glad that you found our benchmark effectively motivates its design and clearly exposes the strengths and limitations of existing proactive methods. We address your concerns below.
>
> **[W1] Evaluation on Additional Datasets:** As shown in the tables below, the relative ranking of defense methods remains largely consistent across different datasets (FFHQ [1] and VGGFace2-HQ [2]). This suggests that the performance ordering among defense methods is driven primarily by the inherent characteristics of each method rather than the dataset, supporting the generalizability of our benchmark's evaluation protocol across different experimental conditions. Due to space limit, we present results for two representative generators (StyleCLIP and SimSwap); however, we observed consistent trends across all other generator–defense combinations as well. Additionally, while Figure 1 was produced using the CelebA-HQ dataset, we confirmed that analogous patterns hold for the other datasets. Corresponding figures are available at [here](https://anonymous.4open.science/r/ICML2026_9676/image.png).
>
> ## StyleCLIP
> ||Metric|PGD|Disrupting|DF-RAP|Anti|Latent|SCOL|NullSwap|
> |:-:|:-:|:-:|:-:|:-:|:-:|:-:|:-:|:-:|
> |**FFHQ**|MSE↑|4.1773|0.2394|0.3327|0.5960|0.1806|0.0540|0.0107|
> ||LPIPS↑|0.6647|0.3772|0.4744|0.4515|0.6248|0.3198|0.1048|
> ||CIDR↑|0.5644|0.3450|0.3470|0.3143|0.6227|0.4226|0.5611|
> ||BRISQUE↑|28.0250|18.5784|12.0067|18.3265|12.7946|7.0981|11.3217|
> ||AvgRank↓|1.25|4.25|4.00|4.00|3.00|5.75|5.75|
> |**VGGFace2-HQ**|MSE↑|2.7490|0.2223|0.2908|0.4597|0.1876|0.0092|0.0108|
> ||LPIPS↑|0.6537|0.3401|0.4407|0.4166|0.5911|0.1941|0.0951|
> ||CIDR↑|0.3909|0.1854|0.2028|0.1996|0.4386|0.2193|0.3832|
> ||BRISQUE↑|24.1783|20.5644|11.6138|18.8123|12.2644|5.7525|15.0691|
> ||AvgRank↓|1.25|4.50|4.25|3.75|3.25|6.00|5.00|
> ---
> ## SimSwap
> ||Metric|PGD|Disrupting|DF-RAP|Anti|Latent|SCOL|NullSwap|
> |:-:|:-:|:-:|:-:|:-:|:-:|:-:|:-:|:-:|
> |**FFHQ**|MSE↑|0.0369|0.0279|0.0214|0.0169|0.0131|0.0053|0.0058|
> ||LPIPS↑|0.1151|0.0995|0.0892|0.0745|0.0885|0.0422|0.0435|
> ||CIDR↑|0.4160|0.3640|0.3388|0.2902|0.5810|0.3163|0.4597|
> ||BRISQUE↑|4.8177|5.1509|4.3073|3.4305|2.8553|1.5410|1.8451|
> ||AvgRank↓|1.75|2.25|3.50|5.00|3.75|6.75|5.00|
> |**VGGFace2-HQ**|MSE↑|0.0346|0.0241|0.0169|0.0129|0.0100|0.0029|0.0042|
> ||LPIPS↑|0.0701|0.0542|0.0476|0.0369|0.0500|0.0163|0.0229|
> ||CIDR↑|0.4183|0.3499|0.3321|0.2678|0.5860|0.2491|0.4913|
> ||BRISQUE↑|26.6891|26.4644|25.8577|25.6680|24.7518|25.6100|25.7300|
> ||AvgRank↓|1.50|2.50|3.75|5.00|4.00|6.75|4.50|
> ---
>
> **[W2] Scope of the Benchmark and Transferability to Other Domains:** We agree that deepfake technology is expanding beyond the facial domain. However, facial deepfakes remain the primary target of existing proactive defense research, yet lack a unified evaluation protocol, making a systematic benchmark in this area most urgent. Moreover, facial deepfakes directly enable serious harms such as identity theft and non-consensual imagery, providing strong practical justification.
>
> Furthermore, we believe that the core insights derived from our benchmark are not inherently limited to the facial domain. For instance, (i) the tendency of white-box optimization to lead to overfitting, and (ii) the trade-off between robustness and visual disruption, stem from fundamental properties of adversarial perturbation and are therefore likely to arise similarly in full-frame or multimodal settings. (iii) Our finding that fidelity metrics and identity metrics capture independent axes of defense performance, such that relying on only one category can yield conflicting interpretations, is also broadly applicable whenever generative model evaluation involves multiple competing objectives.
>
> We also note that the benchmark's evaluation structure, measuring disruption, robustness, and transferability within a unified protocol, is not inherently restricted to the facial domain, and we view the current work as a first step toward extending proactive defense evaluation to other modalities and domains.
>
> [1] Karras, Tero, Samuli Laine, and Timo Aila. "A style-based generator architecture for generative adversarial networks." Proceedings of the IEEE/CVF conference on computer vision and pattern recognition. 2019.
>
> [2] Chen, Xuanhong, et al. "Simswap++: Towards faster and high-quality identity swapping." IEEE Transactions on Pattern Analysis and Machine Intelligence 46.1 (2023): 576-592.

---

> > ### Author Rebuttal · Reviewer_qeeX · 2026-04-02
> >
> > I appreciate the detailed response from authors. Some of my concerns remains, such as the scope of this benchmark. I have noticed similar concerns from other reviewers, and believe the comprehensiveness can be improved. However, adding additional datasets to a benchmark is technically very difficult during such short rebuttal phase. I do acknowledge the merit of this benchmark work and believe this will benefit the deepfake detection community. I will therefore raise my score.

---

> > > ### Author Response · Authors · 2026-04-07
> > >
> > > We are glad to hear that our responses and additional experiments have addressed your concerns. We truly appreciate your time and consideration, as well as the increased score.

---

### Official Review · Reviewer_2kYy · 2026-03-13

**Soundness:** 2
**Presentation:** 3
**Significance:** 3
**Originality:** 2
**Overall Recommendation:** 4
**Confidence:** 2

**Summary:**

This paper constructs a proactive defense benchmark, which systematically evaluate existing deepfake defense methods across multiple generators and metrics. This benchmarks reveals that existing defense methods are much less robust in more realistic scenarios beyond white-box settings.

**Compliance With Llm Reviewing Policy:**

Affirmed.

**Final Justification:**

My concerns have been addressed by the authors. I agree that placing all defenses under a unified protocol is a contribution that can benefit the community. Therefore, I increase my score to 4.

**Key Questions For Authors:**

Please refers to my weaknesses.

**Limitations:**

Limitations of this paper largely overlaps with the weaknesses, so I provide some suggestions here:

1. The authors are encouraged to include more datasets in the evaluation.
2. The authors are encouraged to include a sensitivity analysis on evaluation metrics.

**Strengths And Weaknesses:**

**Strengths**:

1. This benchmark comprehensively evaluates defense methods from multiple perspectives, including disruption, robustness and transferability.
2. This benchmark considers a wide range of practical settings beyond white-box setting.
3. This paper provides valuable insights such as defense methods are much less robust in more realistic scenarios beyond white-box settings.

**Weaknesses**:

1. Relying on a single dataset limits the benchmark’s generality. Table 5 shows that prior proactive defense studies have used a broader range of datasets beyond CelebA-HQ. The authors are encouraged to include more datasets in the evaluation.
2. A follow-up weakness is that, the novelty of this benchmark primarily comes from the new evaluation protocol and the proposed CIDR metric. This benchmark does not introduce a new dataset (and only use one dataset for evaluation) and most of the metrics are adopted from prior work. I am not yet fully convinced that the contribution of this benchmark meets the bar of ICML, but I am not an expert in this field, I will open to other reviewers’ opinions.
3. Lack of discussions on the limitations of this benchmark. The authors are encouraged to discuss the limitations of this benchmark.
4. It is unclear whether the ranking is sensitive to certain metrics. Adding a sensitivity analysis on metrics may be a good idea to investigate the potential bias in evaluation metrics.

I am not an expert in this field, so I will open to other reviewers’ opinions during the rebuttal and would be willing to revise my rating after carefully reading both the authors’ rebuttal and the other reviewers’ comments.

---

> ### Author Rebuttal · Authors · 2026-03-30
>
> We thank the reviewer for the constructive feedback. We are pleased that the reviewer recognized the multi-dimensional evaluation and practical insights our benchmark provides. We address the remaining concerns below.
>
> **[W1] Evaluation on Additional Datasets:** We provide additional experimental results on FFHQ and VGGFace2-HQ datasets and observe consistent trends. We refer the reviewer to the corresponding figures for the other datasets, available at [here](https://anonymous.4open.science/r/ICML2026_9676/image.png), which confirm that analogous patterns hold beyond CelebA-HQ. Please see the reply to Reviewer *qeeX* [W1] due to the space limit.
>
> **[W2] Novelty and Contribution of the Benchmark:** We clarify that the primary contribution of our work lies in establishing a comprehensive and principled evaluation framework for proactive defenses. Benchmark papers proposing novel evaluation frameworks and metrics have been consistently recognized as significant contributions at major ML venues, independent of whether new datasets are introduced. For instance, DeepfakeBench (Yan et al., 2023)  established a standardized evaluation framework for deepfake detection, and has since served as a foundation for subsequent research in that domain. A comparable standard is notably absent in the proactive defense field, and we believe our work fills this gap as a necessary stepping stone for future research.
>
> The key issue we address is not merely which metrics to use, but how to ensure fair and consistent evaluation across methods. In current literature, each defense is evaluated under its own protocol, with different generators, different metrics, and different robustness settings, making it difficult to determine which method actually performs better (Table 4(c)). Our benchmark resolves this by placing all defenses under a unified protocol, adopting established metrics where appropriate. While we adopt widely used metrics to ensure consistency with prior work and enable direct comparison, our contribution goes beyond consolidation. We additionally introduce CIDR (Eq. 2), ROB (Eq. 3), and TE (Eq. 4) where existing metrics fail to provide fair comparison: CIDR corrects for generator-induced identity bias that causes inconsistent rankings across generators (Figure 6), and ROB/TE normalize robustness and transferability scores to make cross-generator comparison meaningful.
>
> **[W3] Discussion on Limitations:** A fundamental limitation of any benchmark lies in its scalability to newly emerging models. In our benchmark, we can consider three axes: datasets, defense methods, and deepfake generators. While our framework is designed to be extensible along three axes, the level of effort required for extension differs across them. Expanding to new datasets or incorporating additional defense methods is relatively straightforward, as they require minimal or no structural changes. However, extending the benchmark to new deepfake generators is more demanding as transferability evaluation must be recomputed across all generator pairs, requiring a full update of the TE dimension. As a result, adapting the benchmark to newly introduced generation paradigms incurs non-trivial computational cost, which is inherent to the nature of transferability evaluation because of pairwise interactions across generators to be measured.
>
> **[W4] Sensitivity Analysis of AvgRank to Metrics:** To examine whether AvgRank is disproportionately influenced by any particular metric, we conducted a leave-one-out sensitivity analysis, systematically excluding each of the four metrics (MSE, LPIPS, CIDR, BRISQUE) one at a time. For each configuration, we measured the resulting ranking changes per generator and computed Kendall's τ, then averaged across generators.
>
> |Except metric|Kendall’s τ|
> |-|-|
> |MSE|0.824|
> |LPIPS|0.794|
> |CIDR|0.729|
> |BRISQUE|0.807|
> ---
> Two observations emerge from this analysis. First, the average τ remains above 0.7 regardless of which metric is excluded, confirming that AvgRank does not over-rely on any single metric. Second, excluding CIDR yields the lowest τ (0.729), indicating that CIDR is not redundant with the other metrics; rather, it captures an independent failure mode specific to identity disruption. This directly supports our central claim in Section 5 that relying on a single metric category can yield conflicting interpretations of defense performance, and empirically demonstrates the necessity of multi-dimensional evaluation.

---

> > ### Author Rebuttal · Reviewer_2kYy · 2026-04-03
> >
> > Thanks for providing detailed responses and experiment results. My concerns have been addressed. I agree that placing all defenses under a unified protocol is a contribution that can benefit the community. Therefore, I will increase my score to 4.

---

> > > ### Author Response · Authors · 2026-04-07
> > >
> > > We sincerely thank the reviewer for the thoughtful feedback and the increased score. We are glad that the merit of our benchmark has been acknowledged.

---

### Official Review · Reviewer_fZU9 · 2026-03-18

**Soundness:** 2
**Presentation:** 2
**Significance:** 2
**Originality:** 1
**Overall Recommendation:** 3
**Confidence:** 3

**Summary:**

The research paper presents the contribution towards AI security in deepfake generation by providing a benchmark framework to evaluate proactive defense methods rather than proposing a new one. It examines how well existing defenses can prevent image misuse by deepfake models, assessing them based on disruption, robustness, and transferability. The authors assess an important domain of generative media security by analyzing whether existing defense mechanisms can prevent the misuse of images by deepfake generation models.
The benchmark evaluates defenses across three main dimensions: disruption (how well the defense prevents generation), robustness (how stable the defense remains under transformations), and transferability (whether the defense generalizes across different deepfake models).

**Compliance With Llm Reviewing Policy:**

Affirmed.

**Key Questions For Authors:**

The authors are requested to provide responses to following comments:
1. The work focuses on evaluation and would be stronger with a novel defense method. If it is already involved in the paper then it has to be explicitly mentioned to give weightage to the work contributed.
2. The mentioned benchmark could be improved by including a wider range of deepfake models.
3. Scalability is a major concern here so it is suggest to perform sclability measure to large diffusion-based models should be explored further.
4. The presented evaluation metrics would benefit from validation on more diverse datasets.
5. The paper requires practical deployment and real-world applicability need deeper discussion.

**Limitations:**

1. It does not clearly propose a new defense method and has limited set of models used in evaluation purpose.
2. There is no such weightage towards scalability to large diffusion models and are limitedly explored.
3. It seems that the metrics may broader validation with more datasets where it also lacks detailed discussion on real-world deployment.

**Strengths And Weaknesses:**

The paper presents a proactive defence mechanism with following strengths as contribution:
1. It is expected to the problem in deepfake security and defense evaluation where it introduces a systematic benchmarking framework.
2. It covers the primary considered aspects such as robustness, disruption, transferability with standardized evaluation.
3. Also, the paper holds empirical testing on various deepfake models
However, the weakness that majorly hinders the paper's strength are:
1. It does not clearly propose a new defense method and has limited set of models used in evaluation purpose.
2. There is no such weightage towards scalability to large diffusion models and are limitedly explored.
3. It seems that the metrics may broader validation with more datasets where it also lacks detailed discussion on real-world deployment.

---

> ### Author Rebuttal · Authors · 2026-03-30
>
> We sincerely appreciate the reviewer's careful reading and valuable comments. We respond to each point below.
>
> **[W1][Q1] Contribution as a Benchmark Paper:** This paper is a benchmark paper that provides a unified evaluation protocol for proactive defense methods against deepfake. Our main contribution lies in systematically comparing existing methods under consistent conditions, which has been lacking in this area. Recent benchmark papers have contributed through systematic evaluation and revealed important gaps through rigorous benchmarking without proposing new methods[1-3]. Closer to our work, DeepfakeBench (Yan et al., 2023) unified fragmented evaluation standards for deepfake detection. We believe a similar effort is needed for proactive defense. As shown in Table 4(c), existing evaluations typically cover only a subset of key dimensions, and each prior work uses different deepfake generation models and settings, making fair comparison difficult. Our benchmark aims to address this gap by providing consistent evaluation conditions across methods. Additionally, we introduce three new metrics that extend beyond commonly used measures: CIDR (Eq. 2), which corrects for generator-induced identity bias; ROB (Eq. 3) and TE (Eq. 4), which quantify robustness and transferability as normalized ratios.
>
> **[W2][Q2][Q3] Scalability to Diffusion-Based Models:** To further validate the generalizability of our benchmark, we additionally evaluated two large diffusion-based face swap models, DiffSwap (Zhao et al., 2023) and DiffFace (Kim et al., 2025). The results on CelebA-HQ are shown below.
>
> ||Metric|PGD|Disrupting|DF-RAP|Anti|Latent|SCOL|NullSwap|
> |:-:|:-:|:-:|:-:|:-:|:-:|:-:|:-:|:-:|
> |**DiffSwap**|MSE↑|0.0046|0.0046|0.0047|0.0043|0.0097|0.0051|0.0053|
> ||LPIPS↑|0.0293|0.0283|0.0294|0.0309|0.0693|0.0309|0.0316|
> ||CIDR↑|0.0363|0.0425|0.0459|0.0430|0.2674|0.1139|0.2131|
> ||BRISQUE↑|4.0007|3.8500|4.0877|3.5432|3.5636|3.8448|3.7283|
> ||AvgRank↓|5.25|5.25|3.50|5.50|2.25|3.50|2.75|
> |**DiffFace**|MSE↑|0.0069|0.0064|0.0068|0.0065|0.0142|0.0101|0.0103|
> ||LPIPS↑|0.0282|0.0269|0.0283|0.0281|0.0528|0.0395|0.0410|
> ||CIDR↑|0.0359|0.0385|0.0534|0.0361|0.6913|0.4120|0.5628|
> ||BRISQUE↑|1.8878|1.7522|1.7187|1.7217|1.3969|2.0021|1.8001|
> ||AvgRank↓|4.50|5.75|4.75|5.75|2.50|2.50|2.25|
>
> These additional experiments reveal a previously underexplored discrepancy between output-space and latent-space attacks. Specifically, output-space methods (e.g., PGD) exhibit a significant drop in effectiveness on diffusion-based face-swap models, whereas latent-space methods (e.g., Latent Attack, NullSwap) remain consistently strong. This gap can be attributed to the iterative sampling process of diffusion models, which progressively attenuates input-level perturbations. We are grateful for the reviewer’s insight to include diffusion-based models, as these results have effectively shown the value of our benchmark in identifying architecture-specific vulnerabilities.
>
> **[W3][Q4] Evaluation on Additional Datasets:** We conducted additional experiments on FFHQ [1] and VGGFace2-HQ [2]. The relative ranking of defense methods remains largely consistent across different datasets, suggesting that performance ordering is driven primarily by the inherent characteristics of each method rather than dataset choice. Please see the detailed tables in our reply to Reviewer *qeeX* [W1] due to the space limit. Additionally, while Figure 1 was produced using the CelebA-HQ dataset, we confirmed that analogous patterns hold for the other datasets. Corresponding figures are available at [here](https://anonymous.4open.science/r/ICML2026_9676/image.png).
>
> **[Q5] Real-World Deployment Implications:** The core practical contribution is the systematic exposure of the vulnerability of current proactive defense methods in real-world scenarios. Most methods exhibit significant performance degradation under both post-processing and cross-generator transfer, providing a clear warning against deploying existing methods as-is and offering guidance for future research. Specifically, white-box performance on a single generator is insufficient; transferability and robustness must be jointly considered. While prior work has examined transferability, our results show that performance degrades far more severely under both cross-generator and post-processing conditions, indicating that these limitations have been substantially underestimated.
>
> [1] Carlini, Nicholas, et al. "AutoAdvExBench: Benchmarking Autonomous Exploitation of Adversarial Example Defenses." International Conference on Machine Learning. PMLR, 2025.
>
> [2] Sun, Guangzhi, et al. "CASE-Bench: Context-Aware SafEty Benchmark for Large Language Models." International Conference on Machine Learning. PMLR, 2025.
>
> [3] Kuntz, Thomas, et al. "OS-Harm: A Benchmark for Measuring Safety of Computer Use Agents." The Thirty-ninth Annual Conference on Neural Information Processing Systems Datasets and Benchmarks Track.

---

### Decision · Program_Chairs · 2026-04-30

**Decision:**

Accept (regular)

**Comment:**

The recommendation is based on the reviewers' comments, the area chair's evaluation, and the author-reviewer discussion.

This paper studies the robustness of deepfake generation under a unified evaluation protocol. All reviewers find the studied setting novel and the results provide new insights. The authors’ rebuttal has successfully addressed the major concerns of reviewers. In the post-rebuttal phase, most reviewers were satisfied with the authors’ responses and agreed on the decision of acceptance. The AC also checked the author's response to the review without reviewer acknowledgment, and the AC believes the concerns were properly addressed.

Overall, I recommend acceptance of this submission. I also expect the authors to include the new results and suggested changes during the rebuttal phase in the final version.